# Prolonged visual experience accelerates developmental synaptic downscaling via epigenetic regulation and Rab5c mediated AMPA receptor trafficking

Lijun Zheng[1,2,3,4], Xinyi Duan[1,4], Wenyi Huang[1], Yuhao Luo[1], Yufei Wu[1], Qihui Lin[1], Xiaohua Wu[1], Lu Han[1] & Wanhua Shen [1] ✉

Environmental light significantly influences neural development, yet the specific mechanisms underlying the effects of prolonged visual experience on homeostatic synaptic scaling remain unclear. Using manipulated ambient light conditions, we observed reduced mEPSC amplitudes and visually evoked responses in 20 hr light/4 hr dark (20LE) compared to a standard 12 hr light/12 hr dark (12LE) reared *Xenopus laevis* tadpoles. Prolonged light exposure accelerates the developmental decline of glutamatergic synaptic transmission via Rab5c-dependent endocytosis of AMPA receptor (AMPAR) subunits GluA1 and GluA2. The synaptic changes were accompanied by increased intrinsic neuronal excitability, but unchanged presynaptic release probability, and coincided with altered dendritic architecture. Notably, synaptic transmission and AMPAR expression were reversible upon re-exposure to standard 12LE conditions. Class I HDAC-mediated histone acetylation links epigenetic regulation to sustained AMPAR downregulation, revealing a two-stage process in which prolonged visual experience drives homeostatic synaptic downscaling through coordinated transcriptional/ epigenetic mechanism and Rab5c-mediated trafficking.

Neuronal circuit development is profoundly shaped by environmental stimuli, with visual experience playing a critical role in modulating synaptic transmission and network function[1–10]. A key mechanism in this process is homeostatic plasticity, which ensures network stability by adjusting intrinsic neuronal excitability and synaptic strength in response to environmental cues[7,11,12]. Visual stimulation drives the activity-dependent refinement of synaptic connections, which is essential for proper sensory processing and circuit maturation[5,6,13]. While the impact of visual patterns on synaptic plasticity is well-established, we therefore investigated how prolonged ambient light exposure (20 h light/4 h dark, 20LE) or partial visual deprivation (4 h light/20 h dark, 4LE) might influence synaptic scaling and underlying mechanisms during early brain development.

Patterned visual stimulation, such as exposure to 1 Hz flickering light for several hours, has been shown to enhance intrinsic neuronal excitability while reducing AMPA receptor (AMPAR)-mediated synaptic drive, thereby influencing circuit development through Hebbian or homeostatic mechanisms[8,14–18]. Synaptic scaling refers to a form of homeostatic plasticity

in which neurons adjust the strength of all their excitatory synapses up or down to stabilize overall activity. Experience-dependent synaptic scaling induced by visual deprivation, requires the coordinated regulation of synaptic strength and neuronal excitability to preserve network stability[7,12]. There is ongoing debate about the specific triggers of synaptic scaling, including whether they arise from changes in synaptic transmission, shifts in firing rate, or alterations in glutamate receptor turnover[19]. Identifying the molecular pathways that control synaptic scaling, particularly the mechanisms involved in reversible scaling, a bidirectional form of synaptic scaling that up- or down-regulates synaptic strength in response to altered and restored sensory input, is crucial for understanding the broader dynamics of activity-dependent synaptic regulation[20]. Therefore, we examined the impact of a single 20-h or 4-h episode of altered light exposure on synaptic function and its effects on AMPAR trafficking in the early developing brain.

Long-term depression (LTD) and long-term potentiation (LTP) rely on AMPAR removal or insertion, mediated by clathrin- or dynamin-

[1]Zhejiang Key Laboratory of Organ Development and Regeneration, College of Life and Environmental Sciences, Hangzhou Normal University, Hangzhou, Zhejiang, China. [2]Chinese Institute for Brain Research, Beijing, China. [3]School of Basic Medical Sciences, Capital Medical University, Beijing, China. [4]These authors contributed equally: Lijun Zheng, Xinyi Duan. ✉e-mail: shen@hznu.edu.cn

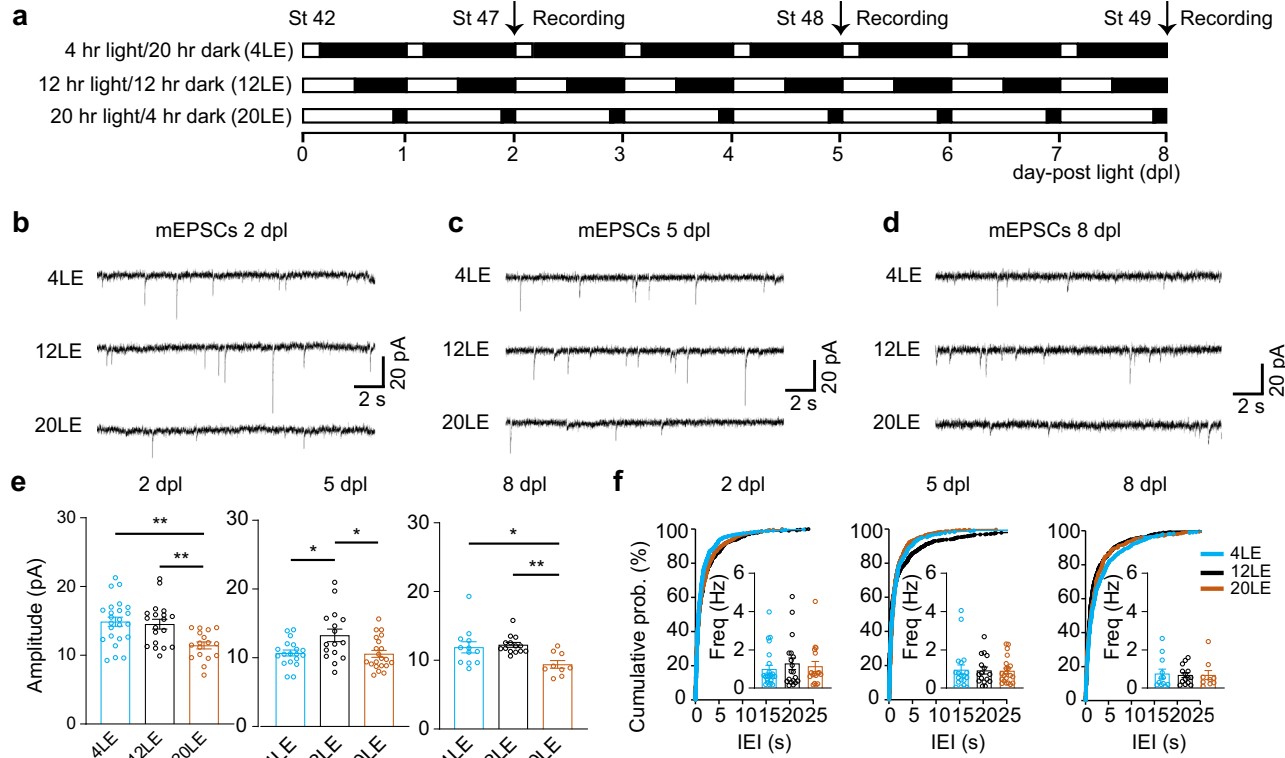

**Fig. 1 | Prolonged light exposure reduces mEPSC amplitudes in tectal neurons.** **a** Schematic representation of light exposure conditions: control (12LE), long-day exposure (20LE), and short-day exposure (4LE). **b–d** Representative mEPSCs traces from tectal neurons exposed to 4LE, 12LE, and 20LE at 2 dpl (**b**), 5 dpl (**c**), and 8 dpl (**d**). Scale bar: 20 pA, 2 s. **e** Quantitative analysis shows that mEPSCs amplitudes decrease in 20LE tadpoles compared to 12LE tadpoles at 2, 5, and 8 dpl. 2 dpl: n = 24,

19, 17; 5 dpl: n = 19, 16, 21; 8 dpl: n = 12, 14, 9 for 4LE, 12LE, and 20LE. **f** No significant changes were observed in inter-event intervals (IEIs) or mEPSC frequencies for 20LE-treated neurons compared to 12LE across all time points. *p < 0.05, **p < 0.01 (Kolmogorov–Smirnov test or ANOVA with Newman–Keuls post hoc test).

dependent mechanisms and lateral diffusion[21–28]. AMPAR trafficking is regulated by the cytoplasmic carboxyl terminus of the receptor, and disruptions to this process reduce excitatory transmission and impair synaptic plasticity[6,10,29]. Numerous molecules or pathways are involved in AMPAR trafficking during synaptic scaling, including Arc/Arg3.1[4,30–32], Ras/Rap[33,34], cadherin–catenin complexes[35], AKAP150[36], and Plk2[37]. The Rab family of small GTPases plays distinct roles in AMPAR dynamics, with Rab5 facilitating receptor internalization during LTD and Rab8/11/39B mediating insertion during LTP[38–42]. Rab5 is a key regulator of early endosome formation and AMPAR endocytosis, linking receptor internalization to homeostatic and activity-dependent synaptic adjustments. Recycling endosomes from the plasma membrane have been implicated in transporting AMPARs during synaptic modification[43–50].

Histone deacetylases (HDACs) modify histones to regulate gene expression for specific cellular functions, and these epigenetic mechanisms contribute to lasting synaptic plasticity and memory consolidation[51–54]. While the role of histone acetylation in synaptic scaling and circuit development remains less clear, our prior work has shown that HDAC1 regulates activity-dependent synaptic and structural plasticity[2], and HDAC2 and HDAC3 are involved in visual experience-dependent radial glia proliferation[55]. Given that histone modification is known to regulate AMPAR expression and memory[56–58], we specifically examined the role of Class I HDACs in modulating synaptic transmission following altered light exposure.

Our findings demonstrate that a progressive decrease in miniature excitatory postsynaptic current (mEPSC) amplitudes in tectal neurons from stages 42 to 49 exhibits a hallmark feature of synaptic downscaling during development. Prolonged light exposure (20LE) accelerated the reduction of excitatory synaptic transmission, and this effect was reversed when tadpoles

were returned to a standard 12-h light/12-h dark cycle (12LE) at later stages. This synaptic scaling was driven by increased light exposure and Rab5c-mediated AMPAR removal. Furthermore, prolonged light conditions increased histone acetylation at specific residues, including histone H3 acetylation at lysine 9 (H3K9Ac), histone H2B acetylation at lysine 5 (H2BK5Ac), and histone H4 acetylation at lysine 8 (H4K8Ac). GluA1 and GluA2 expression levels were modulated by HDAC2/3 activity and pharmacological HDAC inhibition. Collectively, these findings highlight the key roles of Rab5c-dependent AMPAR trafficking and the coordinated epigenetic regulation of GluA1 and GluA2 in driving visual experience-dependent synaptic downscaling and neural circuit maturation.

## Results
### Prolonged light exposure accelerates a developmental decline in glutamatergic synaptic transmission

Activity-dependent neural refinement plays a key role in shaping developing circuits, as observed in patterns of 4 hr patterned visual stimulation at 1 Hz[10,12,18]. To explore how ambient light conditions impact neural development, we initiated differential light exposure in stage 42 tadpoles, when retinal axons first innervate the tectum, begin responding to visual stimulation, and visually driven behaviors emerge[59]. Tadpoles were exposed to three ambient light conditions: control (12 h light/12 h dark, 12LE), prolonged light (20 hr light/4 hr dark, 20LE), and reduced light (4 h light/20 h dark, 4LE) for periods of 2, 5, and 8 days post-light exposure (dpl), continuing through stage 49 to span the critical period of tectal circuit refinement (Fig. 1a).

To assess how development and modified light exposure affect excitatory synaptic transmission, we recorded mEPSCs holding at −60 mV in tadpoles subjected to 4LE, 12LE, and 20LE light conditions

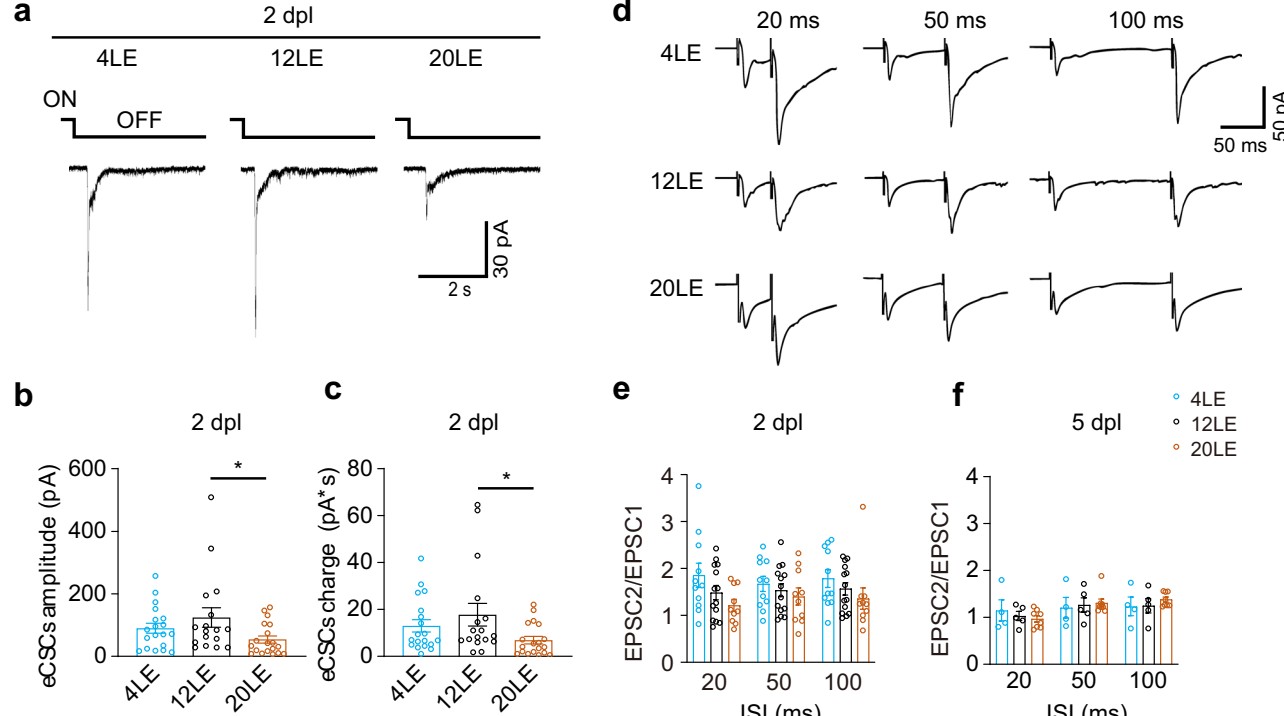

**Fig. 2 | Long-day exposure decreases excitatory synaptic inputs. a** Representative trace of visually evoked excitatory compound synaptic currents (eCSCs) in response to full-field light off visual stimuli (luminance: 20 cd m$^{-2}$) in 4LE, 12LE, and 20LE-treated neurons at 2 dpl. Scale bar: 30 pA, 2 s. **b, c** Quantitative analyses showing that both the amplitude (**b**) and integrated current (**c**) of eCSCs were significantly lower in the 20LE-treated neurons compared to 12LE. $n$ = 19, 17, 19 for 4LE, 12LE, and 20LE. **d** Representative electrophysiological recordings of EPSCs in response to paired-pulse stimuli (intervals of 20, 50, and 100 ms) in neurons from 4LE, 12LE, and 20LE groups at 2 dpl. Stimulus artifacts were removed for clarity. Scale bar: 50 pA, 50 ms. **e, f** No significant differences in paired-pulse ratio (EPSC2/EPSC1) were observed at 2 dpl (**e**) or 5 dpl (**f**) among the three groups. 2 dpl: $n$ = 11, 14, 10; 5 dpl: $n$ = 4, 5, 8 for 4LE, 12LE, and 20LE. *$p < 0.05$.

across the exposure period to track changes in glutamatergic transmission (Fig. 1b–d). A significant decrease in mEPSC amplitude was observed at 8 dpl compared to 2 dpl (Supplementary Fig. 1a–c), while mEPSC frequency remained relatively unchanged (Supplementary Fig. 1d–f), indicating a developmental downregulation of excitatory synaptic drive during tectal maturation. Additionally, tadpoles in the 20LE group exhibited a significant reduction in mEPSC amplitude compared to 12LE across all time points (Fig. 1e), with frequency remaining unaffected (Fig. 1f). These results suggest that prolonged light exposure amplifies the endogenous developmental decrease in excitatory strength. Conversely, 4LE tadpoles showed no detectable changes in mEPSC amplitude or frequency, likely due to the relatively mild reduction in light compared with complete visual deprivation (0LE). Under 0LE conditions, we observed a significant increase in mEPSC amplitude (Supplementary Fig. 2) with unaltered frequency, consistent with homeostatic synaptic upscaling. Together, these findings imply that prolonged light exposure accelerates developmental synaptic downscaling while visual deprivation leads to synaptic upscaling.

### Prolonged light exposure decreases visually evoked synaptic responses

Tectal neurons receive direct excitatory retinal inputs and compound synaptic currents within tectal circuits[23,60,61]. To determine whether altered light exposure affects visually evoked synaptic responses, we recorded excitatory compound synaptic currents (eCSCs) from tectal neurons voltage-clamped at –60 mV and induced by full-field visual stimulation (Fig. 2a) as previously described[1]. In 20LE tadpoles, both the retinotectal input amplitude and the integrated charge of recurrent synaptic responses were significantly reduced compared with 12LE controls, while 4LE tadpoles showed no notable change (Fig. 2b, c). These data demonstrate that prolonged light exposure weakens excitatory synaptic drive onto tectal cells,

while reduced light exposure does not measurably alter visually evoked responses.

To determine if the developmental changes in excitatory currents were due to altered presynaptic release probability, we measured paired-pulse ratios at inter-stimulus intervals (ISIs) of 20, 50, and 100 ms at 2 dpl (Fig. 2d). No significant differences were found across all conditions at both 2 dpl and 5 dpl (Fig. 2e, f), indicating that changes in neurotransmitter release probability are not responsible for the observed reduction in synaptic responses.

### Long-day exposure increases intrinsic neuronal excitability

Reduction of excitatory synaptic transmission may require increased intrinsic excitability to ensure effective neuronal spiking[18]. To assess how intrinsic excitability responds to varying ambient light exposure, we recorded action potentials generated by depolarizing current injections (Fig. 3a). Neurons were subjected to a series of 100 ms depolarizing currents of progressively higher intensity. The number of action potentials generated was counted for each neuron. We found that the 20LE group exhibited a significant increase in intrinsic excitability at all observed time points at 2, 5, and 8 dpl (Fig. 3b–d), while no changes were noted in the 4LE group. Membrane properties, including capacitance (Cm), input resistance (Rin), and resting membrane potential (Vm), were unaffected at 2 dpl (Fig. 3e–g) and 5 dpl or 8 dpl (Supplementary Fig. 3) tadpoles. These results suggest that prolonged light exposure enhances intrinsic excitability without altering passive membrane properties of tectal neurons.

### Transcriptome analysis of light exposure-associated genes

Homeostatic plasticity relies on transcriptional regulation to maintain neural network stability[62–64]. To investigate how light exposure influences early transcriptional responses that may initiate synaptic scaling, we conducted RNA sequencing (RNA-seq) on brain samples collected at 2 and 5

dpl. Expression levels were then compared across three light exposure conditions of 4LE, 12LE, and 20LE. Our analysis identified and annotated 52,426 genes expressed in the optic tectum of tetraploid *Xenopus laevis*, including two homeologous gene sets, designated L and S for the longer and shorter homeologs, respectively.

To understand the transcriptional dynamics, we then sorted the up- and down-regulated genes and conducted a functional annotation analysis

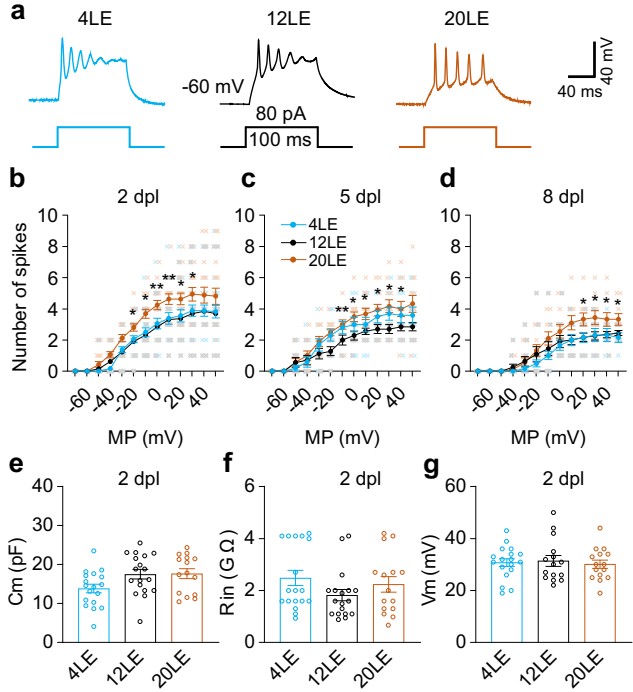

**Fig. 3 | Long-day exposure increases intrinsic neuronal excitability.**
**a** Representative traces of action potentials induced by current injection in neurons from 4LE, 12LE, and 20LE groups. Scale bar: 40 mV, 40 ms. **b–d** Summary data showing that 20LE, but not 4LE, increases intrinsic neuronal excitability compared to 12LE neurons at 2 dpl (**b**), 5 dpl (**c**), and 8 dpl (**d**). 2 dpl: $n = 13, 17, 18$; 5 dpl: $n = 13, 12, 10$; 8 dpl: $n = 11, 9, 6$ for 4LE, 12LE, and 20LE. MP: membrane potential. **e–g** Membrane capacitance (Cm), input resistance (Rin), and resting membrane potential (Vm) were not significantly altered among 4LE, 12LE, and 20LE-treated neurons at 2 dpl. 4LE/12LE/20LE: $n = 18, 22, 15$. $*p < 0.05$, $**p < 0.01$.

of these transcriptional changes (Fig. 4a). We identified 515 differentially expressed genes (DEGs) in the 20LE condition in comparison with 12LE group (Supplementary Fig. 4 and Supplementary Data 1) at 2 dpl. Out of these DEGs, 230 (44%) were upregulated and 285 (54%) were downregulated (Fig. 4b). In contrast, the 4LE condition triggered fewer changes, with only 64 upregulated and 17 downregulated genes at 2 dpl (Supplementary Data 1). Venn diagram analysis indicated that the 20LE condition induced more upregulated genes at 2 dpl than at 5 dpl (Fig. 4b, c). By 5 dpl, the number of DEGs in the 20LE condition was significantly reduced, and a similar trend was observed in the 4LE group (Supplementary Data 1), suggesting that transcriptional responses to light exposure diminish over time. Functional enrichment analysis of DEGs from the 20LE versus 12LE comparison at 2 dpl highlighted key biological processes, including glutamatergic synaptic transmission (GO:0035249) and AMPA glutamate receptor activity (GO:0004971), implicating these pathways in light-dependent modulation of synaptic scaling (Supplementary Fig. 5 and Supplementary Data 1).

To further pinpoint genes linked to excitatory synaptic transmission, we analyzed the expression levels of glutamate receptors. Results indicated a decrease in expression of both *GluA1* and *GluA2* (listed as *gria1.L* and *gria2.S*) in 20LE tadpoles compared to 12LE tadpoles at 2 dpl but not at 5 dpl (Supplementary Data 1). To validate the reliability of the transcriptome sequencing data, we further analyzed these two DEGs using qRT-PCR with GAPDH as an internal control. RNA-seq analysis and qRT-PCR produced consistent decreases in *gria1.L* and *gria2.S* expression in the 20LE group at 2 dpl (Fig. 4d and Supplementary Fig. 6). However, no significant changes were observed in either gene at 5 dpl in the 20LE or 4LE group (Fig. 4e, Supplementary Fig. 7, and Supplementary Data 1), consistent with the RNA-seq results. These findings support the notion that prolonged light exposure reduces synaptic transmission by downregulating gene expression, particularly *GluA1* and *GluA2*, thereby supporting the role of light-induced early transcriptional changes in modulating synaptic activity.

## Impact of prolonged light exposure on AMPAR expression

To assess how prolonged light exposure influences the protein expression of AMPA receptor subunits GluA1 and GluA2, we performed Western blot analysis on brain samples using antibodies against each subunit (Fig. 5a). We found that the protein levels of GluA1 and GluA2 were significantly decreased at 5 and 8 dpl in 20LE tecta (Fig. 5b). Because unbiased GO enrichment of our RNA-seq dataset highlighted vesicle-mediated transport and receptor trafficking pathways, and prior studies implicated Rab5 family proteins in AMPAR endocytosis via endosomes[38,65], we examined Rab5c

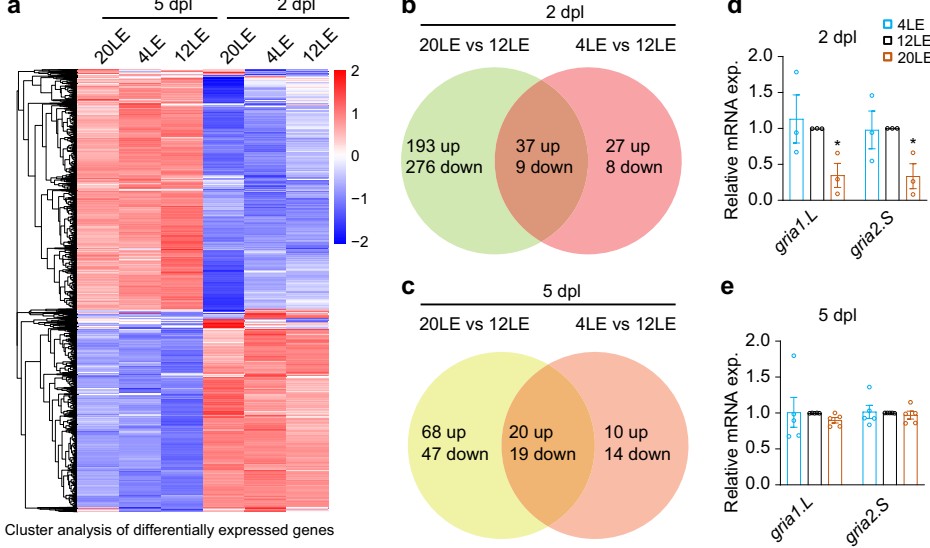

**Fig. 4 | Transcriptomic changes induced by altered light exposure.** **a** Heatmap showing gene expression in tectal neurons exposed to 4LE, 12LE, and 20LE at 2 dpl and 5 dpl. **b, c** Venn diagrams showing the intersections between upregulated and downregulated genes in the tectum under 20LE, 4LE, and 12LE conditions at 2 dpl (**b**) and 5 dpl (**c**). **d, e** qRT-PCR analysis showing changes in *gria1.L* and *gria2.S* expression levels at 2 dpl (**d**) and 5 dpl (**e**), 2 dpl: $n = 3, 3$; 5 dpl: $n = 5, 5$ for *gria1.L* and *gria2.S*. $*p < 0.05$.

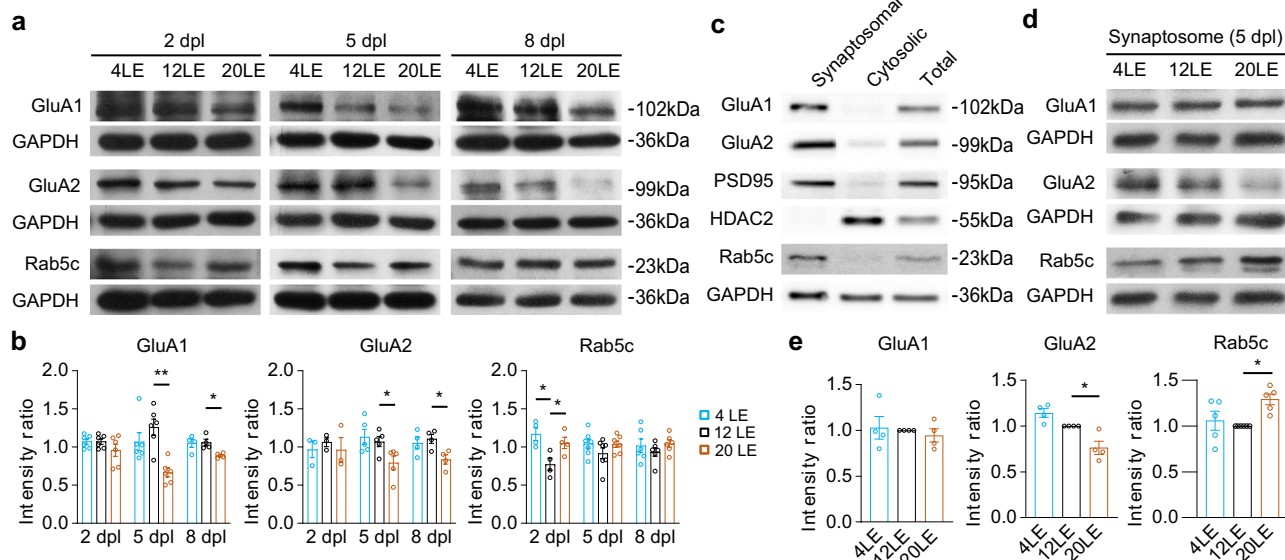

**Fig. 5 | Long-day exposure changes the expression levels of GluA1, GluA2 and Rab5c. a** Representative Western blots showing GluA1, GluA2, and Rab5c protein levels in tadpoles treated with 4LE, 12LE, and 20LE at 2, 5, and 8 dpl. **b** Quantitative analysis indicates a significant decrease in GluA1 and GluA2 expression in 20LE-treated tadpoles compared to 12LE at 5 and 8 dpl, whereas Rab5c is transiently upregulated at 2 dpl in 20LE. GluA1, GluA2 and Rab5c: $n = 3–7$ per group at each time point (two-way ANOVA with post hoc Fisher's LSD test). **c** Representative Western blots of GluA1, GluA2, PSD95, HDAC2 and Rab5c in synaptosomal, cytosolic and total input fractions. **d, e** Synaptosomal analysis shows reduced GluA2 and increased Rab5c levels in 20LE tadpoles compared to 12LE. GluA1: $n = 4, 4, 4$; GluA2: $n = 4, 4, 4$; Rab5c: $n = 5, 6, 5$ for 4LE, 12LE, and 20LE. *$p < 0.05$; **$p < 0.01$ (one-way ANOVA with post hoc Dunnett's test).

and found that it was transiently upregulated at 2 dpl in 20LE, indicating a potential link between Rab5c and AMPAR trafficking under prolonged light exposure (Fig. 5b).

To further investigate the specific effects of light exposure on AMPAR trafficking, we performed Western blot analysis for GluA1, GluA2, and Rab5c on purified tectal synaptosomes, a subcellular fraction enriched in pre- and postsynaptic proteins[66,67], the preparation of which was validated by the detection of GluA1, GluA2, PSD95, HDAC2 and Rab5c (Fig. 5c). Results revealed a selective decrease in synaptic GluA2 expression, with no notable change in GluA1, within the 20LE tectal synaptosomes (Fig. 5d, e). This finding suggests that prolonged light exposure selectively reduces GluA2-containing AMPA receptors at the synaptic level. Additionally, Rab5c expression was significantly elevated in the 20LE synaptosomes, suggesting its recruitment to the synaptic region to facilitate GluA2-containing AMPAR trafficking. Together, these findings align with observed decreases in mEPSC amplitudes from recorded neurons, reinforcing the idea that Rab5c-mediated AMPAR trafficking plays a critical role in synaptic scaling within tectal neurons in response to prolonged light exposure.

### Rab5c regulates neuronal structure development

To assess the role of Rab5c in dendritic arbor development, we performed in vivo imaging on neurons with either Rab5c-GFP overexpression or Rab5c knockdown in tadpoles exposed to regular light (12LE, Fig. 6a) and prolonged light (20LE, Fig. 6b). The knockdown efficiency of Rab5c by Rab5c-MO was confirmed by Western blot analysis (Supplementary Fig. 8a, b). Neurons were imaged 2–3 days post-transfection, a window chosen because it coincides with the earliest functional signatures of Rab5c-dependent synaptic scaling, and aligns with our 2 dpl rescue experiments. Our 3D reconstructions of individual neurons showed that prolonged light exposure (20LE) led to a marked increase in total dendritic branch length (TDBL), branch tip number (TBTN), and dendritic area compared to 12LE control neurons (Fig. 6c–e). The 20LE treatment-induced increase of TDBL and TBTN was reduced by Rab5c-GFP overexpression (Fig. 6c, d). Conversely, Rab5c knockdown significantly elevated TDBL, TBTN, and dendritic areas compared to Ctrl-GFP neurons in 12LE conditions (Fig. 6c–e).

Interestingly, while Rab5c knockdown in 20LE blocked the increase in TBTN (Fig. 6d), it led to an overall expansion in dendritic area compared to Ctrl-GFP-expressing neurons (Fig. 6e). These results indicate that Rab5c modulates light-induced changes in dendritic architecture, highlighting its regulatory role in activity-dependent neuronal development.

### Rab5c-mediated synaptic downscaling is reversible

To assess whether the synaptic downscaling effects of prolonged light exposure could be reversed, we analyzed mEPSCs in tadpoles raised in 20LE condition for two days and then returned to 12LE for two additional days (Fig. 7a). The results showed that the reduced mEPSC amplitudes observed in 20LE-reared tadpoles were fully restored when they were switched back to the 12LE condition (Fig. 7b). Moreover, Rab5c-GFP overexpression further decreased mEPSC amplitudes when compared to 20LE tadpoles (Fig. 7b). Notably, recordings revealed that 20 out of 27 Rab5c-GFP overexpressing neurons became inactive, suggesting that Rab5c might facilitate AMPAR trafficking and synaptic downscaling. Conversely, Rab5c knockdown via Rab5c-MO increased mEPSC amplitudes at both 2 dpl and 5 dpl time points (Fig. 7a–c), underscoring Rab5c's crucial role in AMPAR regulation during early brain development. Supporting these observations, electrophysiological recordings of eCSCs in response to full-field visual stimuli indicated that Rab5c-overexpressing neurons exhibited significantly reduced eCSC amplitudes and charge transfer, while Rab5c knockdown resulted in increases of both eCSC amplitudes and charges (Fig. 7d–f).

To explore the impact of light exposure on AMPAR expression, we performed Western blot analysis on total inputs (Supplementary Fig. 9) or synaptosomal extracts (Fig. 7g) from tadpoles raised in 20LE conditions (20LE + 20LE) and those subsequently returned to 12LE cycle (20LE + 12LE). The analysis showed that the decreases in both GluA1 and GluA2 expression levels were abolished upon the return to the standard 12LE light cycle (Fig. 7h, i and Supplementary Fig. 9). Notably, synaptosomal GluA1 and GluA2 levels significantly increased when tadpoles were re-exposed to the 12LE light condition (Fig. 7h, i), reinforcing the role of AMPAR trafficking in the adaptive regulation of neural circuits in response to environmental light changes. These findings emphasize the involvement of

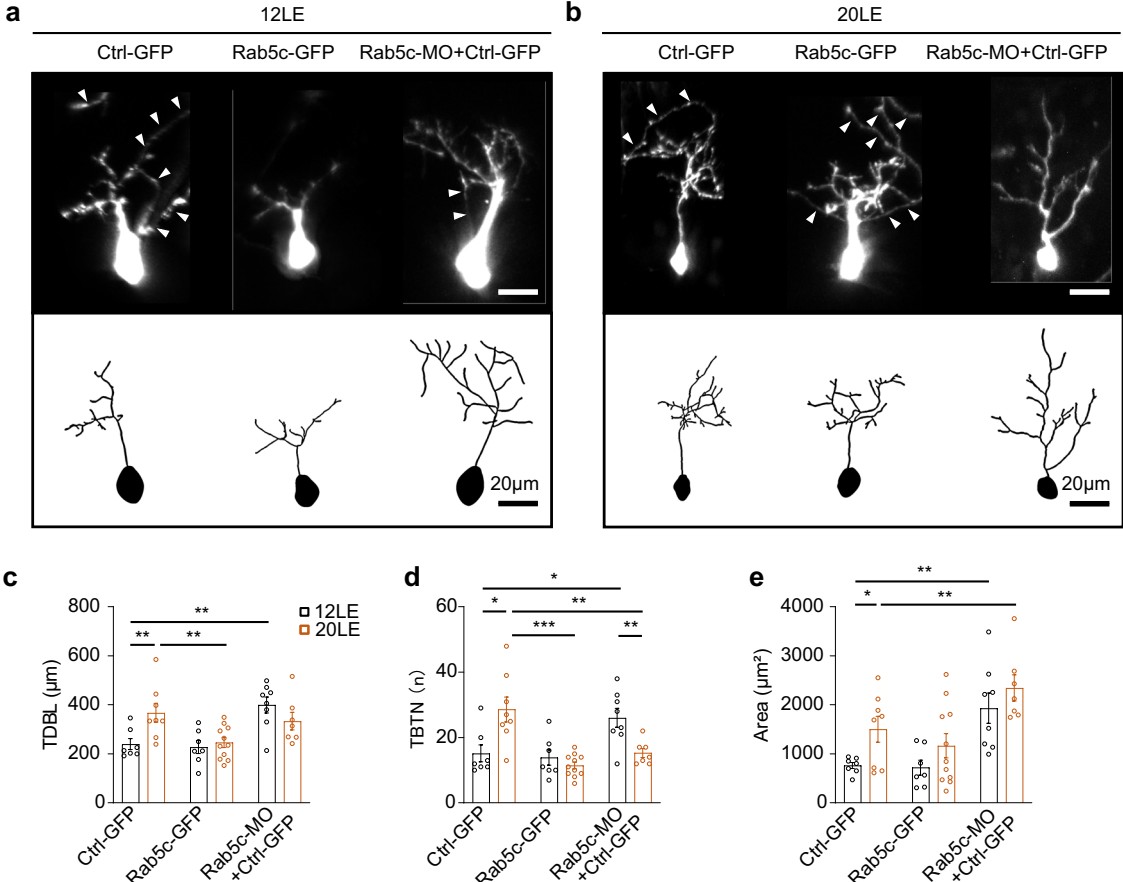

**Fig. 6 | Prolonged light exposure alters neuronal architecture. a, b** Illustrative examples of complete dendritic arbor reconstructions from time-lapse imaging of control (Ctrl-GFP), Rab5c-overexpressing (Rab5c-GFP), and Rab5c knockdown (Rab5c-MO) tectal neurons in 12LE and 20LE tadpoles. Upper panels: raw confocal microscopy images; lower panels: 3D reconstructions using Imaris. Arrowheads indicate axons. **c–e** Quantitative analysis of total dendritic branch length (TDBL), branch tip number (TBTN), and dendritic area. 12LE: $n$ = 7, 7, 8; 20LE: $n$ = 8, 11, 7 for Ctrl-GFP, Rab5c-GFP and Rab5c-MO + Ctrl-GFP. *$p < 0.05$, **$p < 0.01$, ***$p < 0.001$ (two-way ANOVA with post hoc Tukey's HSD test).

Rab5c and AMPAR dynamics in the homeostatic regulation of synaptic strength triggered by alterations in light exposure.

### Prolonged light exposure increases histone acetylation levels
HDACs play a pivotal role in modulating long-term synaptic strength, with epigenetic mechanisms influencing the expression of AMPAR[51,52,56,57]. Because histone acetylation is one of the most dynamic and reversible epigenetic marks linking neuronal activity to transcriptional regulation, we examined histone acetylation changes in tadpoles exposed to 4LE, 12LE, and 20LE cycles at 2, 5, and 8 dpl. Histone acetylation levels were measured by Western blot using antibodies against H3K9Ac, H2BK5Ac, and H4K8Ac on brain homogenates at 2, 5, and 8 dpl (Fig. 8a). Long-day light exposure (20LE) significantly increased acetylation levels at all three histone sites across the examined time (Fig. 8b–d). In contrast, short-day exposure (4LE) showed minimal effects, with decreases observed only in H3K9Ac at 2 dpl and H4K8Ac at 8 dpl. These results suggest that histone acetylation may contribute to synaptic modifications induced by altered light exposure.

### Prolonged light exposure reduces synaptic transmission and glutamatergic receptors through HDAC2/3 mediation
To assess the role of HDAC2 and HDAC3 in regulating glutamatergic synaptic transmission, we recorded mEPSCs from tectal neurons in tadpoles reared under 12LE cycle. Neurons were electroporated with morpholinos targeting HDAC2 (HDAC2-MO) or HDAC3 (HDAC3-MO) and knockdown efficiency was verified by Western blot analysis (Supplementary Fig. 8c–f). Neurons with HDAC2 or HDAC3 knockdown exhibited significantly

reduced mEPSC amplitudes (Fig. 9a, b), consistent with reduced GluA2 expression (Fig. 9j–m). Conversely, overexpression of HDAC1, HDAC2 or HDAC3 increased mEPSC amplitudes compared with Ctrl-GFP neurons (Fig. 9c, d). These results suggest that Class I HDACs play critical roles in maintaining synaptic strength, in line with the prolonged light-induced changes in mEPSCs.

To determine whether HDACs influence intrinsic neuronal excitability, we measured action potential firing rates in 12LE-reared neurons treated with the HDAC inhibitor, trichostatin A (TSA, 10 nM) for two days (Fig. 9e). TSA treatment resulted in a significant increase in action potential firing rates, indicating enhanced intrinsic excitability (Fig. 9f). These results align with the heightened excitability observed in neurons exposed to prolonged light, suggesting a shared mechanism involving HDAC regulation.

We next examined how HDAC inhibition affects AMPAR expression. Western blot analysis showed that TSA treatment significantly reduced GluA1 and GluA2 protein levels in 12LE tadpoles (Fig. 9g–i). Similarly, knockdown of HDAC3 decreased GluA1/2 expression while increasing Rab5c levels, whereas HDAC2 knockdown selectively reduced GluA2 expression (Fig. 9j–m). In contrast, Rab5c knockdown increased histone acetylation at H3K9, H2BK5, and H4K8, indicating reciprocal regulation between endosomal trafficking and epigenetic control. Together, these results highlight the coordinated roles of HDAC2 and HDAC3 in AMPAR subunit expression and synaptic strength. Prolonged light exposure likely reduces glutamatergic receptor levels through HDAC-dependent mechanisms, establishing a functional link between HDAC activity, AMPAR trafficking, and sensory experience-driven synaptic adaptation.

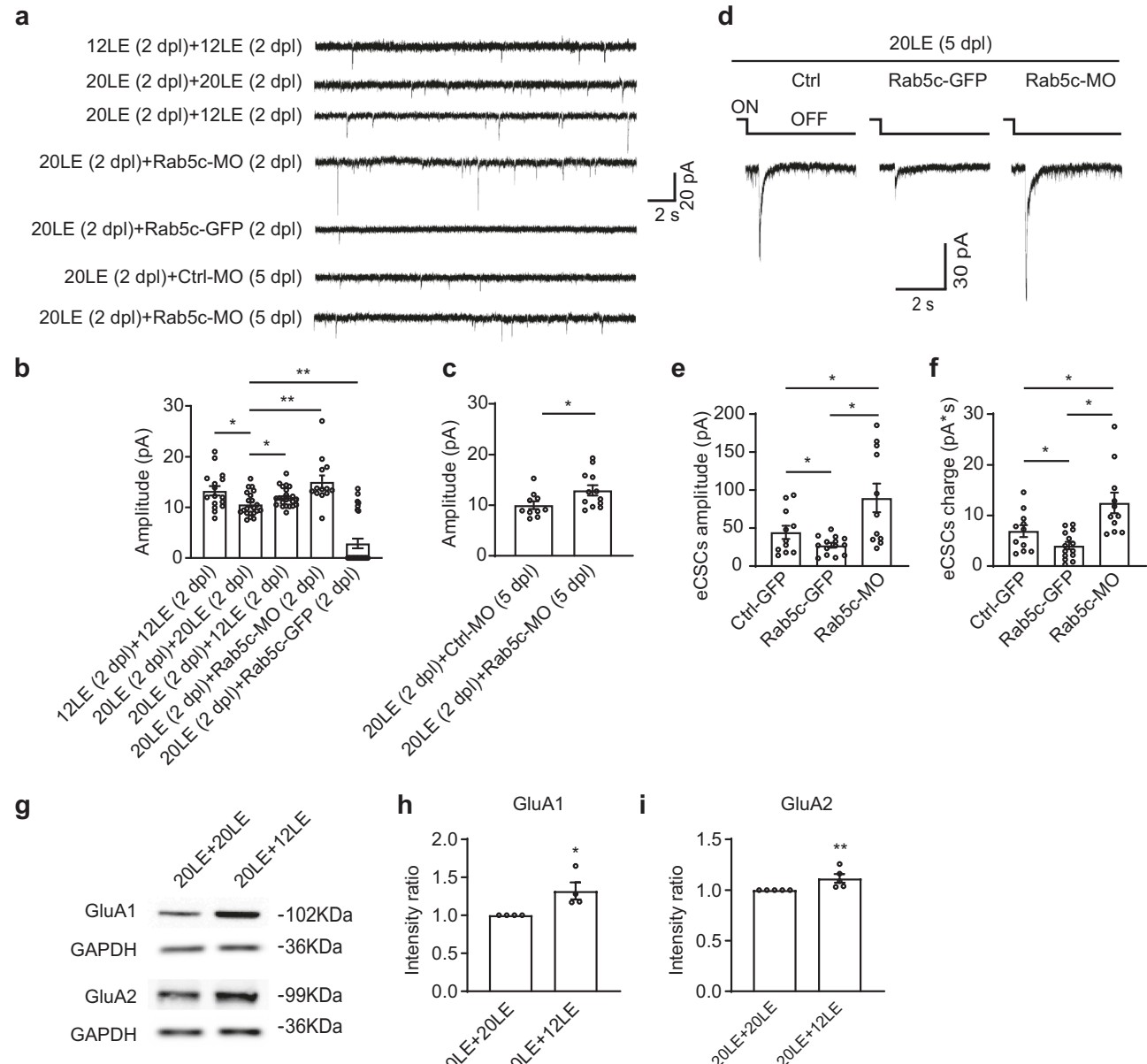

**Fig. 7 | Rab5c mediates AMPA receptor trafficking during synaptic scaling.**
**a** Representative mEPSCs traces from tadpoles under the following treatment groups: 12LE (2 dpl) + 12LE (2 dpl): raised in 12LE for 4 days; 20LE (2 dpl) + 20LE (2 dpl): raised in 20LE for 4 days; 20LE (2 dpl) + 12LE (2 dpl): raised in 20LE for 2 days followed by 12LE for 2 days; 20LE (2 dpl) + Rab5c-MO (2 dpl): raised in 20LE for 2 days followed by Rab5c-MO for 2 days; 20LE (2 dpl) + Rab5c-GFP (2 dpl): raised in 20LE for 2 days followed by Rab5c-GFP for 2 days; 20LE (2 dpl) + Ctrl-MO (5 dpl): raised in 20LE for 2 days followed by Ctrl-MO for 5 days; 20LE (2 dpl) + Rab5c-MO (5 dpl): raised in 20LE for 2 days followed by Rab5c-MO for 5 days.

**b**, **c** Summary of mEPSC amplitude changes. $n = 16, 21, 23, 13, 27$ (**b**, left to right); $n = 10, 12$ for Ctrl-MO and Rab5c-MO (**c**). **d** Representative traces for visually evoked eCSCs in Ctrl, Rab5c-GFP, and Rab5c-MO neurons at 5 dpl. Scale bar: 30 pA, 2 s. **e**, **f** Summary data showing that the amplitude (**e**) and integrated current (**f**) of eCSCs are restored in 20LE + Rab5c-MO compared to 20LE + Ctrl-GFP neurons. $n = 11, 14, 11$ for Ctrl, Rab5c-GFP and Rab5c-MO. **g–i** Western blots show increased GluA1 and GluA2 protein levels of synaptosomal fractions in 20LE + 12LE brains compared to 20LE + 20LE. GluA1 and GluA2: $n = 4, 5$ for 20LE + 20LE, and 20LE + 12LE. $*p < 0.05$, $**p < 0.01$.

## Discussion

This study highlights the developmental internalization of postsynaptic glutamate receptors, which leads to a progressive reduction in postsynaptic AMPAR currents within the optic tectum of *Xenopus laevis* tadpoles. Prolonged sensory activity, achieved by extending light exposure (20LE), further decreases mEPSC amplitudes without affecting mEPSC frequency. The ambient light-dependent synaptic downscaling is coordinated with dendritic structural changes, AMPAR trafficking, and histone acetylation. These findings reveal a coordinated mechanism in which activity-dependent histone acetylation enhances Rab5c-mediated AMPAR endosomal trafficking. HDAC-dependent acetylation reinforces GluA2

downregulation and upregulates Rab5c expression, thereby linking epigenetic remodeling with receptor trafficking to stabilize homeostatic synaptic scaling and promote circuit maturation (Supplementary Fig. 10).

Tectal neurons, which receive direct excitatory glutamatergic inputs from retinal ganglion cells, undergo synapse pruning and receptive field refinement during maturation[68,69]. Enhanced visual stimulation promotes rapid synaptic strengthening via AMPAR insertion into postsynaptic membranes, accompanied by accelerated dendritic arbor growth[1,10]. Our data indicate that prolonged light exposure induces a gradual reduction in glutamatergic synaptic currents, distinct from the rapid changes induced by patterned 1 Hz visual stimulation[70]. This represents a homeostatic

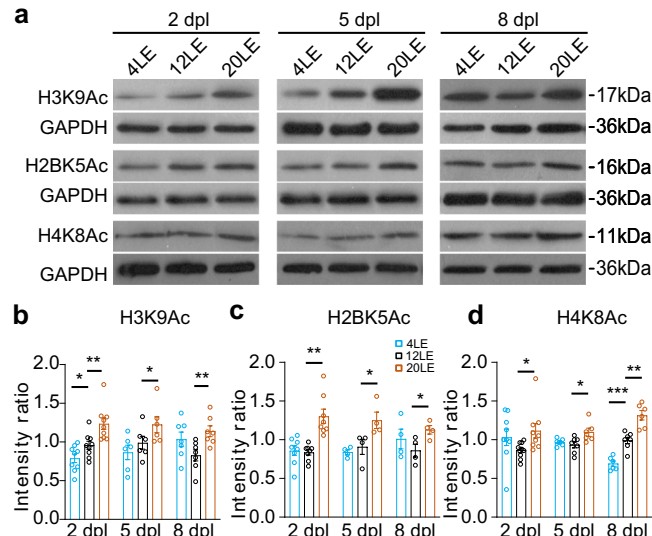

**Fig. 8 | Long-day exposure increases histone acetylation. a** Western blot analysis shows acetylation levels of H3K9Ac, H2BK5Ac and H4K8Ac in 4LE, 12LE, and 20LE groups at 2, 5, and 8 dpl. **b–d** Summary data reveal significantly elevated histone acetylation in 20LE brain compared to 12LE. 2 dpl/5 dpl/8 dpl: H3K9Ac: $n = 9/6/7$, 9/6/7, 9/6/7; H2BK5Ac: $n = 8/4/4$, 8/4/4, 8/4/4; H4K8Ac: $n = 9/6/6$, 9/6/6, 9/6/6 for 4LE, 12LE, and 20LE. *$p < 0.05$; **$p < 0.01$; ***$p < 0.001$ (two-way ANOVA with post hoc Fisher's LSD test).

adaptation over several days, in contrast to the effects of short-term visual stimulation. Previous studies have demonstrated that neural activity deprivation, such as dark rearing or eyelid suturing, results in larger mEPSCs in cultured neurons or cortical tissue[9,71–73]. Similarly, complete visual deprivation (0LE) robustly triggers homeostatic upscaling, as evidenced by a marked increase in mEPSC amplitude in 0LE-treated tadpoles (Supplementary Fig. 2). In contrast, partial sensory deprivation (4LE) produces no significant changes in mEPSC amplitude or frequency. These findings suggest that only substantial reductions in afferent input activate synaptic scaling mechanisms, whereas mild deprivation remains below the threshold for homeostatic adjustment[71]. At 2 dpl, both 4LE and 20LE increase Rab5c protein levels; however, only 20LE reduces mEPSC amplitude and GluA2 levels, suggesting that Rab5c upregulation alone is insufficient for synaptic scaling. Early epigenetic remodeling under 20LE, marked by increased histone acetylation and reduced *GluA1/2* mRNA, creates a permissive environment for Rab5c-mediated GluA2 internalization, highlighting a coordinated mechanism in which transcriptional/epigenetic priming precedes receptor trafficking. In contrast, 4LE elevates Rab5c without eliciting comparable transcriptional or epigenetic changes, limiting its impact on synaptic strength. HDAC3 bridges these processes by regulating both AMPAR transcription and Rab5c abundance, integrating trafficking and transcriptional mechanisms. Transcriptome analyses have further revealed distinct gene expression profiles related to synaptic upscaling or downscaling in response to varied light exposure conditions, underscoring activity-dependent changes in neuronal gene expression[1,64,74,75]. This raises *Xenopus* as a valuable model for investigating synaptic scaling mechanisms by simulating enhanced or diminished sensory experience.

As the brain matures, an excess of immature synapses undergoes pruning to refine functional circuits. Prior studies indicate that visual experience can refine the receptive field by eliminating weak synaptic inputs[76] or reducing retinotectal connections[13,68,77]. Consistent with this, we observed that mEPSC amplitudes progressively decrease as tadpoles develop from stage 42 to stage 49, paralleling findings in rodent pyramidal neurons[7] and *Xenopus* spinal neurons[78]. Prolonged light exposure reduces mEPSC amplitude without affecting input resistance, capacitance, or resting membrane potential, indicating that these synaptic changes arise from

postsynaptic modifications[79]. Additionally, paired-pulse ratios induced by optic nerve stimulation remain unchanged, further indicating that synaptic scaling is predominantly localized at postsynaptic site without affecting synaptic transmitter release probability during early tectal development[80].

Dynamic AMPAR trafficking is influenced by diverse factors, including LTD[81], LTP[82], and cellular stress[56]. Our findings indicate that Rab5c knockdown increases mEPSC amplitudes, while Rab5c overexpression decreases them and mimics the effects of prolonged light exposure, supporting the role of Rab5c-dependent endosomal trafficking in synaptic scaling[6,10,50]. Transcriptome analysis reveals that light exposure leads to a downregulation of *GluA1* and *GluA2* gene expression during the early phase of synaptic scaling, specifically at 2 dpl, with no significant changes observed at later time points. However, the protein levels of both subunits continue to decline in the subsequent stages. This pattern suggests that transcriptional regulation is critical for initiating the earliest phase of synaptic scaling, thereby setting the stage for sustained protein-level adjustments that drive long-term changes in synaptic strength[62–64]. Notably, Rab5c knockdown reverses the decrease in AMPAR currents, implicating Rab5c-dependent endosomal trafficking in the internalization of GluA2-containing receptors. Furthermore, these alterations are reversed when tadpoles are returned to standard environmental conditions, reinforcing the dynamic regulation of AMPAR trafficking in neural circuit maturation[76,83,84]. We focused on the 2 dpl time point because it captures the peak of Rab5c-dependent AMPAR internalization and the onset of histone acetylation, when trafficking and epigenetic changes occur in parallel before circuit-level stabilization at later developmental stages.

Interestingly, Rab5c also modulates dendritic architecture in a context-dependent manner. While 20LE increases Rab5c levels and enhances total dendritic branch length, branch tip number, and dendritic area, Rab5c-GFP overexpression alone under 12LE does not alter dendritic architecture. Moreover, Rab5c-GFP overexpression under 20LE partially attenuates the 20LE-induced increase in dendritic length, suggesting that Rab5c functions to restrain excessive structural growth and may require additional cofactors to coordinate neurite outgrowth[85–87]. Conversely, Rab5c knockdown increases dendritic complexity, likely representing a compensatory response to impaired endosomal trafficking or a release from a negative regulator of neurite growth[88]. These observations indicate that Rab5c contributes to maintaining dendritic homeostasis rather than directly promoting growth, and its functional role is shaped by the cellular environment engaged during prolonged sensory experience.

Visual experience modulates AMPAR subunit composition and trafficking, with specific reductions in GluA2-containing calcium-impermeable AMPARs (CI-AMPARs) following altered neural activity[89]. This contrasts with visual deprivation, which induces an increase in GluA1-containing calcium-permeable AMPARs (CP-AMPARs)[76]. The preferential reduction of CP-AMPARs aligns with synapse maturation during early brain development[8,90,91]. Our findings extend this developmental framework by showing that prolonged visual experience modifies this trajectory through coordinated transcriptional and trafficking mechanisms. Specifically, 20LE decreased both GluA1 and GluA2 protein levels at 5–8 dpl, with synaptosomal GluA2 showing a particularly strong reduction at 5 dpl. Transcriptome analysis revealed a transient downregulation of *GluA1* and *GluA2* mRNA at 2 dpl (Supplementary Fig. 6), coinciding with elevated histone acetylation. These findings support a sequential regulatory model in which an early HDAC-regulated epigenetic phase suppresses GluA1/2 transcription, followed by a Rab5-dependent trafficking mechanism that removes synaptic GluA2, thereby consolidating the long-term downscaling of excitatory synaptic strength. The subunit-specific reduction in synaptic GluA2 expression may result from distinct recycling mechanisms[29,92–94], synaptic removal[95] and lysosomal degradation of AMPARs during synaptic scaling[96,97]. Morphological changes, such as increased dendritic length and branching, coincide with synaptic downscaling and heightened neuronal excitability. These coordinated adjustments suggest a mechanism that stabilizes neural circuits during development, ensuring balanced connectivity and functional integration within the brain[98]. Unlike long-term depression

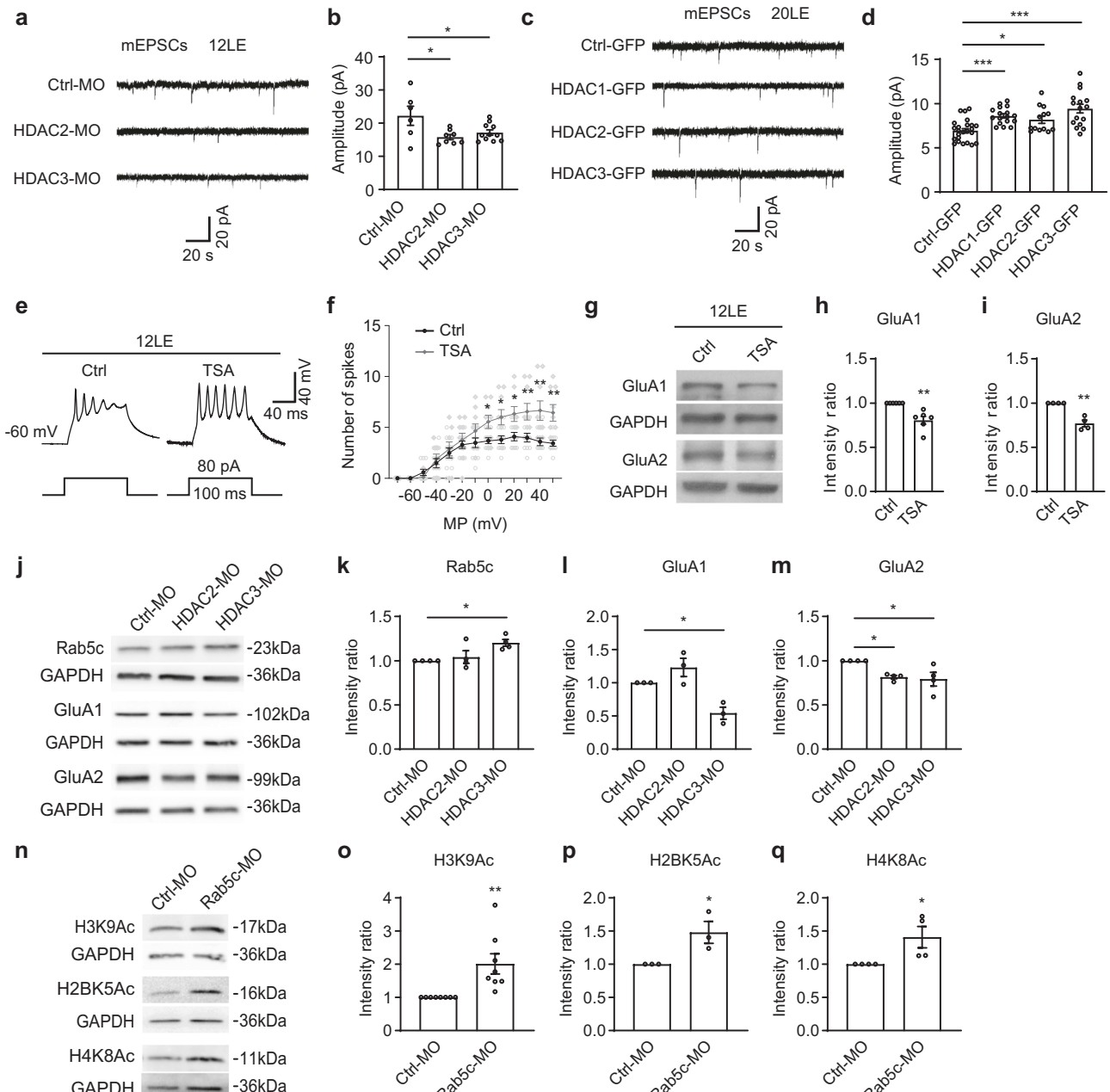

**Fig. 9 | Histone deacetylases regulate mEPSC amplitude, intrinsic excitability and AMPAR expression. a** Representative mEPSC traces from Ctrl-MO, HDAC2-MO, and HDAC3-MO neurons at 2 dpl. Scale bar: 20 pA, 20 s. **b** Quantification shows that HDAC2-MO and HDAC3-MO reduce mEPSC amplitudes. $n$ = 6, 8, 11 for Ctrl-MO, HDAC2-MO, and HDAC3-MO. **c** Representative mEPSC traces from Ctrl-GFP, HDAC1-GFP, HDAC2-GFP and HDAC3-GFP neurons in 20LE tadpoles at 2 dpl. Scale bar: 20 pA, 20 s. **d** Summary data showing that Ctrl-GFP, HDAC1-GFP, HDAC2-GFP, and HDAC3-GFP increase mEPSC amplitudes. $n$ = 23, 17, 14, 16 for Ctrl-GFP, HDAC1-GFP, HDAC2-GFP, and HDAC3-GFP. **e** Representative traces of current injection-induced spikes in control and TSA-treated neurons.

$n$ = 10, 9 for Ctrl and TSA. Scale bar: 40 mV, 40 ms. **f** Summary data showing that TSA treatment increases intrinsic excitability compared to control neurons. **g–i** Western blot analysis showing that TSA treatment downregulates GluA1 and GluA2 protein expression. GluA1: $n$ = 7, 7; GluA2: $n$ = 4, 4 for Ctrl and TSA. **j–m** Western blot analysis showing that HDAC2-MO decreases GluA2 expression, whereas HDAC3-MO downregulates GluA1 and GluA2 but increases Rab5c expression. Rab5c and GluA2: $n$ = 4 per group; GluA1: $n$ = 3 per group. **n–q** Western blot analysis showing that Rab5c-MO increases H3K9Ac, H2BK5Ac, and H4K8Ac levels. H3K9Ac: $n$ = 8 per group; H2BK5Ac: $n$ = 3 per group; H4K8Ac: $n$ = 4 per group. *$p < 0.05$; **$p < 0.01$; ***$p < 0.001$.

(LTD), which involves synaptic loss, prolonged light exposure-induced downscaling maintains dendritic structure while modulating receptor trafficking[99].

Histone acetylation plays a critical role in synaptic plasticity[2,51,52,57,100]. Prolonged light exposure significantly increases histone acetylation levels, with effects that persist across multiple cycles of light exposure. HDAC2/3 knockdown or HDAC inhibition (e.g., TSA treatment) reduces GluA2 protein levels and mEPSC amplitudes, alongside a marked increase in

neuronal excitability, linking HDAC-dependent histone modifications to changes in AMPAR expression and dynamics[56,101]. These findings demonstrate that epigenetic regulation, through changes in histone acetylation, influences glutamatergic transmission and dendritic architecture, facilitating synaptic maturation and circuit refinement[6,10,12,52,76].

In summary, this study elucidates the molecular pathways underlying synaptic scaling during neural development. By linking sensory activity with AMPAR trafficking and epigenetic mechanisms, we provide a

comprehensive framework for understanding how visual experience shapes neural circuits. Further exploration of inhibitory synapses[53,102] and excitation-inhibition balance[3,68] under varying sensory conditions may yield deeper insights into neural plasticity and circuit stabilization[14].

## Methods

### Animals and rearing

Albino *Xenopus laevis* embryos were obtained from in-house mating and reared in a 23 °C incubator in 0.1×Steinberg's solution (in mM: 58.0 NaCl, 0.67 KCl, 0.34 Ca(NO₃)₂, 0.83 MgSO₄, 10.0 HEPES, pH 7.2). For the control group, tadpoles were maintained in a custom-made box with a 12 h light/12 h dark (12LE) cycle. For the prolonged or shortened light exposure groups, tadpoles were exposed to either a 20 h light/4 h dark (20LE) or 4 h light/20 h (4LE) dark cycle. The average luminance of the LED array was 20 cd m⁻². All animal protocols were approved by the Animal Ethics Committee of Hangzhou Normal University (permit: 2023049). We have complied with all relevant ethical regulations for animal use. For experimental manipulations, MS-222 (0.02%, tricaine methanesulfonate, Sigma-Aldrich) was dissolved in 0.1×Steinberg's solution and administered to anesthetize animals. Animal developmental stages were determined according to Nieuwkoop and Faber (NF) stage numbers[103].

### DNA constructs, morpholinos, and transfection

The cDNA library was prepared from the optic tecta of *Xenopus laevis* tadpoles at NF stage 48. Primers were designed for Rab5c (accession XM_041575958) with the following sequences: forward 5'-CCCAAGCT-TATGGCCAGTAGAGGAAC-3' and reverse 5'-TGCTCTAGATTAGT TGCAGCACTGGCT-3'. The Rab5c-GFP plasmid (non-fusion) was constructed by subcloning Rab5c cDNA into the first CMV promoter site of a dual-CMV BICS2-EGFP vector, enabling independent expression of Rab5c and EGFP.

Control morpholino (Ctrl-MO, CCTCTTACCTCAGTTA-CAATTTATA) and *Xenopus* Rab5c morpholino (Rab5c-MO, GGGCTGA CACTGCACTAATGACAAG) were purchased from GeneTools (Philomath, OR). All morpholinos were tagged with lissamine for fluorescent visualization and screening. For the visualization of neuronal dendrites, morpholinos were co-transfected with BICS2-EGFP.

To transfect tectal cells, tadpoles were anesthetized and the plasmid solution (1.5 µg/µL) or morpholinos (10 µM) were pressure-injected into the midbrain ventricle. Whole-brain electroporation was then performed by positioning custom-made platinum electrodes on the skin overlying the optic tectum and delivering a series of current pulses with alternating polarity across the midbrain[3]. Transfected tadpoles were screened for appropriate transfection density 2–3 days post-electroporation using fluorescence microscopy before being used for further experiments.

### RNA isolation and sequencing

After exposure to 4LE, 12LE, or 20LE light cycles for 2 or 5 days, the brains of 40 tadpoles at stage 49 from each group were collected. The optic tecta were dissected and immediately frozen in liquid nitrogen. Total RNA was isolated using TRIzol reagent (Invitrogen) following the manufacturer's protocol. RNA sequencing and analysis were performed by Novogene (Beijing, China) using the Illumina HiSeq 2000 platform. Following quality assurance, the screened clean reads were aligned to the *Xenopus laevis* v10.1 genome (Xenbase database, http://www.xenbase.org). Differentially expressed genes (DEGs) were identified using DESeq (version 1.12.0), and gene expression was assessed using Cufflinks (version 2.1.1). FPKM (fragments per kilobase of exon per million mapped fragments) was used to quantify gene expression. The thresholds to identify DEGs were set at the false discovery rate (FDR) <0.05. Functional annotation analysis was conducted using DAVID.

### Real-time qRT-PCR analysis

Brains of ~30 tadpoles were isolated and total RNA was extracted using the Multisource Total RNA Miniprep Kit (Axygen). cDNA was reverse-transcribed from 0.3 µg of total RNA using the PrimeScript RT Reagent Kit (Takara Bio) and quantified with the UltraSYBR Mixture (Cwbio). RNA quality and purity were verified using a NanoDrop 2000 spectrophotometer (Thermo Fisher Scientific). Samples were analyzed using a CFX96 real-time PCR system (Bio-Rad). Gene expression levels were normalized to GAPDH and analyzed using the $2^{-\Delta\Delta CT}$ method. Primers for qRT-PCR are listed in Supplementary Data 1.

### Electrophysiology

All recordings were performed at room temperature (~23 °C). During the recordings, brains were perfused with extracellular saline containing (in mM: 115 NaCl, 2 KCl, 3 CaCl₂, 1.5 MgCl₂, 5 HEPES, 10 glucose, 0.01 glycine, and 0.05 tubocurarine, pH 7.2, osmolality 255 mOsm). Visually evoked synaptic currents were recorded from tectal neurons in whole-cell mode using a K⁺-based pipette solution (in mM: 110 K-gluconate, 8 KCl, 5 NaCl, 1.5 MgCl₂, 20 HEPES, 0.5 EGTA, 2 ATP, and 0.3 GTP). Action potentials were recorded in current-clamp mode. Recording pipettes were pulled from borosilicate glass and had resistances of 7–9 MΩ. The liquid junction potential was corrected during recordings. Whole-cell recordings were accepted for analysis from cells in which series resistance changed by <10%, and input resistance (0.7–2 GΩ) remained stable. Data were filtered at 2 kHz with a Multiclamp 700B amplifier (Molecular Devices), sampled at 10 kHz and analyzed using ClampFit 10 (Molecular Devices).

For whole-cell recordings, tadpoles were anesthetized, and tectal lobes were dissected along the dorsal midline. mEPSCs were recorded in voltage clamp at –60 mV in the presence of 1 µM extracellular TTX (Alomone Labs) to block action potentials. Excitatory and inhibitory components were distinguished based on their polarity and temporal profile at this holding potential[13]. Five-minute periods of mEPSC recordings were analyzed with the MiniAnalysis Program (Synaptosoft). For electrical stimulation-induced currents, a bipolar stimulating electrode (Frederick Haer) was placed in the optic chiasm to activate retinal ganglion cell (RGC) axons. For in vivo recordings, live tadpoles were immobilized on a Sylgard cushion in the recording chamber with one eye facing the stimulus LED. Full-field visual stimuli were delivered via a green LED fiber, with an inter-stimulus interval of 10 s for ten trials. To record visually evoked compound synaptic currents, we bulk-electroporated transfected tadpoles with Rab5c-MO or DNA constructs expressing BICS2-GFP, or BICS2-Rab5c-GFP. Animals were screened to select those with dense GFP expression or successful morpholino transfection for recordings.

### In vivo time-lapse imaging and analysis

For single-neuron transfection, plasmids or morpholinos were pressure-injected into the tectal tissue, followed by a single electroporation pulse delivered via laterally placed electrodes[10]. The tadpoles successfully transfected with single neurons were screened for subsequent in vivo imaging. To image single neurons, tadpoles were anesthetized and mounted under glass coverslips in a custom-made Sylgard chamber. Fluorescent neurons were imaged using a confocal microscope. Total dendritic branch length (TDBL) and branch tip number (TBTN) were analyzed using Imaris 7.4.2 with Filament Tracer (Bitplane, Zurich, Switzerland). The dendritic arbor was segmented and converted into a 3D surface model, and the total dendritic area was quantified by converting the structure into a triangular mesh and summing the surface areas of all triangles.

### Western blotting

Optic tecta were dissected, homogenized in radioimmunoprecipitation assay (RIPA) buffer supplemented with protease inhibitor cocktail (Sigma-Aldrich) at 4 °C, and centrifuged to collect protein lysates. Proteins were separated by SDS-PAGE, transferred to PVDF membranes, and blocked for 1 h in 4% nonfat milk in TBST (Tris-buffered saline with 0.1% Tween-20). Membranes were incubated overnight at 4 °C with primary antibodies diluted in TBST containing 0.4% nonfat milk. The following primary antibodies were used: anti-GAPDH (1:30,000, Millipore, ABS16), anti-Rab5c (1:2000, Abbkine, ABP55966), anti-GluA1 (1:2000, OriGene,

TA332438S), anti-GluA2 (1:1000, R&D Systems, pps050), anti-HDAC2 (1:1000, Abcam, ab137364), anti-HDAC3 (1:1000, Abcam, ab16047), anti-H3K9Ac (1:2000, Abcam, ab10812), anti-H2BK5Ac (1:2000, Abcam, ab408867), anti-H4K8Ac (1:1000, Abcam, ab45166), and anti-PSD95 (1:2000, Abcam, ab2723). After washing, membranes were incubated for 1 h at room temperature with HRP-conjugated secondary antibodies (1:2000, Cwbio). Protein bands were visualized using enhanced chemiluminescence (ECL, Pierce). All Western blot images were presented at Supplementary Fig. 11.

## Preparation of crude synaptosomal fraction

A crude synaptosomal fraction was prepared using the Syn-PER Synaptic Protein Extraction Reagent (Thermo Scientific) according to the manufacturer's instructions with minor adaptations. Briefly, freshly dissected optic tecta were homogenized in ice-cold Syn-PER Reagent (1 mL per 100 mg tissue) using a Dounce homogenizer with 15–20 gentle strokes. The homogenate was centrifuged at $1200 \times g$ for 10 min at 4 °C to remove nuclei and cellular debris. The resulting supernatant (S1) was centrifuged at $15,000 \times g$ for 20 min at 4 °C. The collected pellet (P2), containing the crude synaptosomal fraction, was resuspended in RIPA buffer for western blotting.

## Statistics and reproducibility

All data are presented as mean ± s.e.m. Data were considered significantly different when $p$ values are less than 0.05. Statistical analyses were conducted using GraphPad Prism 8. Where noted, a two-tailed Student's $t$-test or a nonparametric Wilcoxon signed-rank test was performed for within-cell comparison. For comparisons of multiple groups, either ANOVA with Newman–Keuls test or Kruskal–Wallis test with post hoc Mann–Whitney $U$- test was performed. The statistical test used for each experiment is specified in the Results section. Experiments and analyses were performed blind to the experimental conditions.

## Reporting summary

Further information on research design is available in the Nature Portfolio Reporting Summary linked to this article.

## Data availability

All data and materials generated in this study are available in the main text and supplementary materials. Source data for all graphs can be found in the Supplementary Data. All sequencing data have been deposited at the NCBI Sequence Read Archive under accession no. PRJNA1375510.

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

## Acknowledgements

This work was supported by the Interdisciplinary Research Project of Hangzhou Normal University (2024JCXK01) and the National Natural Science Foundation of China (NSFC 31871041).

## Author contributions

L.Z. and W.S. designed the experiment. L.Z., X.D., W.H., Y.L., Y.W., Q.L., X.W., L.H., and W.S. performed the experiments. L.Z., X.D. and W.S. analyzed the data. W.S. wrote the manuscript.

## Competing interests

The authors declare no competing interests.
