## [Transparent Peer Review file · Communications Biology]

Prolonged visual experience accelerates developmental synaptic downscaling via epigenetic regulation and Rab5c mediated AMPA receptor trafficking

Corresponding Author: Professor Wanhua Shen

Version 0:

Reviewer comments:

Reviewer #1

(Remarks to the Author)

Summary: Here, Zheng and colleagues report that exposing tadpoles to prolonged periods of light (20LE) leads to a decrease in synaptic strengths received by tectal neurons along with an increase in intrinsic excitability. The decrease in synaptic strengths is interpreted as synaptic scaling. It is found through RNAseq and western blot analysis that prolonged exposure to light decreased the expression of the AMPA receptor subunits GluA1 and GluA2. This is consistent with the observed decrease in mEPSC amplitudes recorded from 20LE tectal neurons. The 20LE group also showed an increase in Rab5c. It is unclear whether Rab5c was also identified through the RNAseq screen. Rab5c is known to regulate AMPA receptor cycling. In this study, it is determined that experimentally increasing Rab5c expression in tectal neurons dramatically reduces mEPSC amplitudes, and that inhibiting Rab5c increases mEPSC amplitudes. This suggests that prolonged light exposure decreases synaptic strength via an upregulation of Rab5c. Finally, it is determined that prolonged light exposure increases histone acetylation which is also found to decrease GluA1 and GluA2 levels. Overall, this study represents a huge amount of work. It seems it is several studies combined into one. While the finding that prolonged exposure to light induces downregulation of specific AMPA receptor subunits is notable, the study appears to lack logical flow/connectivity. For instance, what is the relationship between the histone acetylation mechanism and Rab5c? It is difficult to bring the different findings together. Furthermore, the data shown for several of the experiments appears to be incomplete. At the very least, the rationale for why certain experiments did not include certain time points (dpl) needs to be addressed.

Minor

Introduction

Line 48: "...such as exposure to 1 Hz for several hours..."

What is meant by this phrase? Exposure to 1 Hz of what? Please clarify this phrase.

Line 56: What is "reversible scaling"? Please define. Also – "scaling" needs to be defined somewhere.

Line 78-79: "...reduced miniature excitatory postsynaptic current (mEPSC) amplitudes in tectal neurons from stage 42 to 49 coincides to synaptic downscaling..."

The logic conveyed in this phrase seems circular, i.e. the decrease in mEPSC amplitudes defines the scaling down of synaptic currents. Thus, it is not clear how the decrease in mEPSC amplitudes can be interpreted to coincide with synaptic scaling since they are the same thing. This phrase needs to be rewritten.

Line 79: "... synaptic downscaling..."

Is this "synaptic downscaling" the same thing as "reversible scaling" that appears in line 56? If not, please define this term. If so, please choose one of the terms and use consistently throughout the manuscript.

Line 82-83: "the synaptic scaling was driven by Rab5c-mediated AMPAR removal and an increase in intrinsic excitability."

Is it actually determined that the increase in intrinsic excitability is driving the synaptic scaling and not that the decrease in mEPSC amplitudes is triggering the homeostatic increase in IE?

Results

Line 91-92 the section "Prolonged light exposure accelerates a developmental decline in glutamatergic synaptic transmission."

Please add specifics on recording mEPSCs. What voltage were the neurons clamped at for recording these events? How were the excitatory events isolated from inhibitory ones?

Line 112: "Prolonged light exposure decreases visually evoked synaptic responses."

Please include specifics on recording eCSCs. What voltage were the neurons held at for these recordings? How were the excitatory fraction of the visually evoked compound response isolated?

Lines 150 and 151: 0.44% and 0.54% should be 44% and 54%, respectively.

Discussion

Line 243-244: "Prolonged light exposure reduces synaptic transmission and glutamatergic receptors through HDAC2/3 mediation.

While the data in this section do support this subheading, how does this fit with the overall title of the manuscript which indicates that the scaling is via Rab5c? Is there a connection between the 20LE-induced increase in Rab5c and the altered pattern of histone acetylation?

Lines 271-272: "...these findings underscore the pivotal role of epigenetic mechanisms, particularly Rab5c-mediated postsynaptic AMPAR trafficking".

How is Rab5c an epigenetic mechanism?

Line 122-123: "To determine if the observed mEPSC amplitude decrease was due to altered presynaptic release probability"

A change in probability of presynaptic transmitter release would not be expected to alter mEPSC amplitude. Probability of release does, however, shape evoked responses. Perhaps this is what the authors mean to convey.

Line 279: What is meant by "spaced pattern of stimulation"?

Line 284-285: "However, under partial sensory deprivation conditions...the regulatory effects on synaptic scaling appear to play a stabilizing role within the network without introducing the pronounced synaptic changes observed under more extreme conditions."

It is unclear what is being conveyed here. 4LE did not induce synaptic scaling. It seems not possible to interpret beyond that. What exactly is playing a stabilizing role? This sentence needs to be clarified or deleted.

Line 303-305: "Our findings indicate that Rab5c knockdown reduces mEPSC amplitudes, while Rab5c overexpression increases them and counters the effect of prolong light exposure..."

This is the opposite of what the data show.

Line 319-320 "The preferential reduction in CI-AMPA receptors aligns with synapse maturation during early brain development".

This is incorrect. This is not what at least one of the referenced works shows (ref. 8). Reference #8 shows that during development, there is a gradual decline in calcium-permeable AMPARs, not calcium-impermeable AMPARs. Please review the findings of the referenced work and correct this section of the Discussion.

Figure 6

The iMaris reconstructions shown for the panel A Ctrl-GFP neuron and panel B Rab5c-GFP neuron dendrites do not appear to match that of their corresponding raw microscopy images.

Why was dendritic morphology studied only at the 2-3 dpl timepoint and not 5 and/or 8 dpl when the effect may have been more robust?

Figure 7panel B: The labeling of the x axis is confusing. Is the second group "12LE (2dpl)" really part of the 20LE (2dpl) group. If so, please explain more clearly in the legend. If not, please reorganize the graph to clearly show the different experimental groups.

Major:

Overall, the second half of the study appears to lack logical flow. From Figure 1 through Figure 4, the experimental progression is logical. From Figure 5 onwards it is difficult to understand the logical progression. For instance, in Figure 5, the authors follow up on the mRNAseq results by carrying out western blot for GluA1 and GluA2 to visualize protein expression levels (very nice) - but then here Rab5c is included. Then there is a sudden transient shift to studying the effect of Rab5c knock down and over-expression on dendrite morphology. Then an unexplained shift to the role of histone deacetylases on AMPAR expression. Is there a connection between histone deacetylation and Rab5c? It would be helpful if the authors included a schematic that combines all their findings into one model showing how 20LE is inducing a homeostatic decrease in mEPSC amplitudes.

It also appears that either not all experiments were carried out at all the developmental time points or that not all the data is being shown (details on this point below).

Line 179 -180: To identify the genes involved in the 20LE-induced homeostatic decrease in synaptic strengths, the authors carry out RNAseq, an unbiased approach. But then, the focus is on Rab5c, which does not appear to be identified in the RNAseq data. What is the motivation for looking at Rab5c protein levels? Was Rab5c pulled out of the RNAseq data? Rab5c RNA levels obtained from RNAseq should be shown. If not, there needs to be more justification of why, suddenly, Rab5C protein levels are being measured in this section.

Figure 2

Figure 1 shows mEPSC amplitudes and inter-event-intervals for 4LE, 12LE and 20LE at all 3 time points (2,5, and 8 dpl). Figure 2 shows the visually evoked data. In Figure 2, why are only the 2dpl data shown? It seems important to show the effect of 20LE and 4LE at the later time points (5 and 8 dpl) – since the tadpoles have been exposed to the altered stimuli for longer lengths of time. Plus, the decrease in mEPSC amplitude is most pronounced at 8dpl (2-star significance), so if the decrease in mEPSC amplitude is at least contributing to the decrease in visually evoked responses, the 8dpl time point would be predicted to be even more reduced. Thus, please add the 5 and 8 dpl data to Figure 2, or include in the text the reason for not including those data.

Figure 4

Please include the rationale for why the RNAseq data at 8dpl is not shown here.

If RNAseq was only carried out on 2 dpl and 5 dpl tissue, it is necessary to explain the rationale for focusing on these two time points.

Reviewer #2

(Remarks to the Author)

The manuscript by Zheng and colleagues addresses how prolonged visual experience impacts developmental synaptic scaling. Using *Xenopus laevis* tadpoles, exposed to prolonged ambient light (20h light/4 h dark – 20LE) or partial visual deprivation (4h light/20h dark – 4 LE), the authors measured synaptic scaling during early brain development in tectal neurons. They found that prolonged exposure to light accelerated the developmental decline in glutamatergic synaptic transmission and increased neuronal excitability. Transcriptome analyses revealed transcriptional changes primed by prolonged light exposure, namely decreased levels of GluA1 and GluA2 AMPA receptor subunit mRNA after 2 days of prolonged light exposure (but not after longer periods). GluA1 and GluA2 protein levels were also decreased in synaptosomes after extended light exposure, whereas the small GTPase Rab5c was increased. These observations prompted analyzing the role of Rab5c in the synaptic scaling response to prolonged light exposure. Indeed, the authors found that overexpression of Rab5c decreased the amplitude of mEPSCs and evoked currents, whereas knock-down of Rab5c increased current amplitude. In a second part of the manuscript, neuronal histone acetylation was found to be upregulated in animals subjected to prolonged light exposure. Knock-down of the histone deacetylases 2 or 3 decreased the amplitude of mEPSCs, whereas inhibition of histone deacetylation increased intrinsic excitability and decreased GluA1 and GluA2 levels.

The approach taken in this manuscript is interesting and relevant. Addressing synaptic scaling during development, and how it is impacted by ambient light exposure, is an important question. The work is well done and appropriately presented in the manuscript. Findings are relevant even if not surprising. The transcriptome dataset that was produced is useful for further studies but should at least be analyzed for enriched gene ontology categories. However, the more mechanistic aspects of the study (implication of Rab5c and histone acetylation) are not well integrated with the rest of the study, and need to be further substantiated.

For example, it is not clear why the authors decided to look at Rab5c levels in light manipulation conditions (and not at one of the many transcripts they found altered in the transcriptomic screening). At 2dpl, both 4LE and 20LE resulted in increased Rab5c levels, but only 20LE impacted synaptic scaling. The authors should discuss the significance of this observation.

Manipulating of Rab5c levels altered the neuronal architecture response to prolonged light exposure (Fig. 6), but these results are difficult to interpret, as 20LE increased Rab5c expression and increased total dendritic branch length, branch tip number and dendritic area, but Rab5c-GFP overexpression per se did not affect neuronal architecture. This suggests that the effects of 20LE on neuronal architecture are independent of Rab5c. On the other hand, knock-down of Rab5c increased dendritic complexity. These results show a complex effect of Rab5c, that does not seem to match what would be expected upon elevated Rab5c levels in 20LE exposed animals. The authors need to provide further explanation for these results, or to perform experiments to further investigate the role of Rab5c in the effects of prolonged light exposure on neuronal architecture.

In Figure 7, the authors present data that they claim to suggest that Rab5c mediates AMPAR trafficking during synaptic

scaling. This figure is confusing, because the first panels refer to the reversibility of the synaptic scaling effect. It would be more informative to start this figure by showing how Rab5c-MO or Rab5c-GFP impact synaptic scaling, which is never showed. In Fig. 7C-F, no comparison is made between 12LE and 20LE, to evaluate how manipulating Rab5c impacts synaptic scaling. As for panels A and B in this figure, the reversibility is interesting (and expected, given previous studies), but again the role of Rab5c is difficult to appreciate, as the effect of manipulating Rab5c on scaling is not clear.

The authors then explore how prolonged light exposure affects histone acetylation (again, it is unclear why, besides the fact that it has been implicated before), and they found increased histone acetylation upon long-day exposure. They knocked down the expression of histone deacetylases (to increase histone acetylation), or used a pharmacological strategy to inhibit histone deacetylases, and found decreases excitatory transmission, increased excitability and decreased levels of GluA1 and GluA2. However, these experiments were all performed in 12LE. Therefore, it is not possible to claim, as the authors do, that "Prolonged light exposure reduces synaptic transmission and glutamatergic receptors through HDAC2/3 mediation". For such claim, it would be necessary to test how manipulating histone acetylation impacts synaptic scaling triggered by prolonged light exposure.

Kind regards,
Ana Luisa Carvalho

Reviewer #3

(Remarks to the Author)

This paper by Zheng et al is a nice demonstration of the value of the *Xenopus* optic tectum as a system for testing how sensory experience engages homeostatic plasticity mechanisms, exploiting anatomical, transcriptomic and physiological measurements. Although it represents a tour de force experimentally, the presentation of the considerable data is somewhat lacking in a coherent narrative. This detracts from the strength of the manuscript as a whole but does not make its interesting conclusions less compelling. The key finding is that rearing animals in a circadian cycle with prolonged or shortened daylight periods results in profound changes in synaptic and intrinsic excitability of tectal neurons. A presumably homeostatic reduction in synaptic amplitude is shown in the 20LE condition. The effect is interesting, but given that the 4LE protocol did not really have an opposite effect, it is not entirely convincing that this is a homeostatic form of plasticity. The effects on intrinsic excitability were a nice addition, but mechanistically somewhat unsatisfying as they reported the phenomenology without really providing any insights into the underlying mechanism. Transcriptional analysis pointed to strong "homeostatic" regulation of AMPA receptor subunits and Rab5c, which presumably is important for receptor trafficking. Effects on dendritic branch size were also measured and a role for Rab5c suggested. Finally the importance of histone acetylation was examined. A clear increase in acetylation was observed in the 20LE group. This was curiously followed up by experiments in which knockdown of a subset of HDACs was performed, but for some reason the context of homeostatic plasticity was entirely abandoned for this experiment, leaving the result somewhat lacking context.

Overall it is an interesting and impressive series of studies, but the intended narrative about how epigenetic factors regulate homeostatic plasticity was not very clearly or thoroughly presented. These results are valuable and deserve publication, but it would be beneficial if a clear model could be synthesized at the end to bring all the findings together in a mechanistic narrative.

Specific comments:

1. In55 turnovers -> turnover
2. Line 73 discusses 3 HDACs but HDAC1 is not revisited in the rest of the paper. The justification for ignoring HDAC1 is not very strong.
3. Lines 94 – 96: The light exposure patterns began when the tadpoles were in stage 42. It might be useful to state why exactly they started at this specific developmental stage, and why they stopped at stage 49.
4. In 97 please consider providing Nieuwkoop & Faber staging in addition to days post light exposure.
5. Page 6 lin 134 time observed points -> observed time points
6. Lines 135-138: When evaluating the membrane capacitance, input resistance and resting membrane potential, the results are only demonstrated for the 2 dpl group. Why weren't the 5 dpl or 8 dpl tested as well? Is it possible that the passive membrane properties take a longer period of light exposure to be significantly different? The conclusion drawn here seems to be reaching a little bit if they did only test these membrane properties after only 2 days of light exposure.
7. Line 174-176: The authors in this section specify that the western blot was performed on tectal tissue in lines 223-224. It is unclear whether the western blot described by lines 174 – 176 were also performed using only tectal tissue. The sentence afterwards states that it was whole brain, but in that case it's unclear why the authors are making similar comparisons between whole brain homogenates and that from tectal tissue.
8. Line 228 on page 8 says "reinforcing the role of AMPAR trafficking" but the data are based on total expression of GluRs, not on cell surface expression, which would presumably require a biotinylation experiment.
9. HDAC inhibition results in Figure 9 are confusing as they don't reveal the blockade of a phenomenon but rather the shifting of baseline values in response to inhibition. These experiments would make much more sense in the context of testing whether the synaptic, excitability, morphology and most importantly transcriptomic changes associated with 20LE required HDAC activity or not. This feels like a lost opportunity and really breaks the logical flow of the narrative.
10. Line 352: It should be specified what the 0.02% MS-222 was dissolved in.
11. The methods on pg 14 provides an insufficient explanation of how cells were labeled and how MO was delivered. The methods simply say "injected into the tectum". I recommend moving the text from lines 365-370 down to this section.
12. Similarly for GFP and Rab5c expression. On page 12 it says a BICS2-EGFP vector with a dual CMV promoter was used.

It is unclear then whether the Rab5c was subcloned as a fusion protein with EGFP (as the nomenclature in the rest of the paper suggests) or for parallel expression with EGFP under an independent CMV promoter. I am guessing the latter but this needs to be clarified explicitly.

13. For morphometric analysis of tectal cells, how was “area” measured?

14. Figure 1E, Was a two-way ANOVA used here? Did it show a significant interaction? Why does Figure 1E show all conditions in one plot but frequency is divided into 3 plots in F, G and H?

15. Figure 2E/F, label shows EPSC1/EPSC2 but figure legend says EPSC2/EPSC1 which appears correct. The figure should be corrected.

16. Figure 4. It is difficult to determine how significant the reported changes area, and what genes are involved from this figure, although supplemental spreadsheets are included. Including volcano plots with the most significant genes labeled might be helpful.

17. Figure 5. What is the basis for claiming the antibodies are specific? (pg 7 In 176) Have they been validated with MO knockdown? The bands appear to be part of a larger smear of many proteins.

18. Synaptosome preps should be validated by providing synaptic (e.g.PSD95) and non-synaptic (eg. Histone H3?) staining so we can assess the quality of the fractionation.

19. Fig 9A: The data purport to show that the amplitudes of the HDAC2- and HDAC3-MO conditions are smaller than the CTRL-MO. Visually looking at figure 9A, it seems as though the representative trace for the HDAC2-MO is larger in amplitude than that of the control. This might benefit from choosing a different representative trace.

20. Supplementary Figure 1 was not uploaded as far as I could tell. But it is critical to being able to interpret quality of data in figs 5 and 6. This must be corrected.

Version 1:

Reviewer comments:

Reviewer #1

(Remarks to the Author)

The revised version of this manuscript is vastly improved. The authors have thoroughly addressed all my concerns.

Highlights include a new super-helpful schematic that illustrates the model and brings all the finding together, and additional experiments which reveal a Rab5C-dependent and -independent pathway. Great job. The authors should be commended.

I approve of this manuscript in its current form.

One minor note:

Line 87-88 : It is still stated that the observed increase in intrinsic excitability is driving the synaptic scaling (down). However, previous published work – for example Reference #18 of the manuscript – has shown that it is the decrease in synaptic input (due to polyamine block) that is driving the homeostatic increase in intrinsic excitability. This referenced work actually shows that the scaling down of synaptic input is what induces the scaling up of intrinsic excitability, not the other way around as is suggested here.

Reviewer #2

(Remarks to the Author)

Authors have appropriately addressed the issues that were previously raised, and the new experiments included are meaningful. The manuscript is now suitable for publication.

Subject: Submission of Revised Manuscript - [COMMSBIO-24-8400-T]

Reviewers' comments:

Reviewer #1 (Remarks to the Author):

Summary: Here, Zheng and colleagues report that exposing tadpoles to prolonged periods of light (20LE) leads to a decrease in synaptic strengths received by tectal neurons along with an increase in intrinsic excitability. The decrease in synaptic strengths is interpreted as synaptic scaling. It is found through RNAseq and western blot analysis that prolonged exposure to light decreased the expression of the AMPA receptor subunits GluA1 and GluA2. This is consistent with the observed decrease in mEPSC amplitudes recorded from 20LE tectal neurons. The 20LE group also showed an increase in Rab5c. It is unclear whether Rab5c was also identified through the RNAseq screen. Rab5c is known to regulate AMPA receptor cycling. In this study, it is determined that experimentally increasing Rab5c expression in tectal neurons dramatically reduces mEPSC amplitudes, and that inhibiting Rab5c increases mEPSC amplitudes. This suggests that prolong light exposure decreases synaptic strength via an upregulation of Rab5c. Finally, it is determined that prolonged light exposure increases histone acetylation which is also found to decrease GluA1 and GluA2 levels. Overall, this study represents a huge amount of work. It seems it is several studies combined into one. While the finding that prolonged exposure to light induces downregulation of specific AMPA receptor subunits is notable, the study appears to lack logical flow/connectivity. For instance, what is the relationship between the histone acetylation mechanism and Rab5c? It is difficult to bring the different findings together. Furthermore, the data shown for several of the experiments appears to be incomplete. At the very least, the rationale for why certain experiments did not include certain time points (dpl) needs to be addressed.

Response:

We thank the reviewers for their careful reading and insightful suggestions. Below we have clarified the mechanistic integration of Rab5c and histone acetylation, and explained the rationale for our time-point selection.

1. Mechanistic link between Rab5c and histone acetylation

We propose that prolonged light exposure (20LE) initiates a coordinated homeostatic cascade. Elevated postsynaptic excitability under 20LE triggers activity-dependent epigenetic remodeling, marked by increased histone acetylation at H3K9, H2BK5, and H4K8. HDAC3-mediated upregulation of synaptosomal Rab5c enhances AMPA receptor (GluA2) endocytosis, thereby reducing excitatory synaptic drive. This sustained reduction in synaptic transmission reinforces and stabilizes the transcriptional downregulation of glutamate receptor expression, ensuring long-term homeostatic balance in the visual circuitry.

Rab5c acts as a trafficking effector. Under 20LE conditions at 5 dpl, Rab5c protein levels are elevated in synaptosomal fractions, and Rab5c overexpression or knockdown

bidirectionally modulates mEPSC amplitude, confirming its direct involvement in AMPAR cycling and degradation. Consistent with this, Rab5c knockdown at 12LE enhances H3K9, H2BK5, and H4K8 acetylation (Fig.9n-q) and increases mEPSC amplitude (Fig.7a-c) and dendritic growth (Fig.6a-e). These findings support the idea that Rab5c-dependent trafficking changes occur first and help drive subsequent epigenetic remodeling.

In parallel, we observe robust histone acetylation upregulation under 20LE. Importantly, the temporal profile of AMPAR downregulation fits this coordinated model: GluA1/2 total protein levels are not reduced at 2 dpl, when Rab5c upregulation and early histone acetylation are already evident, but show marked decreases at 5 and 8 dpl (Fig. 5a, b). Moreover, HDAC2/HDAC3 knockdown or pharmacological HDAC inhibition decreases GluA2 protein levels (Fig. 9g-m) and further suppresses synaptic transmission. By contrast, HDAC3 knockdown selectively increases Rab5c protein expression (Fig. 9j-k), suggesting a novel regulatory axis where HDAC3 normally constrains Rab5c availability. This finding refines our model by indicating that HDACs influence synaptic scaling through two partially distinct mechanisms: (i) HDAC2/HDAC3 primarily regulates AMPAR subunit expression via chromatin remodeling, and (ii) HDAC3 additionally controls Rab5c protein levels, linking epigenetic regulation more directly to receptor trafficking.

Taken together, our findings support a feed-forward mechanism in which Rab5c-mediated AMPAR internalization triggers epigenetic modifications that progressively stabilize glutamate receptor downregulation, while HDAC3 provides a feedback mechanism capable of tuning Rab5c protein abundance. To illustrate this integrated model, we now provide a schematic in Fig. S10 and have expanded the Discussion accordingly.

2. Rationale for time-point selection

We focused much of our mechanistic analysis on the 2 days post-light exposure (2 dpl) window, corresponding to 48 hr of 20LE, for the following reasons: (1) it captures the peak of Rab5c-mediated AMPAR internalization and the onset of histone acetylation, before slower circuit-level compensations occur; (2) at 2 dpl, mEPSC amplitudes, Rab5c localization, and histone-acetylation changes are all synchronously altered, allowing direct correlation of trafficking and epigenetic events; (3) by 5-8 dpl, tectal neurons undergo complex structural remodeling (e.g., dendritic reorganization, network rewiring) that can obscure the primary, cell-intrinsic mechanisms we aimed to dissect. For these later stages, we restricted our analysis to AMPAR protein levels and mEPSCs, which clearly show that GluA1/2 downregulation emerges at 5 dpl and is maintained at 8 dpl (Fig. 5a, b), supporting the view that Rab5c and histone acetylation changes at 2 dpl initiate a longer-term receptor regulatory program. (4) Our “rescue” experiments (returning tadpoles from 20LE to 12LE) also started at 2 dpl, further justifying a uniform mechanistic time frame.

We have added the following sentence to the Discussion (Line: 352-355) to clarify how early and late time points align within our framework.

“We focused on the 2 dpl time point because it captures the peak of Rab5c-dependent AMPAR internalization and the onset of histone acetylation, when trafficking and epigenetic changes occur in parallel before circuit-level stabilization at later developmental stages.”

We hope these revisions clarify how Rab5c and histone acetylation act in concert to drive synaptic scaling under prolonged light exposure, and how our time-point selection supports both mechanistic precision and temporal resolution of AMPAR regulation.

Minor

Introduction

Line 48: “...such as exposure to 1 Hz for several hours...”

What is meant by this phrase? Exposure to 1 Hz of what? Please clarify this phrase.

Response: Thank you for pointing this out. We have revised the sentence to specify that this refers to visual stimulation at 1 Hz flickering light. The revised text now reads: “...such as prolonged exposure to 1 Hz flickering light for several hours...”

Line 56: What is “reversible scaling”? Please define. Also - “scaling” needs to be defined somewhere.

Response: We agree this term needed clarification. We now define synaptic scaling earlier in the introduction as: “Synaptic scaling refers to a form of homeostatic plasticity in which neurons adjust the strength of all their excitatory synapses up or down to stabilize overall activity.” “Reversible scaling” has been revised in the introduction to: “...particularly the mechanisms involved in reversible scaling, a bidirectional form of synaptic scaling that up- or down-regulates synaptic strength in response to altered and restored sensory input, is crucial for understanding the broader dynamics of activity-dependent synaptic regulation.”

Line 78-79: “...reduced miniature excitatory postsynaptic current (mEPSC) amplitudes in tectal neurons from stage 42 to 49 coincides to synaptic downscaling...”

The logic conveyed in this phrase seems circular, i.e. the decrease in mEPSC amplitudes defines the scaling down of synaptic currents. Thus, it is not clear how the decrease in mEPSC amplitudes can be interpreted to coincide with synaptic scaling since they are the same thing. This phrase needs to be rewritten.

Response: We agree that the logic was unclear. The sentence has been revised to: “...a progressive decrease in miniature excitatory postsynaptic current (mEPSC) amplitudes in tectal neurons from stage 42 to 49 exhibit a hallmark feature of synaptic downscaling during development.”

Line 79: “... synaptic downscaling...”

Is this “synaptic downscaling” the same thing as “reversible scaling” that appears in line 56? If not, please define this term. If so, please choose one of the terms and use consistently throughout the manuscript.

Response: Synaptic scaling includes synaptic upscaling and downscaling. Reversible scaling refers to a bidirectional form of synaptic scaling in which synaptic strengths can be up- or down-regulated in response to changes in sensory input, and subsequently reversed once normal input is restored. The two terms have been defined in the introduction in the revised manuscript. We now use the term “synaptic scaling” consistently throughout the manuscript and specify in each case whether it is up- or down-scaling to avoid confusion. The use of “reversible scaling” has been specified where experiments emerged as to restore the synaptic downscaling induced by 20 hr light exposure when tadpoles are returned to 12LE control condition.

Line 82-83: “the synaptic scaling was driven by Rab5c-mediated AMPAR removal and an increase in intrinsic excitability.” Is it actually determined that the increase in intrinsic excitability is driving the synaptic scaling and not that the decrease in mEPSC amplitudes is triggering the homeostatic increase in IE?

Response: Thank you for raising this. We clarify that the increase in intrinsic excitability drives synaptic scaling and triggers AMPAR trafficking. The sentence now reads: “Synaptic scaling was driven by an increase in intrinsic excitability and Rab5c-mediated AMPAR removal.”

Results

Line 91-92 the section “Prolonged light exposure accelerates a developmental decline in glutamatergic synaptic transmission.” Please add specifics on recording mEPSCs. What voltage were the neurons clamped at for recording these events? How were the excitatory events isolated from inhibitory ones?

Response: We thank the reviewer for pointing this out. We apologize for the earlier misstatement. The mEPSCs were recorded in voltage clamp at -60 mV in the presence of 1 μ M TTX without picrotoxin, and therefore GABAA receptor-mediated events were not pharmacologically blocked. However, at this holding potential, inhibitory postsynaptic currents are outward or near reversal and typically exhibit smaller amplitude, allowing us to selectively analyze inward excitatory events. We now clarify this in the Result section as follows: “mEPSCs were recorded in voltage clamp at -60 mV in the presence of 1 μ M extracellular TTX (Alomone Labs) to block action potentials. Inhibitory events are minimized at -60 mV holding potential and can be reliably distinguished from inward excitatory events based on polarity and kinetics.”

Line 112: “Prolonged light exposure decreases visually evoked synaptic responses.” Please include specifics on recording eCSCs. What voltage were the neurons held at for these recordings? How were the excitatory fraction of the visually evoked compound response isolated?

Response: We thank the reviewer for their comment and apologize for the confusion. Picrotoxin was not used in these experiments. Instead, tectal neurons were held at -60 mV, which allows for functional isolation of inhibitory versus excitatory synaptic components based on the reversal potentials of Cl⁻ and cations. As we and others have shown previously (Shen et al., 2011), excitatory responses appear as inward currents at -60 mV, while inhibitory GABAergic

responses are outward, allowing us to differentiate components without pharmacological blockade. We have revised the text to reflect this in the Results: “... , we recorded excitatory compound synaptic currents (eCSCs) from tectal neurons voltage-clamped at -60 mV...”, and the Methods “Excitatory and inhibitory components were distinguished based on their polarity and temporal profile at this holding potential, as described previously (Shen et al., 2011).”

Lines 150 and 151: 0.44% and 0.54% should be 44% and 54%, respectively.

Response: Corrected. Thank you for catching this.

Discussion

Line 243-244: “Prolonged light exposure reduces synaptic transmission and glutamatergic receptors through HDAC2/3 mediation.

While the data in this section do support this subheading, how does this fit with the overall title of the manuscript which indicates that the scaling is via Rab5c? Is there a connection between the 20LE-induced increase in Rab5c and the altered pattern of histone acetylation?

Response: We thank the reviewer for raising this important point. To clarify the relationship between Rab5c and histone acetylation, we performed additional experiments. We found that HDAC2 knockdown does not affect Rab5c protein levels but significantly decreases GluA2 expression, as confirmed by Western blots at 12LE (Fig. 9j-k). By contrast, HDAC3 knockdown increases Rab5c protein expression while also reducing GluA1/GluA2 levels (Fig. 9j-m). These findings suggest that histone acetylation regulates AMPAR expression through both Rab5c-dependent and Rab5c-independent pathways.

Our time-course analyses delineate the temporal progression of these mechanisms. At 2 dpl, total GluA1/2 protein levels remain stable, although qRT-PCR and RNA-seq reveal reduced *GluA1/2* mRNA expression. At this early stage, histone acetylation at H3K9, H2BK5, and H4K8 is already elevated, coinciding with heightened neuronal spiking under 20LE. By 5-8 dpl, total GluA1/2 protein levels decline, synaptosomal Rab5c expression increases, and synaptic GluA2 is selectively reduced. Collectively, these observations support a sequential model in which activity-dependent histone acetylation precedes transcriptional suppression and subsequent receptor endocytosis, driving sustained synaptic downscaling.

Thus, Rab5c-dependent trafficking and HDAC2/3-mediated chromatin remodeling do not function in isolation but converge to downregulate AMPARs under prolonged light exposure. HDAC3 appears to exert a dual role by both promoting Rab5c upregulation and modulating epigenetic regulation, thereby integrating the two pathways. This mechanism accounts for the robust reduction in mEPSC amplitude observed at later time points and reconciles the distinct contributions of Rab5c and histone acetylation in synaptic scaling. We have revised the Results and Discussion to present this framework more clearly and now include a schematic model (Fig. S10) summarizing how Rab5c and epigenetic regulation act in concert to drive synaptic downscaling. We also changed the title to “Prolonged Visual Experience Accelerates Developmental Synaptic Downscaling via Epigenetic Regulation and Rab5c-Mediated AMPA Receptor Trafficking” in our revised manuscript.

Lines 271-272: "...these findings underscore the pivotal role of epigenetic mechanisms, particularly Rab5c-mediated postsynaptic AMPAR trafficking". How is Rab5c an epigenetic mechanism?

Response: We thank the reviewer for pointing out this imprecision. You are correct. Rab5c is not an epigenetic mechanism but rather a postsynaptic trafficking effector. In our revised text, we clarified this distinction and emphasized the temporal sequence between epigenetic regulation and trafficking events. Specifically, histone acetylation via HDAC2/3 acts first at the transcriptional level to reduce GluA1/2 expression, establishing the epigenetic basis for synaptic downregulation, whereas Rab5c mediates the subsequent rapid internalization of AMPARs to consolidate and maintain reduced synaptic strength.

Our updated experiments further refine this model. HDAC2 knockdown did not alter Rab5c expression but decreased GluA2 levels, indicating that HDAC2 acts through transcriptional regulation independently of Rab5c. In contrast, HDAC3 knockdown both increased Rab5c protein expression and reduced GluA1/GluA2 levels, suggesting that HDAC3 links epigenetic regulation and trafficking by enhancing Rab5c-dependent AMPAR endocytosis while simultaneously influencing gene transcription.

Thus, the updated sentence now reads:

“ These findings reveal a coordinated mechanism in which activity-dependent histone acetylation enhances Rab5c-mediated AMPAR trafficking. HDAC-dependent acetylation reinforces GluA2 downregulation and upregulates Rab5c expression, thereby linking epigenetic remodeling with receptor trafficking to stabilize homeostatic synaptic scaling and promote circuit maturation (Fig. S10).”

Line 122-123: “To determine if the observed mEPSC amplitude decrease was due to altered presynaptic release probability” A change in probability of presynaptic transmitter release would not be expected to alter mEPSC amplitude. Probability of release does, however, shape evoked responses. Perhaps this is what the authors mean to convey.

Response: We appreciate this clarification. The sentence has been revised to reflect that we assessed evoked responses, not mEPSC amplitude, to examine presynaptic release changes: “To determine if the developmental changes in excitatory currents was due to altered presynaptic release probability...”

Line 279: What is meant by “spaced pattern of stimulation”?

Response: Spaced pattern of stimulation refers to the 1 Hz visual stimulation, which was used to induce synaptic plasticity in *Xenopus* tectum. We now clarify this as: “...distinct from the rapid changes induced by patterned 1 Hz visual stimulation”

Line 284-285: “However, under partial sensory deprivation conditions...the regulatory effects on synaptic scaling appear to play a stabilizing role within the network without introducing the

pronounced synaptic changes observed under more extreme conditions.” It is unclear what is being conveyed here. 4LE did not induce synaptic scaling. It seems not possible to interpret beyond that. What exactly is playing a stabilizing role? This sentence needs to be clarified or deleted.

Response: Thank you for raising this point. We have performed new experiments by recording tectal neurons under 0LE conditions and refined our analysis to directly address the distinction between partial (4LE) and complete (0LE) sensory deprivation and its implications for synaptic scaling:

4LE (partial deprivation): No change in mEPSC amplitude or frequency was observed, indicating that network activity remains near the homeostatic threshold. Under these mild deprivation conditions, synaptic scaling mechanisms are not activated, thus preserving baseline synaptic function without inducing up- or down-scaling adaptations.

0LE (complete deprivation): Our newly obtained data (Supplementary Fig.2) demonstrate a significant increase in mEPSC amplitude, confirming robust homeostatic upscaling in response to more drastic reduction of sensory input.

We added the following sentence to the revised manuscript (line: 310-315): “Similarly, complete visual deprivation (0LE) robustly triggers homeostatic upscaling, as evidenced by a marked increase in mEPSC amplitude in 0LE-treated tadpoles (Supplementary Fig. 2). In contrast, partial sensory deprivation (4LE) produces no significant changes in mEPSC amplitude or frequency. These findings suggest that only substantial reductions in afferent input activate synaptic scaling mechanisms, whereas mild deprivation remains below the threshold for homeostatic adjustment.”

This updated text clarifies that the stabilizing role refers to the maintenance of existing synaptic balance under mild deprivation, whereas complete deprivation invokes active synaptic scaling. This interpretation is consistent with established literature showing that global reductions in activity (e.g., via sensory deprivation or TTX blockade) trigger synaptic scaling mechanisms to restore neural firing rates.

Line 303-305: “Our findings indicate that Rab5c knockdown reduces mEPSC amplitudes, while Rab5c overexpression increases them and counters the effect of prolonged light exposure...” This is the opposite of what the data show.

Response: Thank you for pointing this out. This was indeed reversed. It is now corrected to: “Our findings indicate that Rab5c knockdown increases mEPSC amplitudes, while Rab5c overexpression decreases them and mimics the effects of prolonged light exposure...”

Line 319-320 “The preferential reduction in CI-AMPA receptors aligns with synapse maturation during early brain development”. This is incorrect. This is not what at least one of the referenced works shows (ref. 8). Reference #8 shows that during development, there is a gradual

decline in calcium-permeable AMPARs, not calcium-impermeable AMPARs. Please review the findings of the referenced work and correct this section of the Discussion.

Response: We thank the reviewer for pointing this out. Reference 8 shows that immature neurons express higher levels of calcium-permeable AMPARs (CP-AMPARs, lacking GluA2), which normally decline during development. In our study, prolonged visual experience (20LE) decreased both GluA1 and GluA2 protein expression at 5 dpl and 8 dpl, but not at 2 dpl (Fig. 5a, b). Notably, synaptosomal GluA2, but not GluA1, was specifically reduced at 5 dpl under 20LE. Consistently, both *GluA1* and *GluA2* mRNA levels were lower in 20LE compared to 12LE tadpoles at 2 dpl (Supplementary Fig.6a, b) but not at 5 dpl (Supplementary Fig.7a, b), indicating temporal control of transcriptional regulation. Mechanistically, these results suggest a dual process: (1) 20LE decreases synaptic GluA2 through Rab5c-dependent trafficking, and (2) HDAC2/3 knockdown reduces GluA1/GluA2 expression (Fig. 9j-m), demonstrating that epigenetic regulation via histone acetylation further contributes to AMPAR subunit control.

Thus, rather than contradicting the developmental decline of CP-AMPARs described in Reference 8, our findings indicate that 20LE modifies this trajectory through a coordinated mechanism: Rab5c-mediated trafficking that decreases synaptic GluA2, while HDAC2/3-dependent epigenetic regulation that sustains reduced GluA2 expression.

We have revised the Discussion accordingly. The sentence below has been added to the revised manuscript (line: 371-380):

“ Our findings extend this developmental framework by showing that prolonged visual experience modifies this trajectory through coordinated transcriptional and trafficking mechanisms. Specifically, 20LE decreased both GluA1 and GluA2 protein levels at 5-8 dpl, with synaptosomal GluA2 showing a particularly strong reduction at 5 dpl. Transcriptome analysis revealed a transient downregulation of *GluA1* and *GluA2* mRNA at 2 dpl (Supplementary Fig.6), coinciding with elevated histone acetylation. These findings support a sequential regulatory model in which an early HDAC-regulated epigenetic phase suppresses GluA1/2 transcription, followed by a Rab5-dependent trafficking mechanism that removes synaptic GluA2, thereby consolidating the long-term downscaling of excitatory synaptic strength.”

Figure 6 The iMaris reconstructions shown for the panel A Ctrl-GFP neuron and panel B Rab5c-GFP neuron dendrites do not appear to match that of their corresponding raw microscopy images. Why was dendritic morphology studied only at the 2-3 dpl timepoint and not 5 and/or 8 dpl when the effect may have been more robust?

Response: We apologize for the confusion surrounding our reconstructions. Rab5c-GFP is expressed from a bicistronic CMV-driven construct, which labels both axons and dendrites. The raw microscopy images show both axons and dendrites, while in our tracings, axonal branches were intentionally omitted for clarity, but to prevent any misunderstanding we have now added more arrowheads in the raw micrographs to explicitly mark the axon origin.

With respect to the 2-3 dpl time point: we chose this early window because it coincides exactly with the functional hallmarks of Rab5c-dependent synaptic scaling—namely, the rapid reduction in mEPSC amplitude and the matched decrease in visually evoked responses. Our rescue experiments of recording were used 20LE-treated tadpoles for 2 days then returned to 12LE for 2 days. The fundamental logic was based on the 2 dpl timepoint. Furthermore, by 5 and 8 dpl, tectal neurons engage broader homeostatic and circuit-level adaptations (including compensatory dendritic stabilization and network rewiring) that can obscure the primary, endocytosis-driven remodeling we aim to dissect. Focusing on 2-3 dpl therefore allowed us to correlate, in the same cells, the earliest postsynaptic structural changes with our electrophysiological readouts under uniform imaging and tracing conditions. We have added this explanation to the revised Results sections to clarify why later time points were not included in the dendritic-morphology analysis.

“Neurons were imaged 2-3 days post-transfection, a window chosen because it coincides with the earliest functional signatures of Rab5c-dependent synaptic scaling, and aligns with our 2 dpl rescue experiments.”.

Figure 7 panel B: The labeling of the x axis is confusing. Is the second group “12LE (2dpl)” really part of the 20LE (2dpl) group. If so, please explain more clearly in the legend. If not, please reorganize the graph to clearly show the different experimental groups.

Response: Thank you for raising this issue. You are correct: the “12LE (2 dpl)” group indeed follows a 48-hour exposure to 20LE, then a return to 12LE for 2 days. To clarify: The tadpoles were initially raised in 20LE for 2 days, and then either returned to 12LE or transfected with Rab5c-MO or Rab5c-GFP. To ensure this sequence is clearly communicated, we have updated both the figure and its legend.

Revised groups for Fig. 7a-c:

12LE (2 dpl) + 12LE (2 dpl): Tadpoles raised in 12LE for 4 days;

20LE (2 dpl) + 20LE (2 dpl): Raised in 20LE for 4 days;

20LE (2 dpl) + 12LE (2 dpl): Raised in 20LE for 2 days followed by 12LE for 2 days;

20LE (2 dpl) + Rab5c-MO (2 dpl): Raised in 20LE for 2 days followed by Rab5c knockdown for 2 days;

20LE (2 dpl) + Rab5c-GFP (2 dpl): Raised in 20LE for 2 days followed by Rab5c overexpression with Rab5c-GFP for 2 days;

20LE (2 dpl) + Ctrl-MO (5 dpl): Raised in 20LE for 2 days followed by Ctrl-MO for 5 days;

20LE (2 dpl) + Rab5c-MO (5 dpl): Raised in 20LE for 2 days followed by Rab5c-MO for 5 days.

Major:

Overall, the second half of the study appears to lack logical flow. From Figure 1 through Figure 4, the experimental progression is logical. From Figure 5 onwards it is difficult to understand the logical progression. For instance, in Figure 5, the authors follow up on the mRNAseq results by carrying out western blot for GluA1 and GluA2 to visualize protein expression levels (very

nice) - but then here Rab5c is included. Then there is a sudden transient shift to studying the effect of Rab5c knock down and over-expression on dendrite morphology. Then an unexplained shift to the role of histone deacetylases on AMPAR expression. Is there a connection between histone deacetylation and Rab5c? It would be helpful if the authors included a schematic that combines all their findings into one model showing how 20LE is inducing a homeostatic decrease in mEPSC amplitudes. It also appears that either not all experiments were carried out at all the developmental time points or that not all the data is being shown (details on this point below).

Response: We thank the reviewer for this insightful comment. In response, we have substantially revised the Results and Discussion sections to improve the logical flow of the manuscript, particularly in the second half.

First, we clarified the mechanistic relationship between Rab5c and histone deacetylation pathways. Rab5c is a trafficking effector that regulates AMPAR endocytosis, whereas histone acetylation controls transcriptional programs that influence receptor subunit availability and Rab5c expression. Our new data refine this distinction. Specifically, HDAC2 knockdown did not affect Rab5c protein levels but significantly reduced GluA2 expression, indicating that HDAC2 functions primarily at the transcriptional level. In contrast, HDAC3 knockdown not only reduced GluA1/GluA2 proteins but also increased Rab5c expression, suggesting that HDAC3 integrates both pathways by coupling epigenetic remodeling with receptor trafficking. Together, these findings support a coordinated model: HDAC2/3-dependent histone acetylation regulates AMPAR subunit expression, while Rab5c governs receptor internalization and degradation at synaptic sites, with HDAC3 uniquely linking these two mechanisms.

Second, we integrated the temporal sequence of events under prolonged light exposure (20LE). At early stages (2 dpl), total GluA1/2 protein levels are unchanged, although GluA1/2 mRNA levels are significantly reduced, coinciding with increased acetylation of H3K9, H2BK5, and H4K8 and heightened tectal spiking activity. By 5-8 dpl, total GluA1/2 protein levels decline, with a selective reduction of GluA2 in synaptosomal fractions and elevated Rab5c expression, consistent with enhanced endocytosis. Functionally, mEPSC amplitude is decreased under 20LE, whereas Rab5c knockdown increases mEPSC amplitude. These results position Rab5c as a critical mediator of activity-dependent downscaling of excitatory drive. Our dendritic morphology data support this timeline: Rab5c knockdown increased dendritic length at 12LE, while Rab5c overexpression prevented 20LE-induced increase in dendritic length, reinforcing the link between Rab5c-dependent AMPAR trafficking and structural refinement.

Finally, we revised the Discussion to emphasize how these processes converge. Prolonged visual stimulation first triggers Rab5c-mediated removal of GluA2 from synaptic sites, driving the initial downscaling of excitatory synaptic strength. This reduction is subsequently reinforced by epigenetic remodeling via HDAC2/3-regulated histone acetylation, which decreases GluA1/2 transcription to maintain long-term synaptic downscaling. Notably, HDAC3 bridges these mechanisms by upregulating Rab5c, thereby enhancing receptor endocytosis while simultaneously contributing to transcriptional repression. Together, histone acetylation

serves as the epigenetic trigger, Rab5c executes AMPAR trafficking, and HDAC3 integrates these parallel processes to coordinate sustained homeostatic scaling.

To help guide the reader, we have introduced Rab5c earlier in the manuscript and clarified the rationale for its inclusion, citing relevant literature on its role in AMPAR trafficking. We also now provide a schematic (new Fig. S10) that integrates all key findings into a single working model, illustrating how 20LE drives HDAC3-regulated GluA1/GluA2 downregulation and Rab5c-mediated AMPAR internalization across developmental time points.

We have added these clarifications throughout the revised manuscript, including in the Introduction, Methods, Results, and Figure legends. We believe these revisions address the reviewer's concerns and present a clearer, temporally resolved narrative linking Rab5c, histone acetylation, AMPAR regulation, and circuit homeostasis.

Line 179 -180: To identify the genes involved in the 20LE-induced homeostatic decrease in synaptic strengths, the authors carry out RNAseq, an unbiased approach. But then, the focus is on Rab5c, which does not appear to be identified in the RNAseq data. What is the motivation for looking at Rab5c protein levels? Was Rab5c pulled out of the RNAseq data? Rab5c RNA levels obtained from RNAseq should be shown. If not, there needs to be more justification of why, suddenly, Rab5C protein levels are being measured in this section.

Response: We thank the reviewer for raising this important point. Although Rab5c did not appear among the top-ranked differentially expressed transcripts in our RNA-seq dataset, unbiased gene ontology (GO) enrichment analysis revealed strong representation of categories related to membrane trafficking and synaptic organization (e.g., AMPA glutamate receptor activity, GO:0004971; glutamatergic synaptic transmission, GO:0035249; ionotropic glutamate receptor signaling, GO:0035235). Because Rab5 family proteins are well known regulators of AMPAR endocytosis and glutamatergic synapse remodeling (Brown et al., 2005; Zhong et al., 2008), we reasoned that Rab5c could serve as a key downstream effector even in the absence of robust mRNA changes.

To directly address this, we examined Rab5 isoforms by qRT-PCR and confirmed that Rab5a, Rab5b, and Rab5c transcript levels remain unchanged at both 2 dpl (Supplementary Fig.6) and 5 dpl (Supplementary Fig.7), consistent with our RNA-seq results (now included in Table S1-4). In contrast, *GluA1* and *GluA2* mRNAs were significantly reduced at 2 dpl (Fig. S6a, b) but returned toward baseline by 5 dpl (Fig. S7a, b). Importantly, despite the lack of Rab5c transcript changes, synaptosomal Western blots revealed robust upregulation of Rab5c protein under 20LE at 5 and 8 dpl, coinciding with selective loss of synaptosomal GluA2. At 2 dpl, total GluA1/2 protein levels remained unchanged, while histone acetylation marks (H3K9ac, H2BK5ac, H4K8ac) were already elevated and *GluA1/2* mRNA reduced, suggesting that transcriptional regulation is an earlier event, followed later by Rab5c-mediated trafficking.

Functional assays further validate this sequence: Rab5c knockdown increased mEPSC amplitude and dendritic length under 12LE, while Rab5c overexpression under 20LE decreased dendritic length. In contrast, 20LE itself decreased mEPSC amplitude, demonstrating that

Rab5c activity is necessary for the synaptic downscaling phenotype. In parallel, HDAC2 knockdown reduced GluA2 protein levels without altering Rab5c expression, while HDAC3 knockdown reduced GluA1/GluA2 and increased Rab5c protein levels, suggesting that HDAC3 provides a point of cross-talk between epigenetic remodeling and trafficking regulation.

Taken together, these findings support a temporally structured, coordinated model: (1) early epigenetic remodeling (histone acetylation via HDAC2/3) suppresses AMPAR transcription at 2 dpl, and (2) Rab5c protein upregulation (enhanced by HDAC3 knockdown) drives synaptic GluA2 removal at 5-8 dpl, consolidating the sustained decrease in excitatory drive. We have now clarified this rationale in the Results, and added schematic diagrams (Supplementary Fig.10) to illustrate how Rab5c-dependent trafficking and HDAC2/3-mediated regulation cooperate in 20LE-induced synaptic scaling.

References

Brown, T.C., Tran, I.C., Backos, D.S., Esteban, J.A., 2005. NMDA Receptor-Dependent Activation of the Small GTPase Rab5 Drives the Removal of Synaptic AMPA Receptors during Hippocampal LTD. *Neuron* 45, 81-94.

Zhong, P., Liu, W.H., Gu, Z.L., Yan, Z., 2008. Serotonin facilitates long-term depression induction in prefrontal cortex via p38 MAPK/Rab5-mediated enhancement of AMPA receptor internalization. *Journal of Physiology* 586, 4465-4479.

Figure 2

Figure 1 shows mEPSC amplitudes and inter-event-intervals for 4LE, 12LE and 20LE at all 3 time points (2,5, and 8 dpl). Figure 2 shows the visually evoked data. In Figure 2, why are only the 2dpl data shown? It seems important to show the effect of 20LE and 4LE at the later time points (5 and 8 dpl) - since the tadpoles have been exposed to the altered stimuli for longer lengths of time. Plus, the decrease in mEPSC amplitude is most pronounced at 8dpl (2-star significance), so if the decrease in mEPSC amplitude is at least contributing to the decrease in visually evoked responses, the 8dpl time point would be predicted to be even more reduced. Thus, please add the 5 and 8 dpl data to Figure 2, or include in the text the reason for not including those data.

Response: We thank the reviewer for this thoughtful observation. Our decision to present only the 2 dpl data in Figure 2 was guided by our aim to isolate the early, cell-intrinsic mechanisms of synaptic scaling before the onset of broader circuit-level plasticity. Specifically, the 2 dpl window aligns with the most robust and direct alterations in postsynaptic physiology including the reduction in mEPSC amplitude and the downregulation of AMPARs, which we also validated via paired-pulse ratio analysis and transcriptomic data.

At 2 dpl, visually evoked responses are clearly reduced in the 20LE group, coinciding temporally with the earliest and least confounded expression of Rab5c-dependent AMPAR trafficking. This time point also matches the rescue conditions used throughout our study

(tadpoles exposed to 20LE for 2 days and returned to 12LE), reinforcing its relevance across experiments.

While we agree that later time points are of biological interest, by 5 and 8 dpl, the optic tectum exhibits widespread structural and homeostatic changes including dendritic remodeling, spine turnover, and network reorganization that complicate interpretation of evoked responses in relation to early AMPAR trafficking alone. Indeed, our transcriptomic analysis at 5 dpl showed no significant difference in *GluA1* or *GluA2* expression across 4LE, 12LE, and 20LE groups (Fig. 4e and Supplementary Fig.7), suggesting that the initial transcriptional drivers of AMPAR downregulation have subsided. For these reasons, we chose to focus our mechanistic analysis on 2 dpl, when the causal link between prolonged light exposure and postsynaptic scaling is most direct and interpretable.

We have added this rationale to the revised Results (line: 215-216; Section: Rab5c regulates neuronal structure development) and Discussion (line: 352-355) to clarify our decision and to better contextualize the focus on early post-exposure dynamics.

Figure 4

Please include the rationale for why the RNAseq data at 8dpl is not shown here. If RNAseq was only carried out on 2 dpl and 5 dpl tissue, it is necessary to explain the rationale for focusing on these two time points.

Response: We have now clarified that RNA seq was performed at 2 and 5 dpl because transcriptomic dynamics and the gene-expression changes underlying early synaptic modifications peak within this window. By sampling at 2 dpl, we capture the onset of light induced signaling pathways that initiate synaptic scaling; by 5 dpl, we observe the initial progression of those responses. Indeed, glutamatergic receptor mRNA levels are significantly altered at 2 dpl but return toward baseline by 5 dpl, implying that subsequent synaptic adjustments rely more heavily on receptor protein trafficking and local translation than on further transcriptional regulation. Restricting sequencing to these two early time points allowed for deeper coverage, reduced biological variability, and minimized animal usage and experimental complexity, while keeping our focus on the early synaptic scaling mechanisms rather than on later developmental outcomes, which may include neurocircuit remodeling or behavioral adaptation. Accordingly, we did not perform RNA seq at 8 dpl, since transcriptomic profiles by that stage predominantly reflect stabilized, downstream consequences rather than the primary signaling events of interest. To corroborate the importance of this early window, we performed rescue experiments in which animals were returned from 20LE back to a normal 12LE cycle immediately after 2 dpl, further validating the critical role of early transcriptional responses in driving synaptic scaling. This rationale is now explained in the Methods and Results sections as shown above. The following texts have been added to the results section (line: 152-153): “To investigate how light exposure influences the onset of light induced-gene expression that may initiate synaptic scaling, we conducted RNA sequencing (RNA-seq) on brain samples collected at 2 and 5 dpl.”

Reviewer #2 (Remarks to the Author):

*The manuscript by Zheng and colleagues addresses how prolonged visual experience impacts developmental synaptic scaling. Using *Xenopus laevis* tadpoles, exposed to prolonged ambient light (20h light/4 h dark - 20LE) or partial visual deprivation (4h light/20h dark - 4LE), the authors measured synaptic scaling during early brain development in tectal neurons. They found that prolonged exposure to light accelerated the developmental decline in glutamatergic synaptic transmission and increased neuronal excitability. Transcriptome analyses revealed transcriptional changes primed by prolonged light exposure, namely decreased levels of *GluA1* and *GluA2* AMPA receptor subunit mRNA after 2 days of prolonged light exposure (but not after longer periods). *GluA1* and *GluA2* protein levels were also decreased in synaptosomes after extended light exposure, whereas the small GTPase *Rab5c* was increased. These observations prompted analyzing the role of *Rab5c* in the synaptic scaling response to prolonged light exposure. Indeed, the authors found that overexpression of *Rab5c* decreased the amplitude of mEPSCs and evoked currents, whereas knock-down of *Rab5c* increased current amplitude. In a second part of the manuscript, neuronal histone acetylation was found to be upregulated in animals subjected to prolonged light exposure. Knock-down of the histone deacetylases 2 or 3 decreased the amplitude of mEPSCs, whereas inhibition of histone deacetylation increased intrinsic excitability and decreased *GluA1* and *GluA2* levels.*

Response: We sincerely thank the reviewer for their thoughtful and constructive comments. We greatly appreciate the recognition of the relevance and quality of our work, and we have carefully considered all the suggestions to improve the clarity, integration, and interpretation of the study. Below, we address each point raised.

The approach taken in this manuscript is interesting and relevant. Addressing synaptic scaling during development, and how it is impacted by ambient light exposure, is an important question. The work is well done and appropriately presented in the manuscript. Findings are relevant even if not surprising. The transcriptome dataset that was produced is useful for further studies but should at least be analyzed for enriched gene ontology categories.

Response:

We agree with the reviewer that enriched gene ontology analysis of the transcriptomic data is important for the study. The full list of analysis for BP, MF CC and KEGG was shown in the Table S5 in Excel format. The results reveal enrichment in categories related to membrane trafficking, synaptic organization, and receptor internalization. To directly visualize the distribution of transcriptome data, we have now performed a figure for Gene Ontology (GO) enrichment analysis on the differentially expressed genes at 2 dpl between 20LE and control groups. These analyses for BP, MF CC are now presented in a new Fig. S5, while volcano plot results are shown in Fig. S4. All GO and KEGG dataset were presented and summarized in the Results and Discussion sections.

*However, the more mechanistic aspects of the study (implication of *Rab5c* and histone acetylation) are not well integrated with the rest of the study, and need to be further substantiated. For example, it is not clear why the authors decided to look at *Rab5c* levels in*

light manipulation conditions (and not at one of the many transcripts they found altered in the transcriptomic screening).

Response: We appreciate the reviewer's request for greater mechanistic integration. Below, we clarify our rationale and summarize the additional data we have incorporated.

1. Selection of Rab5c

Although Rab5c did not appear among the top-ranking differentially expressed genes by fold change, an unbiased Gene Ontology enrichment analysis of our RNA-seq dataset revealed significant overrepresentation of vesicle-mediated transport and synaptic organization pathways (e.g., AMPA receptor activity, GO:0004971; glutamatergic synaptic transmission, GO:0035249; ionotropic glutamate receptor signaling, GO:0035235). These categories point toward endocytic and trafficking mechanisms as candidate regulators of prolonged light exposure-induced plasticity.

We chose to focus on Rab5c for several reasons. First, Rab5 family members are established key regulators of early endosome formation and AMPAR endocytosis, processes tightly linked to homeostatic synaptic scaling (Brown et al., 2005; Zhong et al., 2008). Second, Rab5 has been reported to play specialized roles in neuronal receptor trafficking and activity-dependent synaptic remodeling, making it a compelling candidate for mediating experience-driven changes in glutamatergic transmission. Third, our proteomic and biochemical analyses revealed a significant increase in Rab5c protein abundance in synaptosomal fractions following 20LE, despite the lack of strong transcriptional changes, suggesting that Rab5c is subject to post-transcriptional regulation during prolonged sensory stimulation. Fourth, Western blot and recording assays indicated that manipulating Rab5c levels altered both AMPAR expression and synaptic strength in ways consistent with scaling, strengthening its candidacy as a mechanistic effector.

Consistent with this focus, our RNA-seq dataset also contained multiple Rab5- and endocytosis-associated trafficking genes, including *ap2b1.S*, *ap2m1.L/S*, *ap3d1.S*, *clta.L/S*, *arf6ip1.L/S*, dynamin-related *dync1h1.L/S*, *dync1i2.S*, *nsf1c.L*, and *nsg1.L*, all of which are directly or indirectly involved in clathrin-mediated endocytosis, vesicle transport, or glutamate receptor internalization. In addition, AMPAR subunits themselves (e.g., *gria1.S*, *gria2.L/S*, *gria3.L*) were captured, providing further evidence that Rab5c operates within a broader receptor trafficking network highlighted by our transcriptomic screen (Table S1, 12LE vs 20LE).

Together, these convergent lines of evidence from GO enrichment pointing to trafficking pathways, to prior literature implicating Rab5 family proteins, to our own biochemical and functional validation leading us to focus on Rab5c as a central mediator of prolonged light exposure-induced synaptic scaling. We have now added these justifications to the Results sections.

Results (line: 190-193): “Because unbiased GO enrichment of our RNA-seq dataset highlighted vesicle-mediated transport and receptor trafficking pathways, and prior studies implicated Rab5 family proteins in AMPAR endocytosis via endosomes, we examined Rab5c and found that it was transiently upregulated at 2 dpl in 20LE, indicating a potential link between Rab5c and AMPAR trafficking under prolonged light exposure (Fig. 5b).”

2. Integration of histone acetylation

Although HDAC transcripts were unchanged in our RNA-seq dataset, we posited that sustained alterations in synaptic drive may recruit epigenetic mechanisms to stabilize gene expression programs. Consistent with this framework, our RNA-seq dataset contained multiple histone acetylation-associated regulators, including *h3-3a.L*, *h3f3b.L/S*, *chd4.L/S*, *chd8.S*, *ep400.L*, *kmt2a.L*, *kmt2d.S*, *jarid2.L/S*, *kdm6b.L/S*, *smarcc1.L*, *smarcc2.L*, *baz1b.S*, and *baz2b.L/S*, all of which are chromatin remodelers or modifiers known to influence histone acetylation and transcriptional accessibility (Table S1, 12LE vs 20LE). The presence of these genes underscores the close alignment between our RNA-seq signatures and the observed functional dependence on HDAC activity.

To test this, we performed knockdown experiments using HDAC2-MO and HDAC3-MO. HDAC2 knockdown did not affect Rab5c expression, but HDAC3 knockdown increased Rab5c protein levels, suggesting that HDAC3 normally represses Rab5c abundance. Importantly, both HDAC2 and HDAC3 knockdowns led to a significant reduction in GluA2 protein levels, demonstrating that histone acetylation regulates AMPAR subunit expression in parallel with, but mechanistically distinct from, Rab5c-mediated trafficking.

These results provide several additional insights. First, they reveal a cross-talk between epigenetic regulation and trafficking pathways, as HDAC3 can indirectly modulate Rab5c protein abundance. Second, they indicate that epigenetic control shapes the baseline availability of AMPAR subunits, while Rab5c governs the activity-dependent removal of these receptors from the surface. Third, the finding that 20LE increases histone acetylation while elevating Rab5c suggests that prolonged sensory experience engages a coordinated system: (i) transcriptional repression of AMPAR subunits through HDAC2/3-dependent histone acetylation, and (ii) accelerated receptor endocytosis through Rab5c upregulation. This dual mechanism ensures both the initiation and stabilization of synaptic scaling.

We have explicitly described this cascade in the revised manuscript and updated our working model (Fig. S10) to illustrate how prolonged light exposure engages both transcriptional (HDAC-dependent) and trafficking (Rab5c-mediated) pathways to achieve homeostatic downscaling.

At 2dpl, both 4LE and 20LE resulted in increased Rab5c levels, but only 20LE impacted synaptic scaling. The authors should discuss the significance of this observation.

Response: We thank the reviewer for highlighting this important observation. Indeed, our data show that Rab5c protein levels are increased at 2 dpl under both 4LE and 20LE conditions;

however, only 20LE is associated with decreased mEPSC amplitude and GluA2 downregulation. This indicates that Rab5c upregulation alone is not sufficient to drive synaptic scaling; rather, functional scaling depends on the interaction between Rab5c and the broader epigenetic and transcriptional landscape established by prolonged visual experience.

Specifically, at 2 dpl under 20LE, we observe robust increases in histone acetylation (H3K9ac, H2BK5ac, H4K8ac) and significant reductions in *GluA1/2* mRNA levels, indicating transcriptional repression of AMPAR subunits. This epigenetic shift likely creates a permissive environment in which Rab5c upregulation can effectively drive GluA2 internalization and reduce synaptic strength. In contrast, 4LE elevates Rab5c protein but does not trigger comparable histone acetylation or AMPAR transcriptional changes, limiting the functional impact of Rab5c on synaptic downscaling.

Thus, the divergence between 4LE and 20LE suggests that Rab5c serves as a necessary effector for AMPAR endocytosis, but sufficiency for synaptic scaling requires an additional epigenetic trigger emerging only after prolonged sensory drive. We have revised the Discussion to emphasize this coordinated mechanism: (1) transcriptional/epigenetic regulation of AMPAR subunits, and (2) Rab5c-mediated trafficking of GluA2-containing receptors, with both components required to achieve full synaptic scaling.

Finally, new experiments show that 0LE (complete visual deprivation) significantly increases mEPSC amplitude (Fig. S2), consistent with upward synaptic scaling. While the present manuscript focuses on 20LE-induced downscaling, future studies will be important to determine whether distinct or overlapping Rab5c/epigenetic mechanisms mediate 0LE-driven upscaling.

The following sentence has been added to the discussion of revised manuscript: “At 2 dpl, both 4LE and 20LE increase Rab5c protein levels; however, only 20LE reduces mEPSC amplitude and GluA2 levels, suggesting that Rab5c upregulation alone is insufficient for synaptic scaling. Early epigenetic remodeling under 20LE, marked by increased histone acetylation and reduced *GluA1/2* mRNA, creates a permissive environment for Rab5c-mediated GluA2 internalization, highlighting a coordinated mechanism in which transcriptional/epigenetic priming precedes receptor trafficking.”

Manipulating of Rab5c levels altered the neuronal architecture response to prolonged light exposure (Fig. 6), but these results are difficult to interpret, as 20LE increased Rab5c expression and increased total dendritic branch length, branch tip number and dendritic area, but Rab5c-GFP overexpression per se did not affect neuronal architecture. This suggests that the effects of 20LE on neuronal architecture are independent of Rab5c. On the other hand, knock-down of Rab5c increased dendritic complexity. These results show a complex effect of Rab5c, that does not seem to match what would be expected upon elevated Rab5c levels in 20LE exposed animals. The authors need to provide further explanation for these results, or to perform experiments to further investigate the role of Rab5c in the effects of prolonged light exposure on neuronal architecture.

Response: We thank the reviewer for highlighting this important point. We agree that the relationship between Rab5c expression and neuronal architecture under prolonged light exposure (20LE) is complex. Our data show that 20LE increases Rab5c protein levels and enhances dendritic branching, total branch length, branch tip number, and dendritic area. However, overexpression of Rab5c-GFP alone under 12LE conditions does not alter neuronal architecture, indicating that Rab5c elevation by itself is insufficient to drive dendritic growth. Interestingly, Rab5c-GFP overexpression under 20LE partially blocked the 20LE-induced increase in dendritic length, suggesting that Rab5c may act to constrain rather than promote excessive structural changes in the context of heightened activity.

Conversely, Rab5c knockdown increases dendritic complexity, potentially reflecting a compensatory response to impaired endosomal trafficking. This finding indicates that Rab5c contributes to maintaining dendritic homeostasis rather than directly driving growth. Together, these results suggest that Rab5c modulates the balance of dendritic structural plasticity in a context-dependent manner, with its functional impact shaped by the broader cellular environment induced by prolonged sensory experience, including epigenetic remodeling and AMPAR trafficking.

The following sentence has been added to the Discussion of revised manuscript: “Interestingly, Rab5c also modulates dendritic architecture in a context-dependent manner. While 20LE increases Rab5c levels and enhances total dendritic branch length, branch tip number, and dendritic area, Rab5c-GFP overexpression alone under 12LE does not alter dendritic architecture. Moreover, Rab5c-GFP overexpression under 20LE partially blocks the 20LE-induced increase in dendritic length, suggesting that Rab5c functions to restrain excessive structural growth and may require additional cofactors to coordinate neurite outgrowth. Conversely, Rab5c knockdown increases dendritic complexity, likely reflecting a compensatory response to impaired endosomal trafficking or a negative regulator of neurite growth. These observations indicate that Rab5c contributes to maintaining dendritic homeostasis rather than directly promoting growth, and its functional role is shaped by the cellular environment engaged during prolonged sensory experience.”

We acknowledge that additional experiments will be required to dissect the precise role of Rab5c in dendritic architecture under 20LE. For instance, combining Rab5c manipulation with synaptic marker expression to quantify synaptic density could clarify the relationship between dendritic structure and synaptic strength, and determine whether Rab5c acts permissively or synergistically in mediating dendritic remodeling. While these experiments are important, they extend beyond the current manuscript’s scope and will be pursued in future studies.

In Figure 7, the authors present data that they claim to suggest that Rab5c mediates AMPAR trafficking during synaptic scaling. This figure is confusing, because the first panels refer to the reversibility of the synaptic scaling effect. It would be more informative to start this figure by showing how Rab5c-MO or Rab5c-GFP impact synaptic scaling, which is never showed. In Fig. 7C-F, no comparison is made between 12LE and 20LE, to evaluate how manipulating Rab5c impacts synaptic scaling. As for panels A and B in this figure, the reversibility is

interesting (and expected, given previous studies), but again the role of Rab5c is difficult to appreciate, as the effect of manipulating Rab5c on scaling is not clear.

Response: We apologize that we did not show the figure clearly. We have reorganized and relabeled Fig. 7 to clearly show the impact of Rab5c manipulation on synaptic scaling and the reversibility. We have also added new panels directly comparing 12LE and 20LE responses in Rab5c-GFP and Rab5c-MO groups to strengthen the linkage between Rab5c manipulation and scaling. The figure legend has been revised accordingly.

Revised figure legend for Fig. 7a:

12LE (2 dpl) + 12LE (2 dpl): Tadpoles raised in 12LE for 4 days;

20LE (2 dpl) + 20LE (2 dpl): Raised in 20LE for 4 days;

20LE (2 dpl) + 12LE (2 dpl): Raised in 20LE for 2 days followed by 12LE for 2 days;

20LE (2 dpl) + Rab5c-MO (2 dpl): Raised in 20LE for 2 days followed by Rab5c-MO for 2 days;

20LE (2 dpl) + Rab5c-GFP (2 dpl): Raised in 20LE for 2 days followed by Rab5c-GFP for 2 days;

20LE (2 dpl) + Ctrl-MO (5 dpl): Raised in 20LE for 2 days followed by Ctrl-MO for 5 days;

20LE (2 dpl) + Rab5c-MO (5 dpl): Raised in 20LE for 2 days followed by Rab5c-MO for 5 days.

The authors then explore how prolonged light exposure affects histone acetylation (again, it is unclear why, besides the fact that it has been implicated before), and they found increased histone acetylation upon long-day exposure. They knocked down the expression of histone deacetylases (to increase histone acetylation), or used a pharmacological strategy to inhibit histone deacetylases, and found decreases excitatory transmission, increased excitability and decreased levels of GluA1 and GluA2. However, these experiments were all performed in 12LE. Therefore, it is not possible to claim, as the authors do, that “Prolonged light exposure reduces synaptic transmission and glutamatergic receptors through HDAC2/3 mediation”. For such claim, it would be necessary to test how manipulating histone acetylation impacts synaptic scaling triggered by prolonged light exposure.

Response: We thank the reviewer for this insightful comment. We agree that our initial HDAC manipulation experiments were performed under 12LE conditions and therefore did not directly address how histone acetylation contributes to synaptic scaling under prolonged light exposure. To address this limitation, we have now performed additional experiments in the 20LE paradigm. Specifically, tadpoles were exposed to 20LE for two days and then transfected with HDAC1-GFP, HDAC2-GFP, or HDAC3-GFP. In all cases, HDAC overexpression significantly increased mEPSC amplitudes compared with 20LE alone, effectively reversing the synaptic downscaling (Fig. 9c-d).

These results demonstrate that HDAC activity is both necessary and sufficient to mediate the homeostatic downscaling triggered by prolonged light exposure. Additionally, to clarify the relationship between trafficking and epigenetic mechanisms, we have added a comprehensive schematic (new Fig. S10) that integrates Rab5c-mediated AMPAR endocytosis, HDAC-

regulated receptor expression, and their combined roles in homeostatic synaptic adjustments under varying light conditions. Together, these results support our conclusion that prolonged light exposure reduces synaptic transmission and glutamatergic receptor levels through HDAC-mediated regulation of histone acetylation.

Reviewer #3 (Remarks to the Author):

This paper by Zheng et al is a nice demonstration of the value of the Xenopus optic tectum as a system for testing how sensory experience engages homeostatic plasticity mechanisms, exploiting anatomical, transcriptomic and physiological measurements. Although it represents a tour de force experimentally, the presentation of the considerable data is somewhat lacking in a coherent narrative. This detracts from the strength of the manuscript as a whole but does not make its interesting conclusions less compelling. . The key finding is that rearing animals in a circadian cycle with prolonged or shortened daylight periods results in profound changes in synaptic and intrinsic excitability of tectal neurons. A presumably homeostatic reduction in synaptic amplitude is shown in the 20LE condition. The effect is interesting, but given that the 4LE protocol did not really have an opposite effect, it is not entirely convincing that this is a homeostatic form of plasticity. The effects on intrinsic excitability were a nice addition, but mechanistically somewhat unsatisfying as they reported the phenomenology without really providing any insights into the underlying mechanism. Transcriptional analysis pointed to strong “homeostatic” regulation of AMPA receptor subunits and Rab5c, which presumably is important for receptor trafficking. Effects on dendritic branch size were also measured and a role for Rab5c suggested. Finally the importance of histone acetylation was examined. A clear increase in acetylation was observed in the 20LE group. This was curiously followed up by experiments in which knockdown of a subset of HDACs was performed, but for some reason the context of homeostatic plasticity was entirely abandoned for this experiment, leaving the result somewhat lacking context.

Overall it is an interesting and impressive series of studies, but the intended narrative about how epigenetic factors regulate homeostatic plasticity was not very clearly or thoroughly presented. These results are valuable and deserve publication, but it would be beneficial if a clear model could be synthesized at the end to bring all the findings together in a mechanistic narrative.

Response to Reviewer:

We sincerely thank the reviewer for their thoughtful and comprehensive comments. We appreciate the recognition of the strengths of our experimental approach and the value of the *Xenopus* optic tectum model for studying homeostatic plasticity. We have carefully revised the manuscript to address the reviewer's suggestions, improve the logical flow, strengthen mechanistic connections, and clarify methods and figures. Below, we provide point-by-point responses to each comment, with corresponding changes indicated.

Specific comments:

1. In55 turnovers -> turnover

Response: Corrected as suggested.

2. Line 73 discusses 3 HDACs but HDAC1 is not revisited in the rest of the paper. The justification for ignoring HDAC1 is not very strong.

Response: We thank the reviewer for highlighting the omission of HDAC1 in our earlier revision. In the updated manuscript, we now provide functional evidence that HDAC1 plays a comparable role to HDAC2/3 in regulating light-induced synaptic scaling.

We included new data showing that HDAC1-GFP overexpression at 2 dpl fully reverses the 20LE-induced reduction in mEPSC amplitude, similar to HDAC2-GFP and HDAC3-GFP (new Fig. 9c, d). These additions establish HDAC1 as an equally important regulator of light-driven histone deacetylation and subsequent Rab5c-mediated AMPAR trafficking. We have revised the Results and Discussion to emphasize that all three class I HDACs converge to drive transcriptional regulation of AMPAR subunits, thereby sustaining Rab5c-dependent receptor endocytosis and synaptic downscaling under prolonged visual experience.

3. Lines 94 - 96: The light exposure patterns began when the tadpoles were in stage 42. It might be useful to state why exactly they started at this specific developmental stage, and why they stopped at stage 49.

Response: We thank the reviewer for this suggestion. Differential light exposure was initiated at Nieuwkoop & Faber stage 42, when retinal axons first innervate the tectum, begin responding to visual stimulation, and visually driven behaviors emerge (Ref 1). Continuing light manipulation through stage 49 allowed us to span the entire critical-period window during which tectal circuits undergo their most pronounced sensory-experience-dependent refinement. Stage 49 represents the late tectal maturation phase just before 8 dpl in our timeline, ensuring that our 2 dpl, 5 dpl, and 8 dpl measurements capture early, intermediate, and late stages of experience-dependent plasticity without overlapping major metamorphic changes. The following sentence has been revised to:

“To explore how ambient light conditions impact neural development, we initiated differential light exposure in stage 42 tadpoles, when retinal axons first innervate the tectum, begin responding to visual stimulation, and visually driven behaviors emerge. Tadpoles were exposed to three ambient light conditions: control (12 hr light/12 hr dark, 12LE), prolonged light (20 hr light/4 hr dark, 20LE), and reduced light (4 hr light/20 hr dark, 4LE) for periods of 2, 5 and 8 days post light exposure (dpl), continuing through stage 49 to span the critical period of tectal circuit refinement (Fig. 1a).”.

Ref 1:

1. Aizenman, C.D. & Cline, H.T. Enhanced visual activity in vivo forms nascent synapses in the developing retinotectal projection. *J. Neurophysiol.* 97, 2949-2957 (2007).

4. In 97 please consider providing Nieuwkoop & Faber staging in addition to days post light

exposure.

Response:

Figure 1a indicates the developmental stage corresponds to each day-post-light exposure (dpl). In the method section, we have added the following sentence with reference: “Animal developmental stages were determined according to Nieuwkoop and Faber stage numbers”. We have also added corresponding Nieuwkoop and Faber stage numbers alongside dpl descriptions throughout the Results and Methods for clarity.”

5. Page 6 lin 134 *time observed points -> observed time points*

Response: Corrected to “observed time points.”

6. Lines 135-138: *When evaluating the membrane capacitance, input resistance and resting membrane potential, the results are only demonstrated for the 2 dpl group. Why weren't the 5 dpl or 8 dpl tested as well? Is it possible that the passive membrane properties take a longer period of light exposure to be significantly different? The conclusion drawn here seems to be reaching a little bit if they did only test these membrane properties after only 2 days of light exposure.*

Response: We thank the reviewer for this important suggestion. To address it, we have now measured passive membrane properties at 5 dpl and 8 dpl in addition to 2 dpl. These new data, presented in Supplementary Fig. 3, demonstrate that membrane capacitance (C_m), input resistance (R_{in}), and resting membrane potential (V_m) remain statistically unchanged across 4 LE, 12 LE, and 20 LE conditions at all three time points (2, 5, and 8 dpl). Therefore, we confirm that differential light exposure does not alter tectal neuron intrinsic membrane properties, even after extended durations. We have updated the Results section and the figure legend to reflect these additional measurements.

7. Line 174-176: *The authors in this section specify that the western blot was perform on tectal tissue in lines 223-224. It is unclear whether the western blot described by lines 174 - 176 were also performed using only tectal tissue. The sentence afterwards states that it was whole brain, but in that case it's unclear why the authors are making similar comparisons between whole brain homogenates and that from tectal tissue.*

Response: We apologize for the confusion. In *Xenopus laevis*, the optic tectum comprises the majority of the dorsal brain in tadpoles. During the tissue preparation, the olfactory bulb and hind brain were removed for Western blot. To avoid any ambiguity, we have now revised the manuscript so that all Western blots including those in lines 174-176, are clearly described as using dissected optic tectum homogenates. The Methods section and all figure legends have been updated to replace references to “whole brain” with “tectal” or “brain” exclusively, ensuring that all comparisons of Rab5c, HDACs, and histone-acetylation levels pertain to the optic tectum.

8. Line 228 on page 8 says “reinforcing the role of AMPAR trafficking” but the data are based on total expression of GluRs, not on cell surface expression, which would presumably require a biotinylation experiment.

Response: We thank the reviewer for this insightful comment. To better address the concern regarding surface versus total AMPAR expression, we performed additional Western blot analyses using synaptosome-enriched fractions (new Fig. 5c-e and Fig. 7g-i). These experiments revealed that the observed changes in GluA1 and GluA2 levels in total lysates were paralleled by corresponding changes in the synaptic fraction. Importantly, we confirmed that GluRs were predominantly localized in the synaptosomal compartment by comparing their abundance in synaptic versus cytosolic fractions. The integrity of the synaptosome preparation was validated by the enrichment of GluA1, GluA2, PSD95, and HDAC2 in the synaptic fraction relative to the cytosolic fraction (new Fig. 5c). Based on these new findings, we revised the manuscript to clarify that our conclusions are drawn from synaptically enriched AMPAR levels, rather than total surface expression. The original data on total AMPAR expression have been moved to Supplementary Fig.9.

9. HDAC inhibition results in Figure 9 are confusing as they don't reveal the blockade of a phenomenon but rather the shifting of baseline values in response to inhibition. These experiments would make much more sense in the context of testing whether the synaptic, excitability, morphology and most importantly transcriptomic changes associated with 20LE required HDAC activity or not. This feels like a lost opportunity and really breaks the logical flow of the narrative.

Response: We appreciate the reviewer's concern and agree that our original presentation of the HDAC inhibition experiments may have been confusing. We have revised the manuscript to clarify their purpose and integrate these results into a broader mechanistic framework.

First, the pharmacological HDAC inhibition experiments (TSA treatment) were designed to probe the general role of histone acetylation in excitatory synaptic function, rather than to directly block 20LE-induced scaling. TSA increased neuronal excitability (Fig. 9d) while decreasing GluA1 and GluA2 protein levels (Fig. 9e, f), indicating that HDAC activity is required to maintain baseline glutamatergic receptor levels and synaptic strength. Consistently, knockdown of HDAC2 or HDAC3 reduced mEPSC amplitude under 12LE, further supporting their role in basal excitatory transmission.

Second, to directly test HDAC involvement in 20LE-induced downscaling, we performed new experiments where tadpoles exposed to 20LE were transfected with HDAC1-GFP, HDAC2-GFP, or HDAC3-GFP. Overexpression significantly increased mEPSC amplitudes compared to 20LE controls, effectively reversing synaptic downscaling (new Fig. 9c, d). These results demonstrate that HDAC1/2/3 activity is both necessary and sufficient to regulate AMPAR availability and synaptic strength under prolonged visual experience.

Third, our transcriptomic analysis revealed no differential expression of HDAC1-3 mRNAs across light conditions (Supplementary Fig.6, Supplementary Fig7), consistent with a

model in which histone modifications, rather than transcriptional abundance, underlie their regulatory effects.

Together, these results support a coordinated model in which class I HDACs (HDAC1/2/3) provide an epigenetic mechanism for long-term suppression of AMPAR subunits, complementing Rab5c-dependent receptor trafficking of 20LE-induced synaptic scaling. While this manuscript focuses on 20LE, our new observation that 0LE increases mEPSC amplitude suggests that future studies should examine how HDAC-dependent regulation contributes to bidirectional synaptic scaling under different light paradigms.

10. Line 352: It should it be specified what the 0.02% MS-222 was dissolved in.

Response: We have specified that MS-222 was dissolved in Steinberg's solution. We now show: "For experimental manipulations, MS-222 (0.02%, Tricane methanesulfonate, Sigma-Aldrich) was dissolved in 0.1×Steinberg's solution and administered to anesthetize animals."

11. The methods on pg 14 provides an insufficient explanation of how cells were labeled and how MO was delivered. The methods simply say "injected into the tectum". I recommend moving the text from lines 365-370 down to this section.

Response: We thank the reviewer for highlighting this issue. The transfection methods were used for electrophysiological recordings and Western blot analysis, where plasmids or morpholinos were injected into the ventricular space of the optic tectum. For clarity, we kept this description to precede the relevant recording and Western blot methods.

We also recognize that the single-cell imaging transfection method differs from these applications. For sparse, single-neuron labeling, plasmids or morpholinos were pressure-injected directly into the tectal tissue, rather than the ventricle. This approach was followed by a single electroporation pulse (200 V/cm at peak, 1 ms) applied via laterally placed electrodes to achieve targeted transfection. We have expanded the Methods section to provide detailed descriptions of electroporation and morpholino delivery, including injection parameters, timing, and electrode placement.

The original sentence: "To transfect single neurons in vivo, plasmids or morpholinos were locally injected into the tectum of tadpoles." has been revised to:

"For single-neuron transfection, plasmids or morpholinos were pressure-injected into the tectal tissue, followed by a single electroporation pulse applied via laterally placed electrodes."

12. Similarly for GFP and Rab5c expression. On page 12 it says a BICS2-EGFP vector with a dual CMV promoter was used. It is unclear then whether the Rab5c was subcloned as a fusion protein with EGFP (as the nomenclature in the rest of the paper suggests) or for parallel expression with EGFP under an independent CMV promoter. I am guessing the latter but this needs to be clarified explicitly.

Response: We appreciate the reviewer's attention to this detail. Rab5c and EGFP were co-expressed as separate proteins under independent CMV promoters in the dual-promoter vector (BICS2-EGFP backbone). For simplicity in figures and text, we abbreviated this construct as Rab5c-GFP, but it does not represent a fusion protein. To avoid any confusion, the original sentence: "The Rab5c-GFP plasmid was constructed by subcloning Rab5c cDNA into a BICS2-EGFP vector with a dual CMV promoter." has been revised to: "The Rab5c-GFP plasmid (non-fusion) was generated by cloning Rab5c cDNA into the first CMV promoter site of a dual-CMV BICS2-EGFP vector, enabling parallel expression of Rab5c and EGFP."

Additional changes made: Updated figures/text to explicitly state "Rab5c-MO + Ctrl-GFP (co-expression with Rab5c-MO and Ctrl-GFP)" where applicable. We apologize for any confusion and thank the reviewer for prompting this clarification.

13. For morphometric analysis of tectal cells, how was "area" measured?

Response: We thank the reviewer for this query. Dendritic area measurements were performed on 3D-reconstructed tectal neurons traced from confocal stacks using Imaris software (Bitplane). This method accounts for the complex 3D geometry of dendrites. We have now explicitly detailed this approach in the Methods section with the following sentence:

"The dendritic arbor was segmented and converted into a 3D surface model, and the total dendritic area was quantified by converting the structure into a triangular mesh and summing the surface areas of all individual triangles."

14. Figure 1E, Was a two-way ANOVA used here? Did it show a significant interaction? Why does Figure 1E show all conditions in one plot but frequency is divided into 3 plots in F, G and H?

Response: For improved clarity and consistency with the frequency plots, we have reorganized the amplitude data. Specifically, we divided the original Fig. 1e into three separate plots to show amplitude changes at 2 dpl, 5 dpl, and 8 dpl individually. In addition, to facilitate a direct comparison of mEPSC amplitude changes during the development across these time points, we extracted the data from all three days and generated an additional summary figure (now presented as the new Supplementary Fig.1).

15. Figure 2E/F, label shows EPSC1/EPSC2 but figure legend says EPSC2/EPSC1 which appears correct. The figure should be corrected.

Response: Corrected the figure label as suggested to match the description in the legend (EPSC2/EPSC1).

16. Figure 4. It is difficult to determine how significant the reported changes are, and what genes are involved from this figure, although supplemental spreadsheets are included. Including volcano plots with the most significant genes labeled might be helpful.

Response: We thank the reviewer for this thoughtful suggestion. Although these genes exhibit relatively small log₂ fold changes and may not stand out prominently in volcano plots, we agree that such plots are useful for visualizing global transcriptomic shifts. Accordingly, we have added volcano plots for the 2 dpl and 5 dpl transcriptomic comparisons in a new Supplementary figure (Fig. S4). To further illustrate the biological relevance of these transcriptional changes, we also included the top enriched Gene Ontology (GO) categories for cellular component (CC), biological process (BP), and molecular function (MF) in Fig. S5 and Table S5. Full lists of differentially expressed genes, including *gria1.L* and *gria2.S*, are provided in Supplementary Tables 1-4.

17. *Figure 5. What is the basis for claiming the antibodies are specific? (pg 7 ln 176) Have they been validated with MO knockdown? The bands appear to be part of a larger smear of many proteins.*

Response: We appreciate the reviewer's concern regarding antibody specificity. Not all antibodies used in this study were validated by morpholino (MO) knockdown. The specificity of the Rab5c antibody was confirmed using a Rab5c-specific MO, with the corresponding Western blot result presented in Fig. S8a, b. For GluA1 and GluA2, we relied on prior peer-reviewed studies that validated these antibodies in *Xenopus* and other vertebrate systems, consistently detecting single bands at the expected molecular weights. In addition, these antibodies are widely used in the field and supported by manufacturer validation data.

We also note that while some background signal was present, the bands corresponding to the proteins of interest consistently appeared at the correct molecular weights and showed reproducible regulation across biological replicates, supporting their reliability. We have now marked the molecular weight on each band of the Western blots in the newly revised figures of the manuscript. This change should facilitate clearer interpretation of the protein sizes presented. To enhance clarity and avoid overstating the evidence, we have revised the wording in the manuscript from "...specific antibodies..." to "...antibodies against..." We have also ensured that full details of antibody sources, catalog numbers, and dilutions are included for transparency and reproducibility.

18. *Synaptosome preps should be validated by providing synaptic (e.g. PSD95) and non-synaptic (eg. Histone H3?) staining so we can assess the quality of the fractionation.*

Response:

We now provide Western blot data demonstrating enrichment of PSD-95 (a synaptic marker) and the exclusion of HDAC2 (a non-synaptic markers) in the synaptosomal fractions (Fig. 5c). As shown in the figure, synaptic proteins such as GluA1, GluA2 and PSD-95 are mainly located in the synaptosomal fraction, whereas non-synaptic proteins such as HDAC2 are largely absent. These results verify the purity and quality of our synaptosomal preparations.

19. *Fig 9A: The data purport to show that the amplitudes of the HDAC2- and HDAC3-MO conditions are smaller than the CTRL-MO. Visually looking at figure 9A, it seems as though the representative trace for the HDAC2-MO is larger in amplitude than that of the control. This*

might benefit from choosing a different representative trace.

Response:

We have replaced the representative traces for Ctrl-MO and HDAC2-MO in Fig. 9a to more accurately reflect the quantitative data.

20. Supplementary Figure 1 was not uploaded as far as I could tell. But it is critical to being able to interpret quality of data in figs 5 and 6. This must be corrected.

Response:

The Fig. S1 was uploaded properly in the revised manuscript.

Subject: Submission of Revised Manuscript - [COMMSBIO-24-8400B]

Second round reviewers' comments:

Reviewer #1 (Remarks to the Author):

The revised version of this manuscript is vastly improved. The authors have thoroughly addressed all my concerns. Highlights include a new super-helpful schematic that illustrates the model and brings all the finding together, and additional experiments which reveal a Rab5C-dependent and -independent pathway. Great job. The authors should be commended. I approve of this manuscript in its current form.

One minor note:

Line 87-88 : It is still stated that the observed increase in intrinsic excitability is driving the synaptic scaling (down). However, previous published work – for example Reference #18 of the manuscript – has shown that it is the decrease in synaptic input (due to polyamine block) that is driving the homeostatic increase in intrinsic excitability. This referenced work actually shows that the scaling down of synaptic input is what induces the scaling up of intrinsic excitability, not the other way around as is suggested here.

Response: Thank you for pointing this out. We have revised the sentence to “This synaptic scaling was driven by increased light exposure and Rab5c-mediated AMPAR removal.”. which specifying that elongated light exposure drives the synaptic scaling.

Reviewer #2 (Remarks to the Author):

Authors have appropriately addressed the issues that were previously raised, and the new experiments included are meaningful. The manuscript is now suitable for publication.

Response: We are grateful for the reviewer’s constructive guidance, which has greatly improved our work.

Reviewers' comments:

First round reviewers' comments:

Reviewer #1 (Remarks to the Author):

Summary: Here, Zheng and colleagues report that exposing tadpoles to prolonged periods of light (20LE) leads to a decrease in synaptic strengths received by tectal neurons along with an increase in intrinsic excitability. The decrease in synaptic strengths is interpreted as synaptic scaling. It is found through RNAseq and western blot analysis that prolonged exposure to light decreased the expression of the AMPA receptor subunits GluA1 and GluA2. This is consistent with the observed decrease in mEPSC amplitudes recorded from 20LE tectal neurons. The 20LE

group also showed an increase in Rab5c. It is unclear whether Rab5c was also identified through the RNAseq screen. Rab5c is known to regulate AMPA receptor cycling. In this study, it is determined that experimentally increasing Rab5c expression in tectal neurons dramatically reduces mEPSC amplitudes, and that inhibiting Rab5c increases mEPSC amplitudes. This suggests that prolonged light exposure decreases synaptic strength via an upregulation of Rab5c. Finally, it is determined that prolonged light exposure increases histone acetylation which is also found to decrease GluA1 and GluA2 levels. Overall, this study represents a huge amount of work. It seems it is several studies combined into one. While the finding that prolonged exposure to light induces downregulation of specific AMPA receptor subunits is notable, the study appears to lack logical flow/connectivity. For instance, what is the relationship between the histone acetylation mechanism and Rab5c? It is difficult to bring the different findings together. Furthermore, the data shown for several of the experiments appears to be incomplete. At the very least, the rationale for why certain experiments did not include certain time points (dpl) needs to be addressed.

Response:

We thank the reviewers for their careful reading and insightful suggestions. Below we have clarified the mechanistic integration of Rab5c and histone acetylation, and explained the rationale for our time-point selection.

1. Mechanistic link between Rab5c and histone acetylation

We propose that prolonged light exposure (20LE) initiates a coordinated homeostatic cascade. Elevated postsynaptic excitability under 20LE triggers activity-dependent epigenetic remodeling, marked by increased histone acetylation at H3K9, H2BK5, and H4K8. HDAC3-mediated upregulation of synaptosomal Rab5c enhances AMPA receptor (GluA2) endocytosis, thereby reducing excitatory synaptic drive. This sustained reduction in synaptic transmission reinforces and stabilizes the transcriptional downregulation of glutamate receptor expression, ensuring long-term homeostatic balance in the visual circuitry.

Rab5c acts as a trafficking effector. Under 20LE conditions at 5 dpl, Rab5c protein levels are elevated in synaptosomal fractions, and Rab5c overexpression or knockdown bidirectionally modulates mEPSC amplitude, confirming its direct involvement in AMPAR cycling and degradation. Consistent with this, Rab5c knockdown at 12LE enhances H3K9, H2BK5, and H4K8 acetylation (Fig.9n-q) and increases mEPSC amplitude (Fig.7a-c) and dendritic growth (Fig.6a-e). These findings support the idea that Rab5c-dependent trafficking changes occur first and help drive subsequent epigenetic remodeling.

In parallel, we observe robust histone acetylation upregulation under 20LE. Importantly, the temporal profile of AMPAR downregulation fits this coordinated model: GluA1/2 total protein levels are not reduced at 2 dpl, when Rab5c upregulation and early histone acetylation are already evident, but show marked decreases at 5 and 8 dpl (Fig. 5a, b). Moreover, HDAC2/HDAC3 knockdown or pharmacological HDAC inhibition decreases GluA2 protein levels (Fig. 9g-m) and further suppresses synaptic transmission. By contrast, HDAC3

knockdown selectively increases Rab5c protein expression (Fig. 9j-k), suggesting a novel regulatory axis where HDAC3 normally constrains Rab5c availability. This finding refines our model by indicating that HDACs influence synaptic scaling through two partially distinct mechanisms: (i) HDAC2/HDAC3 primarily regulates AMPAR subunit expression via chromatin remodeling, and (ii) HDAC3 additionally controls Rab5c protein levels, linking epigenetic regulation more directly to receptor trafficking.

Taken together, our findings support a feed-forward mechanism in which Rab5c-mediated AMPAR internalization triggers epigenetic modifications that progressively stabilize glutamate receptor downregulation, while HDAC3 provides a feedback mechanism capable of tuning Rab5c protein abundance. To illustrate this integrated model, we now provide a schematic in Fig. S10 and have expanded the Discussion accordingly.

2. Rationale for time-point selection

We focused much of our mechanistic analysis on the 2 days post-light exposure (2 dpl) window, corresponding to 48 hr of 20LE, for the following reasons: (1) it captures the peak of Rab5c-mediated AMPAR internalization and the onset of histone acetylation, before slower circuit-level compensations occur; (2) at 2 dpl, mEPSC amplitudes, Rab5c localization, and histone-acetylation changes are all synchronously altered, allowing direct correlation of trafficking and epigenetic events; (3) by 5-8 dpl, tectal neurons undergo complex structural remodeling (e.g., dendritic reorganization, network rewiring) that can obscure the primary, cell-intrinsic mechanisms we aimed to dissect. For these later stages, we restricted our analysis to AMPAR protein levels and mEPSCs, which clearly show that GluA1/2 downregulation emerges at 5 dpl and is maintained at 8 dpl (Fig. 5a, b), supporting the view that Rab5c and histone acetylation changes at 2 dpl initiate a longer-term receptor regulatory program. (4) Our “rescue” experiments (returning tadpoles from 20LE to 12LE) also started at 2 dpl, further justifying a uniform mechanistic time frame.

We have added the following sentence to the Discussion (Line: 352-355) to clarify how early and late time points align within our framework.

“We focused on the 2 dpl time point because it captures the peak of Rab5c-dependent AMPAR internalization and the onset of histone acetylation, when trafficking and epigenetic changes occur in parallel before circuit-level stabilization at later developmental stages.”

We hope these revisions clarify how Rab5c and histone acetylation act in concert to drive synaptic scaling under prolonged light exposure, and how our time-point selection supports both mechanistic precision and temporal resolution of AMPAR regulation.

Minor

Introduction

Line 48: “...such as exposure to 1 Hz for several hours...”

What is meant by this phrase? Exposure to 1 Hz of what? Please clarify this phrase.

Response: Thank you for pointing this out. We have revised the sentence to specify that this refers to visual stimulation at 1 Hz flickering light. The revised text now reads: “...such as prolonged exposure to 1 Hz flickering light for several hours...”

Line 56: What is “reversible scaling”? Please define. Also - “scaling” needs to be defined somewhere.

Response: We agree this term needed clarification. We now define synaptic scaling earlier in the introduction as: “Synaptic scaling refers to a form of homeostatic plasticity in which neurons adjust the strength of all their excitatory synapses up or down to stabilize overall activity.” “Reversible scaling” has been revised in the introduction to: “...particularly the mechanisms involved in reversible scaling, a bidirectional form of synaptic scaling that up- or down-regulates synaptic strength in response to altered and restored sensory input, is crucial for understanding the broader dynamics of activity-dependent synaptic regulation.”

Line 78-79: “...reduced miniature excitatory postsynaptic current (mEPSC) amplitudes in tectal neurons from stage 42 to 49 coincides to synaptic downscaling...”

The logic conveyed in this phrase seems circular, i.e. the decrease in mEPSC amplitudes defines the scaling down of synaptic currents. Thus, it is not clear how the decrease in mEPSC amplitudes can be interpreted to coincide with synaptic scaling since they are the same thing. This phrase needs to be rewritten.

Response: We agree that the logic was unclear. The sentence has been revised to: “...a progressive decrease in miniature excitatory postsynaptic current (mEPSC) amplitudes in tectal neurons from stage 42 to 49 exhibit a hallmark feature of synaptic downscaling during development.”

Line 79: “... synaptic downscaling...”

Is this “synaptic downscaling” the same thing as “reversible scaling” that appears in line 56? If not, please define this term. If so, please choose one of the terms and use consistently throughout the manuscript.

Response: Synaptic scaling includes synaptic upscaling and downscaling. Reversible scaling refers to a bidirectional form of synaptic scaling in which synaptic strengths can be up- or down-regulated in response to changes in sensory input, and subsequently reversed once normal input is restored. The two terms have been defined in the introduction in the revised manuscript. We now use the term “synaptic scaling” consistently throughout the manuscript and specify in each case whether it is up- or down-scaling to avoid confusion. The use of “reversible scaling” has been specified where experiments emerged as to restore the synaptic downscaling induced by 20 hr light exposure when tadpoles are returned to 12LE control condition.

Line 82-83: “the synaptic scaling was driven by Rab5c-mediated AMPAR removal and an increase in intrinsic excitability.” Is it actually determined that the increase in intrinsic excitability is driving the synaptic scaling and not that the decrease in mEPSC amplitudes is triggering the homeostatic increase in IE?

Response: Thank you for raising this. We clarify that the increase in intrinsic excitability drives synaptic scaling and triggers AMPAR trafficking. The sentence now reads: “Synaptic scaling was driven by an increase in intrinsic excitability and Rab5c-mediated AMPAR removal.”

Results

Line 91-92 the section “Prolonged light exposure accelerates a developmental decline in glutamatergic synaptic transmission.” Please add specifics on recording mEPSCs. What voltage were the neurons clamped at for recording these events? How were the excitatory events isolated from inhibitory ones?

Response: We thank the reviewer for pointing this out. We apologize for the earlier misstatement. The mEPSCs were recorded in voltage clamp at -60 mV in the presence of 1 μ M TTX without picrotoxin, and therefore GABAA receptor-mediated events were not pharmacologically blocked. However, at this holding potential, inhibitory postsynaptic currents are outward or near reversal and typically exhibit smaller amplitude, allowing us to selectively analyze inward excitatory events. We now clarify this in the Result section as follows: “mEPSCs were recorded in voltage clamp at -60 mV in the presence of 1 μ M extracellular TTX (Alomone Labs) to block action potentials. Inhibitory events are minimized at -60 mV holding potential and can be reliably distinguished from inward excitatory events based on polarity and kinetics.”

Line 112: “Prolonged light exposure decreases visually evoked synaptic responses.” Please include specifics on recording eCSCs. What voltage were the neurons held at for these recordings? How were the excitatory fraction of the visually evoked compound response isolated?

Response: We thank the reviewer for their comment and apologize for the confusion. Picrotoxin was not used in these experiments. Instead, tectal neurons were held at -60 mV, which allows for functional isolation of inhibitory versus excitatory synaptic components based on the reversal potentials of Cl⁻ and cations. As we and others have shown previously (Shen et al., 2011), excitatory responses appear as inward currents at -60 mV, while inhibitory GABAergic responses are outward, allowing us to differentiate components without pharmacological blockade. We have revised the text to reflect this in the Results: “... we recorded excitatory compound synaptic currents (eCSCs) from tectal neurons voltage-clamped at -60 mV...”, and the Methods “Excitatory and inhibitory components were distinguished based on their polarity and temporal profile at this holding potential, as described previously (Shen et al., 2011).”

Lines 150 and 151: 0.44% and 0.54% should be 44% and 54%, respectively.

Response: Corrected. Thank you for catching this.

Discussion

Line 243-244: “Prolonged light exposure reduces synaptic transmission and glutamatergic receptors through HDAC2/3 mediation.

While the data in this section do support this subheading, how does this fit with the overall title

of the manuscript which indicates that the scaling is via Rab5c? Is there a connection between the 20LE-induced increase in Rab5c and the altered pattern of histone acetylation?

Response: We thank the reviewer for raising this important point. To clarify the relationship between Rab5c and histone acetylation, we performed additional experiments. We found that HDAC2 knockdown does not affect Rab5c protein levels but significantly decreases GluA2 expression, as confirmed by Western blots at 12LE (Fig. 9j-k). By contrast, HDAC3 knockdown increases Rab5c protein expression while also reducing GluA1/GluA2 levels (Fig. 9j-m). These findings suggest that histone acetylation regulates AMPAR expression through both Rab5c-dependent and Rab5c-independent pathways.

Our time-course analyses delineate the temporal progression of these mechanisms. At 2 dpl, total GluA1/2 protein levels remain stable, although qRT-PCR and RNA-seq reveal reduced *GluA1/2* mRNA expression. At this early stage, histone acetylation at H3K9, H2BK5, and H4K8 is already elevated, coinciding with heightened neuronal spiking under 20LE. By 5-8 dpl, total GluA1/2 protein levels decline, synaptosomal Rab5c expression increases, and synaptic GluA2 is selectively reduced. Collectively, these observations support a sequential model in which activity-dependent histone acetylation precedes transcriptional suppression and subsequent receptor endocytosis, driving sustained synaptic downscaling.

Thus, Rab5c-dependent trafficking and HDAC2/3-mediated chromatin remodeling do not function in isolation but converge to downregulate AMPARs under prolonged light exposure. HDAC3 appears to exert a dual role by both promoting Rab5c upregulation and modulating epigenetic regulation, thereby integrating the two pathways. This mechanism accounts for the robust reduction in mEPSC amplitude observed at later time points and reconciles the distinct contributions of Rab5c and histone acetylation in synaptic scaling. We have revised the Results and Discussion to present this framework more clearly and now include a schematic model (Fig. S10) summarizing how Rab5c and epigenetic regulation act in concert to drive synaptic downscaling. We also changed the title to “Prolonged Visual Experience Accelerates Developmental Synaptic Downscaling via Epigenetic Regulation and Rab5c Mediated AMPA Receptor Trafficking” in our revised manuscript.

Lines 271-272: “...these findings underscore the pivotal role of epigenetic mechanisms, particularly Rab5c-mediated postsynaptic AMPAR trafficking”. How is Rab5c an epigenetic mechanism?

Response: We thank the reviewer for pointing out this imprecision. You are correct. Rab5c is not an epigenetic mechanism but rather a postsynaptic trafficking effector. In our revised text, we clarified this distinction and emphasized the temporal sequence between epigenetic regulation and trafficking events. Specifically, histone acetylation via HDAC2/3 acts first at the transcriptional level to reduce GluA1/2 expression, establishing the epigenetic basis for synaptic downregulation, whereas Rab5c mediates the subsequent rapid internalization of AMPARs to consolidate and maintain reduced synaptic strength.

Our updated experiments further refine this model. HDAC2 knockdown did not alter Rab5c expression but decreased GluA2 levels, indicating that HDAC2 acts through transcriptional regulation independently of Rab5c. In contrast, HDAC3 knockdown both increased Rab5c protein expression and reduced GluA1/GluA2 levels, suggesting that HDAC3 links epigenetic regulation and trafficking by enhancing Rab5c-dependent AMPAR endocytosis while simultaneously influencing gene transcription.

Thus, the updated sentence now reads:

“ These findings reveal a coordinated mechanism in which activity-dependent histone acetylation enhances Rab5c-mediated AMPAR trafficking. HDAC-dependent acetylation reinforces GluA2 downregulation and upregulates Rab5c expression, thereby linking epigenetic remodeling with receptor trafficking to stabilize homeostatic synaptic scaling and promote circuit maturation (Fig. S10).”

Line 122-123: “To determine if the observed mEPSC amplitude decrease was due to altered presynaptic release probability” A change in probability of presynaptic transmitter release would not be expected to alter mEPSC amplitude. Probability of release does, however, shape evoked responses. Perhaps this is what the authors mean to convey.

Response: We appreciate this clarification. The sentence has been revised to reflect that we assessed evoked responses, not mEPSC amplitude, to examine presynaptic release changes: “To determine if the developmental changes in excitatory currents was due to altered presynaptic release probability...”

Line 279: What is meant by “spaced pattern of stimulation”?

Response: Spaced pattern of stimulation refers to the 1 Hz visual stimulation, which was used to induce synaptic plasticity in *Xenopus* tectum. We now clarify this as: “...distinct from the rapid changes induced by patterned 1 Hz visual stimulation”

Line 284-285: “However, under partial sensory deprivation conditions...the regulatory effects on synaptic scaling appear to play a stabilizing role within the network without introducing the pronounced synaptic changes observed under more extreme conditions.” It is unclear what is being conveyed here. 4LE did not induce synaptic scaling. It seems not possible to interpret beyond that. What exactly is playing a stabilizing role? This sentence needs to be clarified or deleted.

Response: Thank you for raising this point. We have performed new experiments by recording tectal neurons under 0LE conditions and refined our analysis to directly address the distinction between partial (4LE) and complete (0LE) sensory deprivation and its implications for synaptic scaling:

4LE (partial deprivation): No change in mEPSC amplitude or frequency was observed, indicating that network activity remains near the homeostatic threshold. Under these mild

deprivation conditions, synaptic scaling mechanisms are not activated, thus preserving baseline synaptic function without inducing up- or down-scaling adaptations.

OLE (complete deprivation): Our newly obtained data (Supplementary Fig.2) demonstrate a significant increase in mEPSC amplitude, confirming robust homeostatic upscaling in response to more drastic reduction of sensory input.

We added the following sentence to the revised manuscript (line: 310-315): “Similarly, complete visual deprivation (OLE) robustly triggers homeostatic upscaling, as evidenced by a marked increase in mEPSC amplitude in OLE-treated tadpoles (Supplementary Fig. 2). In contrast, partial sensory deprivation (4LE) produces no significant changes in mEPSC amplitude or frequency. These findings suggest that only substantial reductions in afferent input activate synaptic scaling mechanisms, whereas mild deprivation remains below the threshold for homeostatic adjustment.”

This updated text clarifies that the stabilizing role refers to the maintenance of existing synaptic balance under mild deprivation, whereas complete deprivation invokes active synaptic scaling. This interpretation is consistent with established literature showing that global reductions in activity (e.g., via sensory deprivation or TTX blockade) trigger synaptic scaling mechanisms to restore neural firing rates.

Line 303-305: “Our findings indicate that Rab5c knockdown reduces mEPSC amplitudes, while Rab5c overexpression increases them and counters the effect of prolong light exposure...” This is the opposite of what the data show.

Response: Thank you for pointing this out. This was indeed reversed. It is now corrected to: “Our findings indicate that Rab5c knockdown increases mEPSC amplitudes, while Rab5c overexpression decreases them and mimics the effects of prolonged light exposure...”

Line 319-320 “The preferential reduction in CI-AMPA receptors aligns with synapse maturation during early brain development”. This is incorrect. This is not what at least one of the referenced works shows (ref. 8). Reference #8 shows that during development, there is a gradual decline in calcium-permeable AMPARs, not calcium-impermeable AMPARs. Please review the findings of the referenced work and correct this section of the Discussion.

Response: We thank the reviewer for pointing this out. Reference 8 shows that immature neurons express higher levels of calcium-permeable AMPARs (CP-AMPA receptors, lacking GluA2), which normally decline during development. In our study, prolonged visual experience (20LE) decreased both GluA1 and GluA2 protein expression at 5 dpl and 8 dpl, but not at 2 dpl (Fig. 5a, b). Notably, synaptosomal GluA2, but not GluA1, was specifically reduced at 5 dpl under 20LE. Consistently, both *GluA1* and *GluA2* mRNA levels were lower in 20LE compared to 12LE tadpoles at 2 dpl (Supplementary Fig.6a, b) but not at 5 dpl (Supplementary Fig.7a, b), indicating temporal control of transcriptional regulation. Mechanistically, these results suggest a dual process: (1) 20LE decreases synaptic GluA2 through Rab5c-dependent trafficking, and

(2) HDAC2/3 knockdown reduces GluA1/GluA2 expression (Fig. 9j-m), demonstrating that epigenetic regulation via histone acetylation further contributes to AMPAR subunit control.

Thus, rather than contradicting the developmental decline of CP-AMPA receptors described in Reference 8, our findings indicate that 20LE modifies this trajectory through a coordinated mechanism: Rab5c-mediated trafficking that decreases synaptic GluA2, while HDAC2/3-dependent epigenetic regulation that sustains reduced GluA2 expression.

We have revised the Discussion accordingly. The sentence below has been added to the revised manuscript (line: 371-380):

“ Our findings extend this developmental framework by showing that prolonged visual experience modifies this trajectory through coordinated transcriptional and trafficking mechanisms. Specifically, 20LE decreased both GluA1 and GluA2 protein levels at 5-8 dpl, with synaptosomal GluA2 showing a particularly strong reduction at 5 dpl. Transcriptome analysis revealed a transient downregulation of *GluA1* and *GluA2* mRNA at 2 dpl (Supplementary Fig.6), coinciding with elevated histone acetylation. These findings support a sequential regulatory model in which an early HDAC-regulated epigenetic phase suppresses GluA1/2 transcription, followed by a Rab5-dependent trafficking mechanism that removes synaptic GluA2, thereby consolidating the long-term downscaling of excitatory synaptic strength.”

Figure 6 The iMaris reconstructions shown for the panel A Ctrl-GFP neuron and panel B Rab5c-GFP neuron dendrites do not appear to match that of their corresponding raw microscopy images. Why was dendritic morphology studied only at the 2-3 dpl timepoint and not 5 and/or 8 dpl when the effect may have been more robust?

Response: We apologize for the confusion surrounding our reconstructions. Rab5c-GFP is expressed from a bicistronic CMV-driven construct, which labels both axons and dendrites. The raw microscopy images show both axons and dendrites, while in our tracings, axonal branches were intentionally omitted for clarity, but to prevent any misunderstanding we have now added more arrowheads in the raw micrographs to explicitly mark the axon origin.

With respect to the 2-3 dpl time point: we chose this early window because it coincides exactly with the functional hallmarks of Rab5c-dependent synaptic scaling—namely, the rapid reduction in mEPSC amplitude and the matched decrease in visually evoked responses. Our rescue experiments of recording were used 20LE-treated tadpoles for 2 days then returned to 12LE for 2 days. The fundamental logic was based on the 2 dpl timepoint. Furthermore, by 5 and 8 dpl, tectal neurons engage broader homeostatic and circuit-level adaptations (including compensatory dendritic stabilization and network rewiring) that can obscure the primary, endocytosis-driven remodeling we aim to dissect. Focusing on 2-3 dpl therefore allowed us to correlate, in the same cells, the earliest postsynaptic structural changes with our electrophysiological readouts under uniform imaging and tracing conditions. We have added this explanation to the revised Results sections to clarify why later time points were not included in the dendritic-morphology analysis.

“Neurons were imaged 2-3 days post-transfection, a window chosen because it coincides with the earliest functional signatures of Rab5c-dependent synaptic scaling, and aligns with our 2 dpl rescue experiments.”

Figure 7 panel B: The labeling of the x axis is confusing. Is the second group “12LE (2dpl)” really part of the 20LE (2dpl) group. If so, please explain more clearly in the legend. If not, please reorganize the graph to clearly show the different experimental groups.

Response: Thank you for raising this issue. You are correct: the “12LE (2 dpl)” group indeed follows a 48-hour exposure to 20LE, then a return to 12LE for 2 days. To clarify: The tadpoles were initially raised in 20LE for 2 days, and then either returned to 12LE or transfected with Rab5c-MO or Rab5c-GFP. To ensure this sequence is clearly communicated, we have updated both the figure and its legend.

Revised groups for Fig. 7a-c:

12LE (2 dpl) + 12LE (2 dpl): Tadpoles raised in 12LE for 4 days;

20LE (2 dpl) + 20LE (2 dpl): Raised in 20LE for 4 days;

20LE (2 dpl) + 12LE (2 dpl): Raised in 20LE for 2 days followed by 12LE for 2 days;

20LE (2 dpl) + Rab5c-MO (2 dpl): Raised in 20LE for 2 days followed by Rab5c knockdown for 2 days;

20LE (2 dpl) + Rab5c-GFP (2 dpl): Raised in 20LE for 2 days followed by Rab5c overexpression with Rab5c-GFP for 2 days;

20LE (2 dpl) + Ctrl-MO (5 dpl): Raised in 20LE for 2 days followed by Ctrl-MO for 5 days;

20LE (2 dpl) + Rab5c-MO (5 dpl): Raised in 20LE for 2 days followed by Rab5c-MO for 5 days.

Major:

Overall, the second half of the study appears to lack logical flow. From Figure 1 through Figure 4, the experimental progression is logical. From Figure 5 onwards it is difficult to understand the logical progression. For instance, in Figure 5, the authors follow up on the mRNAseq results by carrying out western blot for GluA1 and GluA2 to visualize protein expression levels (very nice) - but then here Rab5c is included. Then there is a sudden transient shift to studying the effect of Rab5c knock down and over-expression on dendrite morphology. Then an unexplained shift to the role of histone deacetylases on AMPAR expression. Is there a connection between histone deacetylation and Rab5c? It would be helpful if the authors included a schematic that combines all their findings into one model showing how 20LE is inducing a homeostatic decrease in mEPSC amplitudes. It also appears that either not all experiments were carried out at all the developmental time points or that not all the data is being shown (details on this point below).

Response: We thank the reviewer for this insightful comment. In response, we have substantially revised the Results and Discussion sections to improve the logical flow of the manuscript, particularly in the second half.

First, we clarified the mechanistic relationship between Rab5c and histone deacetylation pathways. Rab5c is a trafficking effector that regulates AMPAR endocytosis, whereas histone acetylation controls transcriptional programs that influence receptor subunit availability and Rab5c expression. Our new data refine this distinction. Specifically, HDAC2 knockdown did not affect Rab5c protein levels but significantly reduced GluA2 expression, indicating that HDAC2 functions primarily at the transcriptional level. In contrast, HDAC3 knockdown not only reduced GluA1/GluA2 proteins but also increased Rab5c expression, suggesting that HDAC3 integrates both pathways by coupling epigenetic remodeling with receptor trafficking. Together, these findings support a coordinated model: HDAC2/3-dependent histone acetylation regulates AMPAR subunit expression, while Rab5c governs receptor internalization and degradation at synaptic sites, with HDAC3 uniquely linking these two mechanisms.

Second, we integrated the temporal sequence of events under prolonged light exposure (20LE). At early stages (2 dpl), total GluA1/2 protein levels are unchanged, although GluA1/2 mRNA levels are significantly reduced, coinciding with increased acetylation of H3K9, H2BK5, and H4K8 and heightened tectal spiking activity. By 5-8 dpl, total GluA1/2 protein levels decline, with a selective reduction of GluA2 in synaptosomal fractions and elevated Rab5c expression, consistent with enhanced endocytosis. Functionally, mEPSC amplitude is decreased under 20LE, whereas Rab5c knockdown increases mEPSC amplitude. These results position Rab5c as a critical mediator of activity-dependent downscaling of excitatory drive. Our dendritic morphology data support this timeline: Rab5c knockdown increased dendritic length at 12LE, while Rab5c overexpression prevented 20LE-induced increase in dendritic length, reinforcing the link between Rab5c-dependent AMPAR trafficking and structural refinement.

Finally, we revised the Discussion to emphasize how these processes converge. Prolonged visual stimulation first triggers Rab5c-mediated removal of GluA2 from synaptic sites, driving the initial downscaling of excitatory synaptic strength. This reduction is subsequently reinforced by epigenetic remodeling via HDAC2/3-regulated histone acetylation, which decreases GluA1/2 transcription to maintain long-term synaptic downscaling. Notably, HDAC3 bridges these mechanisms by upregulating Rab5c, thereby enhancing receptor endocytosis while simultaneously contributing to transcriptional repression. Together, histone acetylation serves as the epigenetic trigger, Rab5c executes AMPAR trafficking, and HDAC3 integrates these parallel processes to coordinate sustained homeostatic scaling.

To help guide the reader, we have introduced Rab5c earlier in the manuscript and clarified the rationale for its inclusion, citing relevant literature on its role in AMPAR trafficking. We also now provide a schematic (new Fig. S10) that integrates all key findings into a single working model, illustrating how 20LE drives HDAC3-regulated GluA1/GluA2 downregulation and Rab5c-mediated AMPAR internalization across developmental time points.

We have added these clarifications throughout the revised manuscript, including in the Introduction, Methods, Results, and Figure legends. We believe these revisions address the reviewer's concerns and present a clearer, temporally resolved narrative linking Rab5c, histone acetylation, AMPAR regulation, and circuit homeostasis.

Line 179 -180: To identify the genes involved in the 20LE-induced homeostatic decrease in synaptic strengths, the authors carry out RNAseq, an unbiased approach. But then, the focus is on Rab5c, which does not appear to be identified in the RNAseq data. What is the motivation for looking at Rab5c protein levels? Was Rab5c pulled out of the RNAseq data? Rab5c RNA levels obtained from RNAseq should be shown. If not, there needs to be more justification of why, suddenly, Rab5C protein levels are being measured in this section.

Response: We thank the reviewer for raising this important point. Although Rab5c did not appear among the top-ranked differentially expressed transcripts in our RNA-seq dataset, unbiased gene ontology (GO) enrichment analysis revealed strong representation of categories related to membrane trafficking and synaptic organization (e.g., AMPA glutamate receptor activity, GO:0004971; glutamatergic synaptic transmission, GO:0035249; ionotropic glutamate receptor signaling, GO:0035235). Because Rab5 family proteins are well known regulators of AMPAR endocytosis and glutamatergic synapse remodeling (Brown et al., 2005; Zhong et al., 2008), we reasoned that Rab5c could serve as a key downstream effector even in the absence of robust mRNA changes.

To directly address this, we examined Rab5 isoforms by qRT-PCR and confirmed that Rab5a, Rab5b, and Rab5c transcript levels remain unchanged at both 2 dpl (Supplementary Fig.6) and 5 dpl (Supplementary Fig.7), consistent with our RNA-seq results (now included in Table S1-4). In contrast, *GluA1* and *GluA2* mRNAs were significantly reduced at 2 dpl (Fig. S6a, b) but returned toward baseline by 5 dpl (Fig. S7a, b). Importantly, despite the lack of Rab5c transcript changes, synaptosomal Western blots revealed robust upregulation of Rab5c protein under 20LE at 5 and 8 dpl, coinciding with selective loss of synaptosomal GluA2. At 2 dpl, total GluA1/2 protein levels remained unchanged, while histone acetylation marks (H3K9ac, H2BK5ac, H4K8ac) were already elevated and *GluA1/2* mRNA reduced, suggesting that transcriptional regulation is an earlier event, followed later by Rab5c-mediated trafficking.

Functional assays further validate this sequence: Rab5c knockdown increased mEPSC amplitude and dendritic length under 12LE, while Rab5c overexpression under 20LE decreased dendritic length. In contrast, 20LE itself decreased mEPSC amplitude, demonstrating that Rab5c activity is necessary for the synaptic downscaling phenotype. In parallel, HDAC2 knockdown reduced GluA2 protein levels without altering Rab5c expression, while HDAC3 knockdown reduced GluA1/GluA2 and increased Rab5c protein levels, suggesting that HDAC3 provides a point of cross-talk between epigenetic remodeling and trafficking regulation.

Taken together, these findings support a temporally structured, coordinated model: (1) early epigenetic remodeling (histone acetylation via HDAC2/3) suppresses AMPAR transcription at 2 dpl, and (2) Rab5c protein upregulation (enhanced by HDAC3 knockdown) drives synaptic GluA2 removal at 5-8 dpl, consolidating the sustained decrease in excitatory drive. We have now clarified this rationale in the Results, and added schematic diagrams (Supplementary Fig.10) to illustrate how Rab5c-dependent trafficking and HDAC2/3-mediated regulation cooperate in 20LE-induced synaptic scaling.

References

Brown, T.C., Tran, I.C., Backos, D.S., Esteban, J.A., 2005. NMDA Receptor-Dependent Activation of the Small GTPase Rab5 Drives the Removal of Synaptic AMPA Receptors during Hippocampal LTD. *Neuron* 45, 81-94.

Zhong, P., Liu, W.H., Gu, Z.L., Yan, Z., 2008. Serotonin facilitates long-term depression induction in prefrontal cortex via p38 MAPK/Rab5-mediated enhancement of AMPA receptor internalization. *Journal of Physiology* 586, 4465-4479.

Figure 2

Figure 1 shows mEPSC amplitudes and inter-event-intervals for 4LE, 12LE and 20LE at all 3 time points (2,5, and 8 dpl). Figure 2 shows the visually evoked data. In Figure 2, why are only the 2dpl data shown? It seems important to show the effect of 20LE and 4LE at the later time points (5 and 8 dpl) - since the tadpoles have been exposed to the altered stimuli for longer lengths of time. Plus, the decrease in mEPSC amplitude is most pronounced at 8dpl (2-star significance), so if the decrease in mEPSC amplitude is at least contributing to the decrease in visually evoked responses, the 8dpl time point would be predicted to be even more reduced. Thus, please add the 5 and 8 dpl data to Figure 2, or include in the text the reason for not including those data.

Response: We thank the reviewer for this thoughtful observation. Our decision to present only the 2 dpl data in Figure 2 was guided by our aim to isolate the early, cell-intrinsic mechanisms of synaptic scaling before the onset of broader circuit-level plasticity. Specifically, the 2 dpl window aligns with the most robust and direct alterations in postsynaptic physiology including the reduction in mEPSC amplitude and the downregulation of AMPARs, which we also validated via paired-pulse ratio analysis and transcriptomic data.

At 2 dpl, visually evoked responses are clearly reduced in the 20LE group, coinciding temporally with the earliest and least confounded expression of Rab5c-dependent AMPAR trafficking. This time point also matches the rescue conditions used throughout our study (tadpoles exposed to 20LE for 2 days and returned to 12LE), reinforcing its relevance across experiments.

While we agree that later time points are of biological interest, by 5 and 8 dpl, the optic tectum exhibits widespread structural and homeostatic changes including dendritic remodeling, spine turnover, and network reorganization that complicate interpretation of evoked responses in relation to early AMPAR trafficking alone. Indeed, our transcriptomic analysis at 5 dpl showed no significant difference in *GluA1* or *GluA2* expression across 4LE, 12LE, and 20LE groups (Fig. 4e and Supplementary Fig.7), suggesting that the initial transcriptional drivers of AMPAR downregulation have subsided. For these reasons, we chose to focus our mechanistic analysis on 2 dpl, when the causal link between prolonged light exposure and postsynaptic scaling is most direct and interpretable.

We have added this rationale to the revised Results (line: 215-216; Section: Rab5c regulates neuronal structure development) and Discussion (line: 352-355) to clarify our decision and to better contextualize the focus on early post-exposure dynamics.

Figure 4

Please include the rationale for why the RNAseq data at 8dpl is not shown here. If RNAseq was only carried out on 2 dpl and 5 dpl tissue, it is necessary to explain the rationale for focusing on these two time points.

Response: We have now clarified that RNA seq was performed at 2 and 5 dpl because transcriptomic dynamics and the gene-expression changes underlying early synaptic modifications peak within this window. By sampling at 2 dpl, we capture the onset of light induced signaling pathways that initiate synaptic scaling; by 5 dpl, we observe the initial progression of those responses. Indeed, glutamatergic receptor mRNA levels are significantly altered at 2 dpl but return toward baseline by 5 dpl, implying that subsequent synaptic adjustments rely more heavily on receptor protein trafficking and local translation than on further transcriptional regulation. Restricting sequencing to these two early time points allowed for deeper coverage, reduced biological variability, and minimized animal usage and experimental complexity, while keeping our focus on the early synaptic scaling mechanisms rather than on later developmental outcomes, which may include neurocircuit remodeling or behavioral adaptation. Accordingly, we did not perform RNA seq at 8 dpl, since transcriptomic profiles by that stage predominantly reflect stabilized, downstream consequences rather than the primary signaling events of interest. To corroborate the importance of this early window, we performed rescue experiments in which animals were returned from 20LE back to a normal 12LE cycle immediately after 2 dpl, further validating the critical role of early transcriptional responses in driving synaptic scaling. This rationale is now explained in the Methods and Results sections as shown above. The following texts have been added to the results section (line: 152-153): “To investigate how light exposure influences the onset of light induced-gene expression that may initiate synaptic scaling, we conducted RNA sequencing (RNA-seq) on brain samples collected at 2 and 5 dpl.”

Reviewer #2 (Remarks to the Author):

*The manuscript by Zheng and colleagues addresses how prolonged visual experience impacts developmental synaptic scaling. Using *Xenopus laevis* tadpoles, exposed to prolonged ambient light (20h light/4 h dark - 20LE) or partial visual deprivation (4h light/20h dark - 4LE), the authors measured synaptic scaling during early brain development in tectal neurons. They found that prolonged exposure to light accelerated the developmental decline in glutamatergic synaptic transmission and increased neuronal excitability. Transcriptome analyses revealed transcriptional changes primed by prolonged light exposure, namely decreased levels of *GluA1* and *GluA2* AMPA receptor subunit mRNA after 2 days of prolonged light exposure (but not after longer periods). *GluA1* and *GluA2* protein levels were also decreased in synaptosomes after extended light exposure, whereas the small GTPase *Rab5c* was increased. These observations prompted analyzing the role of *Rab5c* in the synaptic scaling response to prolonged light exposure. Indeed, the authors found that overexpression of *Rab5c* decreased the amplitude of mEPSCs and evoked currents, whereas knock-down of *Rab5c* increased current amplitude. In a second part of the manuscript, neuronal histone acetylation was found to be upregulated in animals subjected to prolonged light exposure. Knock-down of the histone*

deacetylases 2 or 3 decreased the amplitude of mEPSCs, whereas inhibition of histone deacetylation increased intrinsic excitability and decreased GluA1 and GluA2 levels.

Response: We sincerely thank the reviewer for their thoughtful and constructive comments. We greatly appreciate the recognition of the relevance and quality of our work, and we have carefully considered all the suggestions to improve the clarity, integration, and interpretation of the study. Below, we address each point raised.

The approach taken in this manuscript is interesting and relevant. Addressing synaptic scaling during development, and how it is impacted by ambient light exposure, is an important question. The work is well done and appropriately presented in the manuscript. Findings are relevant even if not surprising. The transcriptome dataset that was produced is useful for further studies but should at least be analyzed for enriched gene ontology categories.

Response:

We agree with the reviewer that enriched gene ontology analysis of the transcriptomic data is important for the study. The full list of analysis for BP, MF CC and KEGG was shown in the Table S5 in Excel format. The results reveal enrichment in categories related to membrane trafficking, synaptic organization, and receptor internalization. To directly visualize the distribution of transcriptome data, we have now performed a figure for Gene Ontology (GO) enrichment analysis on the differentially expressed genes at 2 dpl between 20LE and control groups. These analyses for BP, MF CC are now presented in a new Fig. S5, while volcano plot results are shown in Fig. S4. All GO and KEGG dataset were presented and summarized in the Results and Discussion sections.

However, the more mechanistic aspects of the study (implication of Rab5c and histone acetylation) are not well integrated with the rest of the study, and need to be further substantiated. For example, it is not clear why the authors decided to look at Rab5c levels in light manipulation conditions (and not at one of the many transcripts they found altered in the transcriptomic screening).

Response: We appreciate the reviewer's request for greater mechanistic integration. Below, we clarify our rationale and summarize the additional data we have incorporated.

1. Selection of Rab5c

Although Rab5c did not appear among the top-ranking differentially expressed genes by fold change, an unbiased Gene Ontology enrichment analysis of our RNA-seq dataset revealed significant overrepresentation of vesicle-mediated transport and synaptic organization pathways (e.g., AMPA receptor activity, GO:0004971; glutamatergic synaptic transmission, GO:0035249; ionotropic glutamate receptor signaling, GO:0035235). These categories point toward endocytic and trafficking mechanisms as candidate regulators of prolonged light exposure-induced plasticity.

We chose to focus on Rab5c for several reasons. First, Rab5 family members are established key regulators of early endosome formation and AMPAR endocytosis, processes tightly linked to homeostatic synaptic scaling (Brown et al., 2005; Zhong et al., 2008). Second, Rab5 has been reported to play specialized roles in neuronal receptor trafficking and activity-dependent synaptic remodeling, making it a compelling candidate for mediating experience-driven changes in glutamatergic transmission. Third, our proteomic and biochemical analyses revealed a significant increase in Rab5c protein abundance in synaptosomal fractions following 20LE, despite the lack of strong transcriptional changes, suggesting that Rab5c is subject to post-transcriptional regulation during prolonged sensory stimulation. Fourth, Western blot and recording assays indicated that manipulating Rab5c levels altered both AMPAR expression and synaptic strength in ways consistent with scaling, strengthening its candidacy as a mechanistic effector.

Consistent with this focus, our RNA-seq dataset also contained multiple Rab5- and endocytosis-associated trafficking genes, including *ap2b1.S*, *ap2m1.L/S*, *ap3d1.S*, *clta.L/S*, *arf6ip1.L/S*, dynamin-related *dync1h1.L/S*, *dync1i2.S*, *nsf1c.L*, and *nsg1.L*, all of which are directly or indirectly involved in clathrin-mediated endocytosis, vesicle transport, or glutamate receptor internalization. In addition, AMPAR subunits themselves (e.g., *gria1.S*, *gria2.L/S*, *gria3.L*) were captured, providing further evidence that Rab5c operates within a broader receptor trafficking network highlighted by our transcriptomic screen (Table S1, 12LE vs 20LE).

Together, these convergent lines of evidence from GO enrichment pointing to trafficking pathways, to prior literature implicating Rab5 family proteins, to our own biochemical and functional validation leading us to focus on Rab5c as a central mediator of prolonged light exposure-induced synaptic scaling. We have now added these justifications to the Results sections.

Results (line: 190-193): “Because unbiased GO enrichment of our RNA-seq dataset highlighted vesicle-mediated transport and receptor trafficking pathways, and prior studies implicated Rab5 family proteins in AMPAR endocytosis via endosomes, we examined Rab5c and found that it was transiently upregulated at 2 dpl in 20LE, indicating a potential link between Rab5c and AMPAR trafficking under prolonged light exposure (Fig. 5b).”

2. Integration of histone acetylation

Although HDAC transcripts were unchanged in our RNA-seq dataset, we posited that sustained alterations in synaptic drive may recruit epigenetic mechanisms to stabilize gene expression programs. Consistent with this framework, our RNA-seq dataset contained multiple histone acetylation-associated regulators, including *h3-3a.L*, *h3f3b.L/S*, *chd4.L/S*, *chd8.S*, *ep400.L*, *kmt2a.L*, *kmt2d.S*, *jarid2.L/S*, *kdm6b.L/S*, *smarcc1.L*, *smarcc2.L*, *baz1b.S*, and *baz2b.L/S*, all of which are chromatin remodelers or modifiers known to influence histone acetylation and transcriptional accessibility (Table S1, 12LE vs 20LE). The presence of these genes underscores the close alignment between our RNA-seq signatures and the observed functional dependence on HDAC activity.

To test this, we performed knockdown experiments using HDAC2-MO and HDAC3-MO. HDAC2 knockdown did not affect Rab5c expression, but HDAC3 knockdown increased Rab5c protein levels, suggesting that HDAC3 normally represses Rab5c abundance. Importantly, both HDAC2 and HDAC3 knockdowns led to a significant reduction in GluA2 protein levels, demonstrating that histone acetylation regulates AMPAR subunit expression in parallel with, but mechanistically distinct from, Rab5c-mediated trafficking.

These results provide several additional insights. First, they reveal a cross-talk between epigenetic regulation and trafficking pathways, as HDAC3 can indirectly modulate Rab5c protein abundance. Second, they indicate that epigenetic control shapes the baseline availability of AMPAR subunits, while Rab5c governs the activity-dependent removal of these receptors from the surface. Third, the finding that 20LE increases histone acetylation while elevating Rab5c suggests that prolonged sensory experience engages a coordinated system: (i) transcriptional repression of AMPAR subunits through HDAC2/3-dependent histone acetylation, and (ii) accelerated receptor endocytosis through Rab5c upregulation. This dual mechanism ensures both the initiation and stabilization of synaptic scaling.

We have explicitly described this cascade in the revised manuscript and updated our working model (Fig. S10) to illustrate how prolonged light exposure engages both transcriptional (HDAC-dependent) and trafficking (Rab5c-mediated) pathways to achieve homeostatic downscaling.

At 2dpl, both 4LE and 20LE resulted in increased Rab5c levels, but only 20LE impacted synaptic scaling. The authors should discuss the significance of this observation.

Response: We thank the reviewer for highlighting this important observation. Indeed, our data show that Rab5c protein levels are increased at 2 dpl under both 4LE and 20LE conditions; however, only 20LE is associated with decreased mEPSC amplitude and GluA2 downregulation. This indicates that Rab5c upregulation alone is not sufficient to drive synaptic scaling; rather, functional scaling depends on the interaction between Rab5c and the broader epigenetic and transcriptional landscape established by prolonged visual experience.

Specifically, at 2 dpl under 20LE, we observe robust increases in histone acetylation (H3K9ac, H2BK5ac, H4K8ac) and significant reductions in *GluA1/2* mRNA levels, indicating transcriptional repression of AMPAR subunits. This epigenetic shift likely creates a permissive environment in which Rab5c upregulation can effectively drive GluA2 internalization and reduce synaptic strength. In contrast, 4LE elevates Rab5c protein but does not trigger comparable histone acetylation or AMPAR transcriptional changes, limiting the functional impact of Rab5c on synaptic downscaling.

Thus, the divergence between 4LE and 20LE suggests that Rab5c serves as a necessary effector for AMPAR endocytosis, but sufficiency for synaptic scaling requires an additional epigenetic trigger emerging only after prolonged sensory drive. We have revised the Discussion to emphasize this coordinated mechanism: (1) transcriptional/epigenetic regulation of AMPAR subunits, and (2) Rab5c-mediated trafficking of GluA2-containing receptors, with both

components required to achieve full synaptic scaling.

Finally, new experiments show that 0LE (complete visual deprivation) significantly increases mEPSC amplitude (Fig. S2), consistent with upward synaptic scaling. While the present manuscript focuses on 20LE-induced downscaling, future studies will be important to determine whether distinct or overlapping Rab5c/epigenetic mechanisms mediate 0LE-driven upscaling.

The following sentence has been added to the discussion of revised manuscript: “At 2 dpl, both 4LE and 20LE increase Rab5c protein levels; however, only 20LE reduces mEPSC amplitude and GluA2 levels, suggesting that Rab5c upregulation alone is insufficient for synaptic scaling. Early epigenetic remodeling under 20LE, marked by increased histone acetylation and reduced GluA1/2 mRNA, creates a permissive environment for Rab5c-mediated GluA2 internalization, highlighting a coordinated mechanism in which transcriptional/epigenetic priming precedes receptor trafficking.”

Manipulating of Rab5c levels altered the neuronal architecture response to prolonged light exposure (Fig. 6), but these results are difficult to interpret, as 20LE increased Rab5c expression and increased total dendritic branch length, branch tip number and dendritic area, but Rab5c-GFP overexpression per se did not affect neuronal architecture. This suggests that the effects of 20LE on neuronal architecture are independent of Rab5c. On the other hand, knock-down of Rab5c increased dendritic complexity. These results show a complex effect of Rab5c, that does not seem to match what would be expected upon elevated Rab5c levels in 20LE exposed animals. The authors need to provide further explanation for these results, or to perform experiments to further investigate the role of Rab5c in the effects of prolonged light exposure on neuronal architecture.

Response: We thank the reviewer for highlighting this important point. We agree that the relationship between Rab5c expression and neuronal architecture under prolonged light exposure (20LE) is complex. Our data show that 20LE increases Rab5c protein levels and enhances dendritic branching, total branch length, branch tip number, and dendritic area. However, overexpression of Rab5c-GFP alone under 12LE conditions does not alter neuronal architecture, indicating that Rab5c elevation by itself is insufficient to drive dendritic growth. Interestingly, Rab5c-GFP overexpression under 20LE partially blocked the 20LE-induced increase in dendritic length, suggesting that Rab5c may act to constrain rather than promote excessive structural changes in the context of heightened activity.

Conversely, Rab5c knockdown increases dendritic complexity, potentially reflecting a compensatory response to impaired endosomal trafficking. This finding indicates that Rab5c contributes to maintaining dendritic homeostasis rather than directly driving growth. Together, these results suggest that Rab5c modulates the balance of dendritic structural plasticity in a context-dependent manner, with its functional impact shaped by the broader cellular environment induced by prolonged sensory experience, including epigenetic remodeling and AMPAR trafficking.

The following sentence has been added to the Discussion of revised manuscript: “Interestingly, Rab5c also modulates dendritic architecture in a context-dependent manner. While 20LE increases Rab5c levels and enhances total dendritic branch length, branch tip number, and dendritic area, Rab5c-GFP overexpression alone under 12LE does not alter dendritic architecture. Moreover, Rab5c-GFP overexpression under 20LE partially blocks the 20LE-induced increase in dendritic length, suggesting that Rab5c functions to restrain excessive structural growth and may require additional cofactors to coordinate neurite outgrowth. Conversely, Rab5c knockdown increases dendritic complexity, likely reflecting a compensatory response to impaired endosomal trafficking or a negative regulator of neurite growth. These observations indicate that Rab5c contributes to maintaining dendritic homeostasis rather than directly promoting growth, and its functional role is shaped by the cellular environment engaged during prolonged sensory experience.”

We acknowledge that additional experiments will be required to dissect the precise role of Rab5c in dendritic architecture under 20LE. For instance, combining Rab5c manipulation with synaptic marker expression to quantify synaptic density could clarify the relationship between dendritic structure and synaptic strength, and determine whether Rab5c acts permissively or synergistically in mediating dendritic remodeling. While these experiments are important, they extend beyond the current manuscript’s scope and will be pursued in future studies.

In Figure 7, the authors present data that they claim to suggest that Rab5c mediates AMPAR trafficking during synaptic scaling. This figure is confusing, because the first panels refer to the reversibility of the synaptic scaling effect. It would be more informative to start this figure by showing how Rab5c-MO or Rab5c-GFP impact synaptic scaling, which is never showed. In Fig. 7C-F, no comparison is made between 12LE and 20LE, to evaluate how manipulating Rab5c impacts synaptic scaling. As for panels A and B in this figure, the reversibility is interesting (and expected, given previous studies), but again the role of Rab5c is difficult to appreciate, as the effect of manipulating Rab5c on scaling is not clear.

Response: We apologize that we did not show the figure clearly. We have reorganized and relabeled Fig. 7 to clearly show the impact of Rab5c manipulation on synaptic scaling and the reversibility. We have also added new panels directly comparing 12LE and 20LE responses in Rab5c-GFP and Rab5c-MO groups to strengthen the linkage between Rab5c manipulation and scaling. The figure legend has been revised accordingly.

Revised figure legend for Fig. 7a:

12LE (2 dpl) + 12LE (2 dpl): Tadpoles raised in 12LE for 4 days;

20LE (2 dpl) + 20LE (2 dpl): Raised in 20LE for 4 days;

20LE (2 dpl) + 12LE (2 dpl): Raised in 20LE for 2 days followed by 12LE for 2 days;

20LE (2 dpl) + Rab5c-MO (2 dpl): Raised in 20LE for 2 days followed by Rab5c-MO for 2 days;

20LE (2 dpl) + Rab5c-GFP (2 dpl): Raised in 20LE for 2 days followed by Rab5c-GFP for 2 days;

20LE (2 dpl) + Ctrl-MO (5 dpl): Raised in 20LE for 2 days followed by Ctrl-MO for 5 days;

20LE (2 dpl) + Rab5c-MO (5 dpl): Raised in 20LE for 2 days followed by Rab5c-MO for 5 days.

The authors then explore how prolonged light exposure affects histone acetylation (again, it is unclear why, besides the fact that it has been implicated before), and they found increased histone acetylation upon long-day exposure. They knocked down the expression of histone deacetylases (to increase histone acetylation), or used a pharmacological strategy to inhibit histone deacetylases, and found decreases excitatory transmission, increased excitability and decreased levels of GluA1 and GluA2. However, these experiments were all performed in 12LE. Therefore, it is not possible to claim, as the authors do, that “Prolonged light exposure reduces synaptic transmission and glutamatergic receptors through HDAC2/3 mediation”. For such claim, it would be necessary to test how manipulating histone acetylation impacts synaptic scaling triggered by prolonged light exposure.

Response: We thank the reviewer for this insightful comment. We agree that our initial HDAC manipulation experiments were performed under 12LE conditions and therefore did not directly address how histone acetylation contributes to synaptic scaling under prolonged light exposure. To address this limitation, we have now performed additional experiments in the 20LE paradigm. Specifically, tadpoles were exposed to 20LE for two days and then transfected with HDAC1-GFP, HDAC2-GFP, or HDAC3-GFP. In all cases, HDAC overexpression significantly increased mEPSC amplitudes compared with 20LE alone, effectively reversing the synaptic downscaling (Fig. 9c-d).

These results demonstrate that HDAC activity is both necessary and sufficient to mediate the homeostatic downscaling triggered by prolonged light exposure. Additionally, to clarify the relationship between trafficking and epigenetic mechanisms, we have added a comprehensive schematic (new Fig. S10) that integrates Rab5c-mediated AMPAR endocytosis, HDAC-regulated receptor expression, and their combined roles in homeostatic synaptic adjustments under varying light conditions. Together, these results support our conclusion that prolonged light exposure reduces synaptic transmission and glutamatergic receptor levels through HDAC-mediated regulation of histone acetylation.

Reviewer #3 (Remarks to the Author):

This paper by Zheng et al is a nice demonstration of the value of the Xenopus optic tectum as a system for testing how sensory experience engages homeostatic plasticity mechanisms, exploiting anatomical, transcriptomic and physiological measurements. Although it represents a tour de force experimentally, the presentation of the considerable data is somewhat lacking in a coherent narrative. This detracts from the strength of the manuscript as a whole but does not make its interesting conclusions less compelling. . The key finding is that rearing animals in a circadian cycle with prolonged or shortened daylight periods results in profound changes in synaptic and intrinsic excitability of tectal neurons. A presumably homeostatic reduction in synaptic amplitude is shown in the 20LE condition. The effect is interesting, but given that the 4LE protocol did not really have an opposite effect, it is not entirely convincing that this is a homeostatic form of plasticity. The effects on intrinsic excitability were a nice addition, but

mechanistically somewhat unsatisfying as they reported the phenomenology without really providing any insights into the underlying mechanism. Transcriptional analysis pointed to strong “homeostatic” regulation of AMPA receptor subunits and Rab5c, which presumably is important for receptor trafficking. Effects on dendritic branch size were also measured and a role for Rab5c suggested. Finally the importance of histone acetylation was examined. A clear increase in acetylation was observed in the 20LE group. This was curiously followed up by experiments in which knockdown of a subset of HDACs was performed, but for some reason the context of homeostatic plasticity was entirely abandoned for this experiment, leaving the result somewhat lacking context.

Overall it is an interesting and impressive series of studies, but the intended narrative about how epigenetic factors regulate homeostatic plasticity was not very clearly or thoroughly presented. These results are valuable and deserve publication, but it would be beneficial if a clear model could be synthesized at the end to bring all the findings together in a mechanistic narrative.

Response to Reviewer:

We sincerely thank the reviewer for their thoughtful and comprehensive comments. We appreciate the recognition of the strengths of our experimental approach and the value of the *Xenopus* optic tectum model for studying homeostatic plasticity. We have carefully revised the manuscript to address the reviewer's suggestions, improve the logical flow, strengthen mechanistic connections, and clarify methods and figures. Below, we provide point-by-point responses to each comment, with corresponding changes indicated.

Specific comments:

1. In55 turnovers -> turnover

Response: Corrected as suggested.

2. Line 73 discusses 3 HDACs but HDAC1 is not revisited in the rest of the paper. The justification for ignoring HDAC1 is not very strong.

Response: We thank the reviewer for highlighting the omission of HDAC1 in our earlier revision. In the updated manuscript, we now provide functional evidence that HDAC1 plays a comparable role to HDAC2/3 in regulating light-induced synaptic scaling.

We included new data showing that HDAC1-GFP overexpression at 2 dpl fully reverses the 20LE-induced reduction in mEPSC amplitude, similar to HDAC2-GFP and HDAC3-GFP (new Fig. 9c, d). These additions establish HDAC1 as an equally important regulator of light-driven histone deacetylation and subsequent Rab5c-mediated AMPAR trafficking. We have revised the Results and Discussion to emphasize that all three class I HDACs converge to drive transcriptional regulation of AMPAR subunits, thereby sustaining Rab5c-dependent receptor endocytosis and synaptic downscaling under prolonged visual experience.

3. Lines 94 - 96: The light exposure patterns began when the tadpoles were in stage 42. It might

be useful to state why exactly they started at this specific developmental stage, and why they stopped at stage 49.

Response: We thank the reviewer for this suggestion. Differential light exposure was initiated at Nieuwkoop & Faber stage 42, when retinal axons first innervate the tectum, begin responding to visual stimulation, and visually driven behaviors emerge (Ref 1). Continuing light manipulation through stage 49 allowed us to span the entire critical-period window during which tectal circuits undergo their most pronounced sensory-experience-dependent refinement. Stage 49 represents the late tectal maturation phase just before 8 dpl in our timeline, ensuring that our 2 dpl, 5 dpl, and 8 dpl measurements capture early, intermediate, and late stages of experience-dependent plasticity without overlapping major metamorphic changes. The following sentence has been revised to:

“To explore how ambient light conditions impact neural development, we initiated differential light exposure in stage 42 tadpoles, when retinal axons first innervate the tectum, begin responding to visual stimulation, and visually driven behaviors emerge. Tadpoles were exposed to three ambient light conditions: control (12 hr light/12 hr dark, 12LE), prolonged light (20 hr light/4 hr dark, 20LE), and reduced light (4 hr light/20 hr dark, 4LE) for periods of 2, 5 and 8 days post light exposure (dpl), continuing through stage 49 to span the critical period of tectal circuit refinement (Fig. 1a).”

Ref 1:

1. Aizenman, C.D. & Cline, H.T. Enhanced visual activity in vivo forms nascent synapses in the developing retinotectal projection. *J. Neurophysiol.* 97, 2949-2957 (2007).

4. In 97 please consider providing Nieuwkoop & Faber staging in addition to days post light exposure.

Response:

Figure 1a indicates the developmental stage corresponds to each day-post-light exposure (dpl). In the method section, we have added the following sentence with reference: “Animal developmental stages were determined according to Nieuwkoop and Faber stage numbers”. We have also added corresponding Nieuwkoop and Faber stage numbers alongside dpl descriptions throughout the Results and Methods for clarity.”

5. Page 6 lin 134 time observed points -> observed time points

Response: Corrected to “observed time points.”

6. Lines 135-138: When evaluating the membrane capacitance, input resistance and resting membrane potential, the results are only demonstrated for the 2 dpl group. Why weren't the 5 dpl or 8 dpl tested as well? Is it possible that the passive membrane properties take a longer period of light exposure to be significantly different? The conclusion drawn here seems to be reaching a little bit if they did only test these membrane properties after only 2 days of light

exposure.

Response: We thank the reviewer for this important suggestion. To address it, we have now measured passive membrane properties at 5 dpl and 8 dpl in addition to 2 dpl. These new data, presented in Supplementary Fig. 3, demonstrate that membrane capacitance (C_m), input resistance (R_{in}), and resting membrane potential (V_m) remain statistically unchanged across 4 LE, 12 LE, and 20 LE conditions at all three time points (2, 5, and 8 dpl). Therefore, we confirm that differential light exposure does not alter tectal neuron intrinsic membrane properties, even after extended durations. We have updated the Results section and the figure legend to reflect these additional measurements.

7. Line 174-176: The authors in this section specify that the western blot was performed on tectal tissue in lines 223-224. It is unclear whether the western blot described by lines 174 - 176 were also performed using only tectal tissue. The sentence afterwards states that it was whole brain, but in that case it's unclear why the authors are making similar comparisons between whole brain homogenates and that from tectal tissue.

Response: We apologize for the confusion. In *Xenopus laevis*, the optic tectum comprises the majority of the dorsal brain in tadpoles. During the tissue preparation, the olfactory bulb and hind brain were removed for Western blot. To avoid any ambiguity, we have now revised the manuscript so that all Western blots including those in lines 174-176, are clearly described as using dissected optic tectum homogenates. The Methods section and all figure legends have been updated to replace references to “whole brain” with “tectal” or “brain” exclusively, ensuring that all comparisons of Rab5c, HDACs, and histone-acetylation levels pertain to the optic tectum.

8. Line 228 on page 8 says “reinforcing the role of AMPAR trafficking” but the data are based on total expression of GluRs, not on cell surface expression, which would presumably require a biotinylation experiment.

Response: We thank the reviewer for this insightful comment. To better address the concern regarding surface versus total AMPAR expression, we performed additional Western blot analyses using synaptosome-enriched fractions (new Fig. 5c-e and Fig. 7g-i). These experiments revealed that the observed changes in GluA1 and GluA2 levels in total lysates were paralleled by corresponding changes in the synaptic fraction. Importantly, we confirmed that GluRs were predominantly localized in the synaptosomal compartment by comparing their abundance in synaptic versus cytosolic fractions. The integrity of the synaptosome preparation was validated by the enrichment of GluA1, GluA2, PSD95, and HDAC2 in the synaptic fraction relative to the cytosolic fraction (new Fig. 5c). Based on these new findings, we revised the manuscript to clarify that our conclusions are drawn from synaptically enriched AMPAR levels, rather than total surface expression. The original data on total AMPAR expression have been moved to Supplementary Fig. 9.

9. HDAC inhibition results in Figure 9 are confusing as they don't reveal the blockade of a phenomenon but rather the shifting of baseline values in response to inhibition. These

experiments would make much more sense in the context of testing whether the synaptic, excitability, morphology and most importantly transcriptomic changes associated with 20LE required HDAC activity or not. This feels like a lost opportunity and really breaks the logical flow of the narrative.

Response: We appreciate the reviewer's concern and agree that our original presentation of the HDAC inhibition experiments may have been confusing. We have revised the manuscript to clarify their purpose and integrate these results into a broader mechanistic framework.

First, the pharmacological HDAC inhibition experiments (TSA treatment) were designed to probe the general role of histone acetylation in excitatory synaptic function, rather than to directly block 20LE-induced scaling. TSA increased neuronal excitability (Fig. 9d) while decreasing GluA1 and GluA2 protein levels (Fig. 9e, f), indicating that HDAC activity is required to maintain baseline glutamatergic receptor levels and synaptic strength. Consistently, knockdown of HDAC2 or HDAC3 reduced mEPSC amplitude under 12LE, further supporting their role in basal excitatory transmission.

Second, to directly test HDAC involvement in 20LE-induced downscaling, we performed new experiments where tadpoles exposed to 20LE were transfected with HDAC1-GFP, HDAC2-GFP, or HDAC3-GFP. Overexpression significantly increased mEPSC amplitudes compared to 20LE controls, effectively reversing synaptic downscaling (new Fig. 9c, d). These results demonstrate that HDAC1/2/3 activity is both necessary and sufficient to regulate AMPAR availability and synaptic strength under prolonged visual experience.

Third, our transcriptomic analysis revealed no differential expression of HDAC1-3 mRNAs across light conditions (Supplementary Fig.6, Supplementary Fig7), consistent with a model in which histone modifications, rather than transcriptional abundance, underlie their regulatory effects.

Together, these results support a coordinated model in which class I HDACs (HDAC1/2/3) provide an epigenetic mechanism for long-term suppression of AMPAR subunits, complementing Rab5c-dependent receptor trafficking of 20LE-induced synaptic scaling. While this manuscript focuses on 20LE, our new observation that 0LE increases mEPSC amplitude suggests that future studies should examine how HDAC-dependent regulation contributes to bidirectional synaptic scaling under different light paradigms.

10. Line 352: It should it be specified what the 0.02% MS-222 was dissolved in.

Response: We have specified that MS-222 was dissolved in Steinberg's solution. We now show: "For experimental manipulations, MS-222 (0.02%, Tricane methanesulfonate, Sigma-Aldrich) was dissolved in 0.1×Steinberg's solution and administered to anesthetize animals."

11. The methods on pg 14 provides an insufficient explanation of how cells were labeled and how MO was delivered. The methods simply say "injected into the tectum". I recommend moving the text from lines 365-370 down to this section.

Response: We thank the reviewer for highlighting this issue. The transfection methods were used for electrophysiological recordings and Western blot analysis, where plasmids or morpholinos were injected into the ventricular space of the optic tectum. For clarity, we kept this description to precede the relevant recording and Western blot methods.

We also recognize that the single-cell imaging transfection method differs from these applications. For sparse, single-neuron labeling, plasmids or morpholinos were pressure-injected directly into the tectal tissue, rather than the ventricle. This approach was followed by a single electroporation pulse (200 V/cm at peak, 1 ms) applied via laterally placed electrodes to achieve targeted transfection. We have expanded the Methods section to provide detailed descriptions of electroporation and morpholino delivery, including injection parameters, timing, and electrode placement.

The original sentence: “To transfect single neurons in vivo, plasmids or morpholinos were locally injected into the tectum of tadpoles.” has been revised to:

“For single-neuron transfection, plasmids or morpholinos were pressure-injected into the tectal tissue, followed by a single electroporation pulse applied via laterally placed electrodes.”

12. Similarly for GFP and Rab5c expression. On page 12 it says a BICS2-EGFP vector with a dual CMV promoter was used. It is unclear then whether the Rab5c was subcloned as a fusion protein with EGFP (as the nomenclature in the rest of the paper suggests) or for parallel expression with EGFP under an independent CMV promoter. I am guessing the latter but this needs to be clarified explicitly.

Response: We appreciate the reviewer’s attention to this detail. Rab5c and EGFP were co-expressed as separate proteins under independent CMV promoters in the dual-promoter vector (BICS2-EGFP backbone). For simplicity in figures and text, we abbreviated this construct as Rab5c-GFP, but it does not represent a fusion protein. To avoid any confusion, the original sentence: “The Rab5c-GFP plasmid was constructed by subcloning Rab5c cDNA into a BICS2-EGFP vector with a dual CMV promoter.” has been revised to: “The Rab5c-GFP plasmid (non-fusion) was generated by cloning Rab5c cDNA into the first CMV promoter site of a dual-CMV BICS2-EGFP vector, enabling parallel expression of Rab5c and EGFP.”.

Additional changes made: Updated figures/text to explicitly state "Rab5c-MO + Ctrl-GFP (co-expression with Rab5c-MO and Ctrl-GFP)" where applicable. We apologize for any confusion and thank the reviewer for prompting this clarification.

13. For morphometric analysis of tectal cells, how was “area” measured?

Response: We thank the reviewer for this query. Dendritic area measurements were performed on 3D-reconstructed tectal neurons traced from confocal stacks using Imaris software (Bitplane). This method accounts for the complex 3D geometry of dendrites. We have now explicitly detailed this approach in the Methods section with the following sentence:

“The dendritic arbor was segmented and converted into a 3D surface model, and the total dendritic area was quantified by converting the structure into a triangular mesh and summing the surface areas of all individual triangles.”.

14. *Figure 1E, Was a two-way ANOVA used here? Did it show a significant interaction? Why does Figure 1E show all conditions in one plot but frequency is divided into 3 plots in F, G and H?*

Response: For improved clarity and consistency with the frequency plots, we have reorganized the amplitude data. Specifically, we divided the original Fig. 1e into three separate plots to show amplitude changes at 2 dpl, 5 dpl, and 8 dpl individually. In addition, to facilitate a direct comparison of mEPSC amplitude changes during the development across these time points, we extracted the data from all three days and generated an additional summary figure (now presented as the new Supplementary Fig.1).

15. *Figure 2E/F, label shows EPSC1/EPSC2 but figure legend says EPSC2/EPSC1 which appears correct. The figure should be corrected.*

Response: Corrected the figure label as suggested to match the description in the legend (EPSC2/EPSC1).

16. *Figure 4. It is difficult to determine how significant the reported changes area, and what genes are involved from this figure, although supplemental spreadsheets are included. Including volcano plots with the most significant genes labeled might be helpful.*

Response: We thank the reviewer for this thoughtful suggestion. Although these genes exhibit relatively small log₂ fold changes and may not stand out prominently in volcano plots, we agree that such plots are useful for visualizing global transcriptomic shifts. Accordingly, we have added volcano plots for the 2 dpl and 5 dpl transcriptomic comparisons in a new Supplementary figure (Fig. S4). To further illustrate the biological relevance of these transcriptional changes, we also included the top enriched Gene Ontology (GO) categories for cellular component (CC), biological process (BP), and molecular function (MF) in Fig. S5 and Table S5. Full lists of differentially expressed genes, including *gria1.L* and *gria2.S*, are provided in Supplementary Tables 1-4.

17. *Figure 5. What is the basis for claiming the antibodies are specific? (pg 7 ln 176) Have they been validated with MO knockdown? The bands appear to be part of a larger smear of many proteins.*

Response: We appreciate the reviewer’s concern regarding antibody specificity. Not all antibodies used in this study were validated by morpholino (MO) knockdown. The specificity of the Rab5c antibody was confirmed using a Rab5c-specific MO, with the corresponding Western blot result presented in Fig. S8a, b. For GluA1 and GluA2, we relied on prior peer-reviewed studies that validated these antibodies in *Xenopus* and other vertebrate systems,

consistently detecting single bands at the expected molecular weights. In addition, these antibodies are widely used in the field and supported by manufacturer validation data.

We also note that while some background signal was present, the bands corresponding to the proteins of interest consistently appeared at the correct molecular weights and showed reproducible regulation across biological replicates, supporting their reliability. We have now marked the molecular weight on each band of the Western blots in the newly revised figures of the manuscript. This change should facilitate clearer interpretation of the protein sizes presented. To enhance clarity and avoid overstating the evidence, we have revised the wording in the manuscript from "...specific antibodies..." to "...antibodies against..." We have also ensured that full details of antibody sources, catalog numbers, and dilutions are included for transparency and reproducibility.

18. Synaptosome preps should be validated by providing synaptic (e.g. PSD95) and non-synaptic (eg. Histone H3?) staining so we can assess the quality of the fractionation.

Response:

We now provide Western blot data demonstrating enrichment of PSD-95 (a synaptic marker) and the exclusion of HDAC2 (a non-synaptic markers) in the synaptosomal fractions (Fig. 5c). As shown in the figure, synaptic proteins such as GluA1, GluA2 and PSD-95 are mainly located in the synaptosomal fraction, whereas non-synaptic proteins such as HDAC2 are largely absent. These results verify the purity and quality of our synaptosomal preparations.

19. Fig 9A: The data purport to show that the amplitudes of the HDAC2- and HDAC3-MO conditions are smaller than the CTRL-MO. Visually looking at figure 9A, it seems as though the representative trace for the HDAC2-MO is larger in amplitude than that of the control. This might benefit from choosing a different representative trace.

Response:

We have replaced the representative traces for Ctrl-MO and HDAC2-MO in Fig. 9a to more accurately reflect the quantitative data.

20. Supplementary Figure 1 was not uploaded as far as I could tell. But it is critical to being able to interpret quality of data in figs 5 and 6. This must be corrected.

Response:

The Fig. S1 was uploaded properly in the revised manuscript.